# Understanding and Improving Graph Injection Attack by Promoting Unnoticeability

**Yongqiang Chen**[1]**, Han Yang**[1]**, Yonggang Zhang**[2]**, Kaili Ma**[1]
[1]The Chinese University of Hong Kong [2]Hong Kong Baptist University
{yqchen,hyang,klma,jcheng}@cse.cuhk.edu.hk csygzhang@comp.hkbu.edu.hk
**Tongliang Liu**[3]**, Bo Han**[2]**, James Cheng**[1]
[3]The University of Sydney
tongliang.liu@sydney.edu.au bhanml@comp.hkbu.edu.hk

## Abstract

Recently Graph Injection Attack (GIA) emerges as a practical attack scenario on Graph Neural Networks (GNNs), where the adversary can merely inject few malicious nodes instead of modifying existing nodes or edges, i.e., Graph Modification Attack (GMA). Although GIA has achieved promising results, little is known about why it is successful and whether there is any pitfall behind the success. To understand the power of GIA, we compare it with GMA and find that GIA can be provably more harmful than GMA due to its relatively high flexibility. However, the high flexibility will also lead to great damage to the homophily distribution of the original graph, i.e., similarity among neighbors. Consequently, the threats of GIA can be easily alleviated or even prevented by homophily-based defenses designed to recover the original homophily. To mitigate the issue, we introduce a novel constraint – *homophily unnoticeability* that enforces GIA to preserve the homophily, and propose Harmonious Adversarial Objective (HAO) to instantiate it. Extensive experiments verify that GIA with HAO can break homophily-based defenses and outperform previous GIA attacks by a significant margin. We believe our methods can serve for a more reliable evaluation of the robustness of GNNs.

## 1 Introduction

Graph Neural Networks (GNNs), as a generalization of deep learning models for graph structured data, have gained great success in tasks involving relational information (Hamilton et al., 2017a; Battaglia et al., 2018; Zhou et al., 2020; Wu et al., 2021; Kipf & Welling, 2017; Hamilton et al., 2017b; Veličković et al., 2018; Xu et al., 2018; 2019b). Nevertheless, GNNs are shown to be inherently vulnerable to adversarial attacks (Sun et al., 2018; Jin et al., 2021), or small intentional perturbations on the input (Szegedy et al., 2014). Previous studies show that moderate changes to the existing topology or node features of the input graph, i.e., Graph Modification Attacks (GMA), can dramatically degenerate the performance of GNNs (Dai et al., 2018; Zügner et al., 2018; Zügner & Günnemann, 2019; Xu et al., 2019a; Chang et al., 2020). Since in many real-world scenarios, it is prohibitively expensive to modify the original graph, recently there is increasing attention paid to Graph Injection Attack (GIA), where the adversary can merely inject few malicious nodes to perform the attack (Wang et al., 2018; Sun et al., 2020; Wang et al., 2020; Zou et al., 2021).

Despite the promising empirical results, why GIA is booming and whether there is any pitfall behind the success remain elusive. To bridge this gap, we investigate both the advantages and limitations of GIA by comparing it with GMA in a unified setting (Sec. 2.2). Our theoretical results show that, in this setting when there is no defense, GIA can be provably more harmful than GMA due to its relatively high flexibility. Such flexibility enables GIA to map GMA perturbations into specific GIA perturbations, and to further optimize the mapped perturbations to amplify the damage (Fig. 1a). However, according to the principle of no free lunch, we further find that the power of GIA is built upon the severe damage to the homophily of the original graph. Homophily indicates the tendency of nodes connecting to others with similar features or labels, which is important for the success of most existing GNNs (McPherson et al., 2001; London & Getoor, 2014; Klicpera et al., 2019; Battaglia

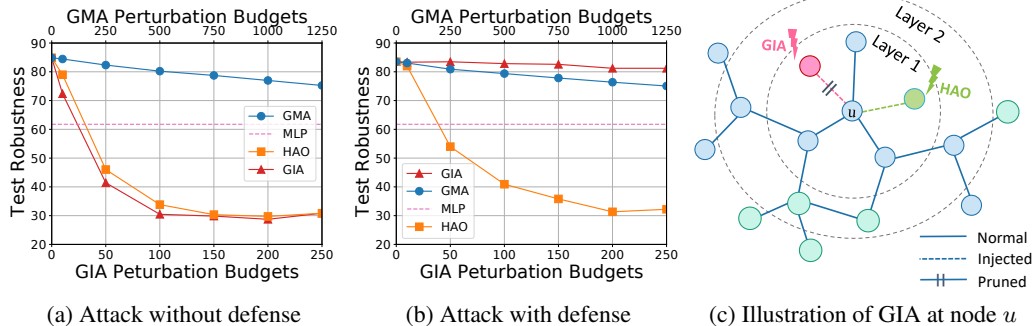

(a) Attack without defense     (b) Attack with defense     (c) Illustration of GIA at node $u$

Figure 1: The lower test robustness indicates better attack performance. (a) Without defenses: GIA performs consistently better than GMA; (b) With defenses: GIA without HAO performs consistently worse than GMA, while GIA with HAO performs the best; (c) Homophily indicates the tendency of similar nodes connecting with each other (blue & green nodes). The malicious (red) nodes and edges injected by GIA without HAO will greatly break the homophily hence can be easily identified and pruned by homophily defenders. GIA with HAO is aware of preserving homophily that attacks the targets by injecting unnoticeable (more similar) but still adversarial (dark green) nodes and edges, which will not be easily pruned hence effectively causing the damage.

et al., 2018; Hou et al., 2020; Zhu et al., 2020; Yang et al., 2021b). The severe damage to homophily will disable the effectiveness of GIA to evaluate robustness because non-robust models can easily mitigate or even prevent GIA merely through exploiting the property of homophily damage.

Specifically, having observed the destruction of homophily, it is straightforward to devise a defense mechanism aiming to recover the homophily, which we term *homophily defenders*. Homophily defenders are shown to have strong robustness against GIA attacks. Theoretically, they can effectively reduce the harm caused by GIA to be lower than GMA. Empirically, simple implementations of homophily defenders with edge pruning (Zhang & Zitnik, 2020) can deteriorate even the state-of-the-art GIA attacks (Zou et al., 2021) (Fig. 1b). Therefore, overlooking the damage to homophily will make GIA powerless and further limit its applications for evaluating the robustness of GNNs.

To enable the effectiveness of GIA in evaluating various robust GNNs, it is necessary to be aware of preserving the homophily when developing GIA. To this end, we introduce a novel constraint – *homophily unnoticeability* that enforces GIA to retain the homophily of the original graph, which can serve as a supplementary for the unnoticeability constraints in graph adversarial learning. To instantiate the homophily unnoticeability, we propose Harmonious Adversarial Objective (HAO) for GIA (Fig. 1c). Specifically, HAO introduces a novel differentiable realization of homophily constraint by regularizing the homophily distribution shift during the attack. In this way, adversaries will not be easily identified by homophily defenders while still performing effective attacks (Fig. 1b). Extensive experiments with 38 defense models on 6 benchmarks demonstrate that GIA with HAO can break homophily defenders and significantly outperform all previous works across all settings, including both non-target attack and targeted attack[1]. Our contributions are summarized as follows:

- We provide a formal comparison between GIA and GMA in a unified setting and find that GIA can be provably more harmful than GMA due to its high flexibility (Theorem 1).

- However, the flexibility of GIA will also cause severe damage to the homophily distribution which makes GIA easily defendable by homophily defenders (Theorem 2).

- To mitigate the issue, we introduce the concept of homophily unnoticeability and a novel objective HAO to conduct homophily unnoticeable attacks (Theorem 3).

## 2 PRELIMINARIES

### 2.1 GRAPH NEURAL NETWORKS

Consider a graph $\mathcal{G} = (A, X)$ with node set $V = \{v_1, v_2, ..., v_n\}$ and edge set $E = \{e_1, e_2, ..., e_m\}$, where $A \in \{0, 1\}^{n \times n}$ is the adjacency matrix and $X \in \mathbb{R}^{n \times d}$ is the node feature matrix. We are

---

[1]Code is available in https://github.com/LFhase/GIA-HAO.

interested in the semi-supervised node classification task (Jin et al., 2021). That is, given the set of labels $Y \in \{0, 1, .., C - 1\}^n$ from $C$ classes, we can train a graph neural network $f_\theta$ parameterized by $\theta$ on the training (sub)graph $\mathcal{G}_{\text{train}}$ by minimizing a classification loss $\mathcal{L}_{\text{train}}$ (e.g., cross-entropy). Then the trained $f_\theta$ can predict the labels of nodes in test graph $\mathcal{G}_{\text{test}}$. A GNN typically follows a neighbor aggregation scheme to recursively update the node representations as:

$$H_u^{(k)} = \sigma(W_k \cdot \rho(\{H_v^{(k-1)}\} | v \in \mathcal{N}(u) \cup \{u\})), \tag{1}$$

where $\mathcal{N}(u)$ is the set of neighbors of node $u$, $H_u^{(0)} = X_u, \forall u \in V$, $H_u^{(k)}$ is the hidden representation of node $u$ after the $k$-th aggregation, $\sigma(\cdot)$ is an activation function, e.g., ReLU, and $\rho(\cdot)$ is an aggregation function over neighbors, e.g., MEAN or SUM.

## 2.2 GRAPH ADVERSARIAL ATTACK[2]

The goal of graph adversarial attack is to fool a GNN model, $f_{\theta^*}$, trained on a graph $\mathcal{G} = (A, X)$ by constructing a graph $\mathcal{G}' = (A', X')$ with limited budgets $\|\mathcal{G}' - \mathcal{G}\| \leq \triangle$. Given a set of victim nodes $V_c \subseteq V$, the graph adversarial attack can be generically formulated as:

$$\min \mathcal{L}_{\text{atk}}(f_{\theta^*}(\mathcal{G}')), \text{ s.t. } \|\mathcal{G}' - \mathcal{G}\| \leq \triangle, \tag{2}$$

where $\theta^* = \arg\min_\theta \mathcal{L}_{\text{train}}(f_\theta(\mathcal{G}_{\text{train}}))$ and $\mathcal{L}_{\text{atk}}$ is usually taken as $-\mathcal{L}_{\text{train}}$. Following previous works (Zügner et al., 2018; Zou et al., 2021), Graph adversarial attacks can be characterized into graph modification attacks and graph injection attacks by their perturbation constraints.

**Graph Modification Attack (GMA).** GMA generates $\mathcal{G}'$ by modifying the graph structure $A$ and the node features $X$ of the original graph $\mathcal{G}$. Typically the constraints in GMA are to limit the number of perturbations on $A$ and $X$, denoted by $\triangle_A$ and $\triangle_X$, respectively, as:

$$\triangle_A + \triangle_X \leq \triangle \in \mathbb{Z}, \|A' - A\|_0 \leq \triangle_A \in \mathbb{Z}, \|X' - X\|_\infty \leq \epsilon \in \mathbb{R}, \tag{3}$$

where the perturbation on $X$ is bounded by $\epsilon$ via L-p norm, since we are using continuous features.

**Graph Injection Attack (GIA).** Differently, GIA generates $\mathcal{G}'$ by injecting a set of malicious nodes $V_{\text{atk}}$ as $X' = \begin{bmatrix} X \\ X_{\text{atk}} \end{bmatrix}, A' = \begin{bmatrix} A & A_{\text{atk}} \\ A_{\text{atk}}^T & O_{\text{atk}} \end{bmatrix}$, where $X_{\text{atk}}$ is the features of the injected nodes, $O_{\text{atk}}$ is the adjacency matrix among injected nodes, and $A_{\text{atk}}$ is the adjacency matrix between the injected nodes and the original nodes. Let $d_u$ denote the degree of node $u$, the constraints in GIA are:

$$|V_{\text{atk}}| \leq \triangle \in \mathbb{Z}, \ 1 \leq d_u \leq b \in \mathbb{Z}, X_u \in \mathcal{D}_X \subseteq \mathbb{R}^d, \forall u \in V_{\text{atk}}, \tag{4}$$

where the number and degrees of the injected nodes are limited, $\mathcal{D}_X = \{C \in \mathbb{R}^d, \min(X) \cdot \mathbf{1} \leq C \leq \max(X) \cdot \mathbf{1}\}$ where $\min(X)$ and $\max(X)$ are the minimum and maximum entries in $X$ respectively.

**Threat Model.** We adopt a unified setting, i.e., evasion, inductive and black-box, which is also used by Graph Robustness Benchmark (Zheng et al., 2021). Evasion: The attack only happens at test time, i.e., $\mathcal{G}_{\text{test}}$, rather than attacking $\mathcal{G}_{\text{train}}$. Inductive: Test nodes are invisible during training. Black-box: The adversary can not access the architecture or the parameters of the target model.

## 3 POWER AND PITFALLS OF GRAPH INJECTION ATTACK

Based on the setting above, we investigate both the advantages and limitations of GIA by comparing it with GMA. While we find GIA is more harmful than GMA when there is no defense (Theorem 1), we also find pitfalls in GIA that can make it easily defendable (Theorem 2).

### 3.1 POWER OF GRAPH INJECTION ATTACK

Following previous works (Zügner et al., 2018), we use a linearized GNN, i.e., $H^{(k)} = \hat{A}^k X\Theta$, to track the changes brought by attacks. Firstly we will elaborate the threats of an adversary as follows.

**Definition 3.1** (Threats). *Consider an adversary $\mathcal{A}$, given a perturbation budget $\triangle$, the threat of $\mathcal{A}$ to a GNN $f_\theta$ is defined as $\min_{\|\mathcal{G}' - \mathcal{G}\| \leq \triangle} \mathcal{L}_{atk}(f_\theta(\mathcal{G}'))$, i.e., the optimal objective value of Eq. 2.*

---

[2]We leave more details and reasons about the setting used in this work in Appendix B.

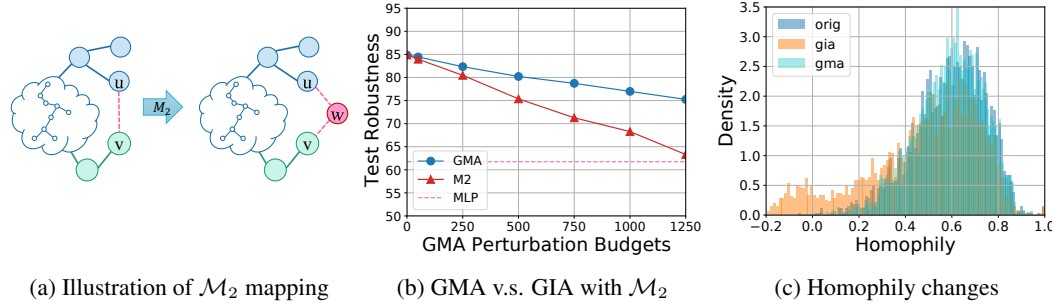

| (a) Illustration of $\mathcal{M}_2$ mapping | (b) GMA v.s. GIA with $\mathcal{M}_2$ | (c) Homophily changes |

Figure 2: Power and pitfalls of Graph Injection Attack

With Definition 3.1, we can quantitatively compare the threats of different adversaries.

**Theorem 1.** *Given moderate perturbation budgets $\triangle_{GIA}$ for GIA and $\triangle_{GMA}$ for GMA, that is, let $\triangle_{GIA} \leq \triangle_{GMA} \ll |V| \leq |E|$, for a fixed linearized GNN $f_\theta$ trained on $\mathcal{G}$, assume that $\mathcal{G}$ has no isolated nodes, and both GIA and GMA follow the optimal strategy, then, $\forall \triangle_{GMA} \geq 0, \exists \triangle_{GIA} \leq \triangle_{GMA}$,*

$$\mathcal{L}_{atk}(f_\theta(\mathcal{G}'_{GIA})) - \mathcal{L}_{atk}(f_\theta(\mathcal{G}'_{GMA})) \leq 0,$$

*where $\mathcal{G}'_{GIA}$ and $\mathcal{G}'_{GMA}$ are the perturbed graphs generated by GIA and GMA, respectively.*

We prove Theorem 1 in Appendix E.1. Theorem 1 implies that GIA can cause more damage than GMA with equal or fewer budgets, which is also verified empirically as shown in Fig. 1a.

Intuitively, the power of GIA mainly comes from its relatively high flexibility in perturbation generation. Such flexibility enables us to find a mapping that can map any GMA perturbations to GIA perturbations, leading the same influences to the predictions of $f_\theta$. We will give an example below.

**Definition 3.2** (Plural Mapping $\mathcal{M}_2$). *$\mathcal{M}_2$ maps a perturbed graph $\mathcal{G}'_{GMA}$ generated by GMA with only edge addition perturbations,[3] to a GIA perturbed graph $\mathcal{G}'_{GIA} = \mathcal{M}_2(\mathcal{G}'_{GMA})$, such that:*

$$f_\theta(\mathcal{G}'_{GIA})_u = f_\theta(\mathcal{G}'_{GMA})_u, \forall u \in V.$$

As illustrated in Fig. 2a, the procedure of $\mathcal{M}_2$ is, for each edge $(u,v)$ added by GMA to attack node $u$, $\mathcal{M}_2$ can inject a new node $w$ to connect $u$ and $v$, and change $X_w$ to make the same effects to the prediction on $u$. Then GIA can be further optimized to bring more damage to node $u$. We also empirically verify the above procedure in Fig. 2b. Details about the comparison are in Appendix C.

### 3.2 PITFALLS IN GRAPH INJECTION ATTACK

Through $\mathcal{M}_2$, we show that the flexibility in GIA can make it more harmful than GMA when there is no defense, however, we also find a side-effect raised in the optimization trajectory of $X_w$ from the above example. Assume GIA uses PGD (Madry et al., 2018) to optimize $X_w$ iteratively, we find:

$$\text{sim}(X_u, X_w)^{(t+1)} \leq \text{sim}(X_u, X_w)^{(t)}, \tag{5}$$

where $t$ is the number of optimization steps and $\text{sim}(X_u, X_v) = \frac{X_u \cdot X_v}{\|X_u\|_2 \|X_v\|_2}$. We prove the statement in Appendix E.4. It implies that, under the mapping $\mathcal{M}_2$, the similarity between injected nodes and targets continues to decrease as the optimization processes, and finally becomes lower than that in GMA. We find this is closely related to the loss of *homophily* of the target nodes.

Before that, we will elaborate the definition of homophily in graph adversarial setting. Different from typical definitions that rely on the label information (McPherson et al., 2001; London & Getoor, 2014; Pei et al., 2020; Zhu et al., 2020), as the adversary does not have the access to all labels, we provide another instantiation of homophily based on node feature similarity as follows:

**Definition 3.3** (Node-Centric Homophily). *The homophily of a node $u$ can be defined with the similarity between the features of node $u$ and the aggregated features of its neighbors:*

$$h_u = sim(r_u, X_u), \ r_u = \sum_{j \in \mathcal{N}(u)} \frac{1}{\sqrt{d_j}\sqrt{d_u}} X_j, \tag{6}$$

*where $d_u$ is the degree of node $u$ and $sim(\cdot)$ is a similarity metric, e.g., cosine similarity.*

---

[3]We focus on edge addition in later discussions since Wu et al. (2019) observed that it produces the most harm in GMA. Discussions about the other GMA operations can be found in Appendix E.2.

We also define edge-centric homophily while we will focus primarily on node-centric homophily. Details and reasons are in Appendix D.1. With Definition 3.3, combining Eq. 5, we have:

$$h_u^{\text{GIA}} \leq h_u^{\text{GMA}},$$

where $h_u^{\text{GIA}}$ and $h_u^{\text{GMA}}$ denote the homophily of node $u$ after GIA and GMA attack, respectively. It implies that GIA will cause more damage to the homophily of the original graph than GMA. To verify the discovery for more complex cases, we plot the homophily distributions in Fig. 2c. The blue part denotes the original homophily distribution. Notably, there is an outstanding out-of-distribution (orange) part caused by GIA, compared to the relatively minor (canny) changes caused by GMA. The same phenomenon also appears in other datasets that can be found in Appendix D.2.

Having observed the huge homophily damage led by GIA, it is straightforward to devise a defense mechanism aiming to recover the original homophily, which we call *homophily defenders*. We theoretically elaborate such defenses in the form of edge pruning[4], adapted from Eq. 1:

$$H_u^{(k)} = \sigma(W_k \cdot \rho(\{\mathbb{1}_{\text{con}}(u, v) \cdot H_v^{(k-1)}\}| \, v \in \mathcal{N}(u) \cup \{u\}). \tag{7}$$

We find that simply pruning the malicious edges identified by a proper condition can empower homophily defenders with strong theoretical robustness against GIA attacks.

**Theorem 2.** *Given conditions in Theorem 1, consider a GIA attack, which* (i) *is mapped by $\mathcal{M}_2$ (Def. 3.2) from a GMA attack that only performs edge addition perturbations, and* (ii) *uses a linearized GNN trained with at least one node from each class in $\mathcal{G}$ as the surrogate model, and* (iii) *optimizes the malicious node features with PGD. Assume that $\mathcal{G}$ has no isolated node, and has node features as $X_u = \frac{C}{C-1} e_{Y_u} - \frac{1}{C-1} \mathbf{1} \in \mathbb{R}^d$, where $Y_u$ is the label of node $u$ and $e_{Y_u} \in \mathbb{R}^d$ is a one-hot vector with the $Y_u$-th entry being $1$ and others being $0$. Let the minimum similarity for any pair of nodes connected in $\mathcal{G}$ be $s_{\mathcal{G}} = \min_{(u,v) \in E} sim(X_u, X_v)$ with $sim(X_u, X_v) = \frac{X_u \cdot X_v}{\|X_u\|_2 \|X_v\|_2}$. For a homophily defender $g_\theta$ that prunes edges $(u, v)$ if $sim(X_u, X_v) \leq s_{\mathcal{G}}$, we have:*

$$\mathcal{L}_{atk}(g_\theta(\mathcal{M}_2(\mathcal{G}'_{GMA}))) - \mathcal{L}_{atk}(g_\theta(\mathcal{G}'_{GMA})) \geq 0.$$

We prove Theorem 2 in Appendix E.3. It implies that, by specifying a mild pruning condition, the homophily defender can effectively reduce the harm caused by GIA to be lower than that of GMA.

Considering a more concrete example with $\mathcal{M}_2$, $X_w$ is generated to make $\mathcal{L}_{\text{atk}}(f_\theta(\mathcal{M}_2(\mathcal{G}'_{\text{GMA}}))) = \mathcal{L}_{\text{atk}}(f_\theta(\mathcal{G}'_{\text{GMA}}))$ on node $u$ at first. Then, due to the flexibility in GIA, $X_w$ can be optimized to some $X'_w$ that greatly destroys the homophily of node $u$, i.e., having a negative cosine similarity score with $u$. Thus, for a graph with relatively high homophily, i.e., $s_{\mathcal{G}} \geq 0$, a mild pruning condition such as $\mathbb{1}_{\text{sim}(u,v) \leq 0}(u, v) = 0$ could prune all the malicious edges generated by GIA while possibly keeping some of those generated by GMA, which makes GIA less threatful than GMA.

In the literature, we find that GNNGuard (Zhang & Zitnik, 2020) serves well for an implementation of homophily defenders as Eq. 7. With GNNGuard, we verify the strong empirical robustness of homophily defenders against GIA. As Fig. 1b depicts, when with homophily defenders, GIA can only cause little-to-no damage, while GMA can still effectively perturb the predictions of the target model on some nodes. To fully demonstrate the power of homophily defenders, we also prove its certified robustness for a concrete GIA case in Appendix E.6.

## 4    HOMOPHILY UNNOTICEABLE GRAPH INJECTION ATTACK

### 4.1    HARMONIOUS ADVERSARIAL OBJECTIVE

As shown in Sec. 3, the flexibility of GIA makes it powerful while dramatically hinders its performance when combating against homophily defenders, because of the great damage to the homophily distribution brought by GIA. This observation motivates us to introduce the concept of *homophily unnoticeability* that enforces GIA to preserve the original homophily distribution during the attack.

---

[4]Actually, homophily defenders can have many implementations other than pruning edges as given in Appendix F, while we will focus on the design above in our discussion.

**Definition 4.1** (Homophily Unnoticeability). *Let the node-centric homophily distribution for a graph $\mathcal{G}$ be $\mathcal{H}_{\mathcal{G}}$. Given the upper bound for the allowed homophily distribution shift $\triangle_{\mathcal{H}} \geq 0$, an attack $\mathcal{A}$ is homophily unnoticeable if:*

$$m(\mathcal{H}_{\mathcal{G}}, \mathcal{H}_{\mathcal{G}'}) \leq \triangle_{\mathcal{H}},$$

*where $\mathcal{G}'$ is the perturbed graph generated by $\mathcal{A}$, and $m(\cdot)$ is a distribution distance measure.*

Intuitively, homophily unnoticeability can be a supplementary for the unnoticeability in graph adversarial attack that requires a GIA adversary to consider how likely the new connections between the malicious nodes and target nodes will appear *naturally*. Otherwise, i.e., unnoticeability is broken, the malicious nodes and edges can be easily detected and removed by database administrators or homophily defenders. However, homophily unnoticeability can not be trivially implemented as a rigid constraint and be inspected incrementally like that for degree distribution (Zügner et al., 2018). For example, a trivial implementation such as clipping all connections that do not satisfy the constraint (Def. 4.1) will trivially clip all the injected edges due to the unconstrained optimization in GIA.

Considering the strong robustness of homophily defenders, we argue that they can directly serve as external examiners for homophily unnoticeability check. Satisfying the homophily constraint can be approximately seen as bypassing the homophily defenders. Obviously, GIA with constraints as Eq. 14 can not guarantee homophily unnoticeability, since it will only optimize towards maximizing the damage by minimizing the homophily of the target nodes. Hence, we propose a novel realization of the homophily constraint for GIA that enforces it to meet the homophily unnoticeability *softly*.

**Definition 4.2** (Harmonious Adversarial Objective (HAO)). *Observing the homophily definition in Eq. 6 is differentiable with respect to $X$, we can integrate it into the objective of Eq. 2 as:*[5]

$$\min_{\|\mathcal{G}'-\mathcal{G}\|\leq\triangle} \mathcal{L}_{atk}^h(f_{\theta^*}(\mathcal{G}')) = \mathcal{L}_{atk}(f_{\theta^*}(\mathcal{G}')) - \lambda C(\mathcal{G}, \mathcal{G}'), \tag{8}$$

*where $C(\mathcal{G}, \mathcal{G}')$ is a regularization term based on homophily and $\lambda \geq 0$ is the corresponding weight.*

One possible implementation is to maximize the homophily for each injected node as:

$$C(\mathcal{G}, \mathcal{G}') = \frac{1}{|V_{\text{atk}}|} \sum_{u \in V_{\text{atk}}} h_u. \tag{9}$$

HAO seizes the possibility of retaining homophily unnoticeability, while still performing effective attacks. Hence, given the homophily distribution distance measure $m(\cdot)$ in Def. 4.1, we can infer:

**Theorem 3.** *Given conditions in Theorem 2, we have $m(\mathcal{H}_{\mathcal{G}}, \mathcal{H}_{\mathcal{G}'_{HAO}}) \leq m(\mathcal{H}_{\mathcal{G}}, \mathcal{H}_{\mathcal{G}'_{GIA}})$, hence:*

$$\mathcal{L}_{atk}(g_{\theta}(\mathcal{G}'_{HAO})) - \mathcal{L}_{atk}(g_{\theta}(\mathcal{G}'_{GIA})) \leq 0,$$

*where $\mathcal{G}'_{HAO}$ is generated by GIA with HAO, and $\mathcal{G}'_{GIA}$ is generated by GIA without HAO.*

We prove Theorem 3 in Appendix E.5. Intuitively, since GIA with HAO can reduce the damage to homophily, it is more likely to bypass the homophily defenders, thus being more threatful than GIA without HAO. We also empirically verify Theorem 3 for more complex cases in the experiments.

## 4.2 ADAPTIVE INJECTION STRATEGIES

GIA is generically composed of two procedures, i.e., node injection and feature update, to solve for $\mathcal{G}' = (A', X')$, where node injection leverages either the gradient information or heuristics to solve for $A'$, and feature update usually uses PGD (Madry et al., 2018) to solve for $X'$. Most previous works separately optimize $A'$ and $X'$ in a greedy manner, which implicitly assumes that the other will be optimized to maximize the harm. However, HAO does not follow the assumption but stops the optimization when the homophily is overly broken. Thus, a more suitable injection strategy for HAO shall *be aware of* retaining the original homophily. To this end, we propose to optimize $A'$ and $X'$ alternatively and introduce three adaptive injection strategies to coordinate with HAO.

---

[5]Note that we only use HAO to solve for $\mathcal{G}'$ while still using the original objective to evaluate the threats.

**Gradient-Driven Injection.** We propose a novel bi-level formulation of HAO to perform the alternative optimization using gradients, where we separate the optimization of $\mathcal{G}' = (A', X')$ as:

$$
\begin{aligned}
X'^* &= \underset{X' \in \Phi(X')}{\arg\min} \; \mathcal{L}_{\text{atk}}(f_{\theta*}(A'^*, X')) - \lambda_A C(\mathcal{G}', \mathcal{G}), \\
s.t. \; A'^* &= \underset{A' \in \Phi(A')}{\arg\min} \; \mathcal{L}_{\text{atk}}(f_{\theta*}(A', X')) - \lambda_X C(\mathcal{G}', \mathcal{G}),
\end{aligned}
\tag{10}
$$

where $\Phi(A')$ and $\Phi(X')$ are the corresponding feasible regions for $A'$ and $X'$ induced by the original constraints. Here we use different homophily constraint weights $\lambda_A$ and $\lambda_X$ for the optimizations of $A'$ and $X'$, since $A'$ is discrete while $X'$ is continuous. We can either adopt Meta-gradients like Metattack (Zügner & Günnemann, 2019) (**MetaGIA**) or directly optimize edge weights to solve for $A'$ (**AGIA**). The detailed induction of meta-gradients and algorithms are given in Appendix G.1.

**Heuristic-Driven Injection.** As the state-of-the-art GIA methods are leveraging heuristics to find $A'$, based on TDGIA (Zou et al., 2021), we also propose a variant (**ATDGIA**) using heuristics as:

$$
s_u = ((1 - p_u)\mathbb{1}(\arg\max(p) = y'_u))(\frac{0.9}{\sqrt{bd_u}} + \frac{0.1}{d_u}),
\tag{11}
$$

where $s_u$ indicates the vulnerability of node $u$ and $\mathbb{1}(\cdot)$ is to early stop destroying homophily.

**Sequential Injection** for large graphs. Since gradient methods require huge computation overhead, we propose a novel divide-and-conquer strategy (**SeqGIA**) to iteratively select some of the most vulnerable targets with Eq. 11 to attack. Detailed algorithm is given in Appendix G.3.

## 5 EXPERIMENTS

### 5.1 SETUP & BASELINES

**Datasets.** We comprehensively evaluate our methods with 38 defense models on 6 datasets. We select two classic citation networks Cora and Citeseer (Yang et al., 2016; Giles et al., 1998) refined by GRB (Zheng et al., 2021). We also use Aminer and Reddit (Tang et al., 2008; Hamilton et al., 2017b; Zeng et al., 2020) from GRB, Arxiv from OGB (Hu et al., 2020), and a co-purchasing network Computers (McAuley et al., 2015) to cover more domains and scales. Details are in Appendix H.1.

**Comparing with previous attack methods.** We incorporate HAO into several existing GIA methods as well as our proposed injection strategies to verify its effectiveness and versatility. First of all, we select **PGD** (Madry et al., 2018) as it is one of the most widely used adversarial attacks. We also select **TDGIA** (Zou et al., 2021) which is the state-of-the-art GIA method. We adopt the implementations in GRB (Zheng et al., 2021) for the above two methods. We exclude FGSM (Goodfellow et al., 2015) and AFGSM (Wang et al., 2020), since PGD is better at dealing with non-linear models than FGSM (Madry et al., 2018), and AFGSM performs comparably with FGSM but is worse than TDGIA as demonstrated by Zou et al. (2021). For GMA methods, we adopt **Metattack** (Zügner & Günnemann, 2019) as one of the bi-level implementations. We exclude Nettack (Zügner et al., 2018) as it is hard to perform incremental updates with GCN (the surrogate model used in our experiments) and leave reinforcement learning methods such as RL-S2V (Dai et al., 2018) and NIPA (Sun et al., 2020) for future work. More details are given in Appendix H.2.

**Categories and complexity analysis of attack methods.** We provide categories and complexity analysis of all attack methods used in our experiments in Table 7, Appendix H.3.

**Competing with different defenses.** We select both popular GNNs and robust GNNs as the defense models. For popular GNNs, we select the three most frequently used baselines, i.e., **GCN** (Kipf & Welling, 2017), **GraphSage** (Hamilton et al., 2017b), and **GAT** (Veličković et al., 2018). For robust GNNs, we select **GCNGuard** (Zhang & Zitnik, 2020) for graph purification approach, and **RobustGCN** (Zhu et al., 2019) for stabilizing hidden representation approach, as representative ones following the surveys (Sun et al., 2018; Jin et al., 2021). Notably, the author-released GCNGuard implementation requires $O(n^2)$ complexity, which is hard to scale up. To make the comparison fair, following the principle of homophily defenders, we implement two efficient robust alternatives, i.e., Efficient GCNGuard (**EGuard**) and Robust Graph Attention Network (**RGAT**). More details are given in Appendix F.2. Besides, we exclude the robust GNNs learning in a transductive manner like ProGNN (Jin et al., 2020) that can not be adapted in our setting.

Table 1: Performance of non-targeted attacks against different models

| | HAO | Cora (↓) | | | Citeseer(↓) | | | Computers(↓) | | | Arxiv(↓) | | |
|---|---|---|---|---|---|---|---|---|---|---|---|---|---|
| | | Homo | Robust | Combo | Homo | Robust | Combo | Homo | Robust | Combo | Homo | Robust | Combo |
| Clean | | 85.74 | 86.00 | 87.29 | 74.85 | 75.46 | 75.87 | 93.17 | 93.17 | 93.32 | 70.77 | 71.27 | 71.40 |
| PGD | | 83.08 | 83.08 | 85.74 | 74.70 | 74.70 | 75.19 | 84.91 | 84.91 | 91.41 | 68.18 | 68.18 | 71.11 |
| PGD | ✓ | 52.60 | 62.60 | 77.99 | 69.05 | 69.05 | 73.04 | 79.33 | 79.33 | 87.83 | 55.38 | 62.89 | 68.68 |
| MetaGIA[†] | | 83.61 | 83.61 | 85.86 | 74.70 | 74.70 | 75.15 | 84.91 | 84.91 | 91.41 | 68.47 | 68.47 | 71.09 |
| MetaGIA[†] | ✓ | 49.25 | 69.83 | 76.80 | 68.04 | 68.04 | 71.25 | 78.96 | 78.96 | 90.25 | 57.05 | 63.30 | 69.97 |
| AGIA[†] | | 83.44 | 83.44 | 85.78 | 74.72 | 74.72 | 75.29 | 85.21 | 85.21 | 91.40 | 68.07 | 68.07 | 71.01 |
| AGIA[†] | ✓ | 47.24 | 61.59 | 75.25 | 70.24 | 70.24 | 71.80 | 75.14 | 75.14 | 86.02 | 59.32 | 65.62 | 69.92 |
| TDGIA | | 83.44 | 83.44 | 85.72 | 74.76 | 74.76 | 75.26 | 88.32 | 88.32 | 91.40 | 64.49 | 64.49 | 70.97 |
| TDGIA | ✓ | 56.95 | 73.38 | 79.45 | 60.91 | 60.91 | 72.51 | 74.77 | 74.77 | 90.42 | 49.36 | 60.72 | 63.57 |
| ATDGIA | | 83.07 | 83.07 | 85.39 | 74.72 | 74.72 | 75.12 | 86.03 | 86.03 | 91.41 | 66.95 | 66.95 | 71.02 |
| ATDGIA | ✓ | 42.18 | 70.30 | 76.87 | 61.08 | 61.08 | 71.22 | 80.86 | 80.86 | 84.60 | 45.59 | 63.30 | 64.31 |
| MLP | | | 61.75 | | | 65.55 | | | 84.14 | | | 52.49 | |

[↓]The lower number indicates better attack performance. [†]Runs with SeqGIA framework on Computers and Arxiv.

Table 2: Performance of targeted attacks against different models

| | HAO | Computers(↓) | | | Arxiv(↓) | | | Aminer(↓) | | | Reddit(↓) | | |
|---|---|---|---|---|---|---|---|---|---|---|---|---|---|
| | | Homo | Robust | Combo | Homo | Robust | Combo | Homo | Robust | Combo | Homo | Robust | Combo |
| Clean | | 92.68 | 92.68 | 92.83 | 69.41 | 71.59 | 72.09 | 62.78 | 66.71 | 66.97 | 94.05 | 97.15 | 97.13 |
| PGD | | 88.13 | 88.13 | 91.56 | 69.19 | 69.19 | 71.31 | 53.16 | 53.16 | 56.31 | 92.44 | 92.44 | 93.03 |
| PGD | ✓ | 71.78 | 71.78 | 85.81 | 36.06 | 37.22 | 69.38 | 34.62 | 34.62 | 39.47 | 56.44 | 86.12 | 84.94 |
| MetaGIA[†] | | 87.67 | 87.67 | 91.56 | 69.28 | 69.28 | 71.22 | 48.97 | 48.97 | 52.35 | 92.40 | 92.40 | 93.97 |
| MetaGIA[†] | ✓ | 70.21 | 71.61 | 85.83 | 38.44 | 38.44 | 48.06 | 41.12 | 41.12 | 45.16 | 46.75 | 90.06 | 90.78 |
| AGIA[†] | | 87.57 | 87.57 | 91.58 | 66.19 | 66.19 | 70.06 | 50.50 | 50.50 | 53.69 | 91.62 | 91.62 | 93.66 |
| AGIA[†] | ✓ | 69.96 | 71.58 | 85.72 | 38.84 | 38.84 | 68.97 | 35.94 | 35.94 | 42.66 | 80.69 | 88.84 | 90.44 |
| TDGIA | | 87.21 | 87.21 | 91.56 | 63.66 | 63.66 | 71.06 | 51.34 | 51.34 | 54.82 | 92.19 | 92.19 | 93.62 |
| TDGIA | ✓ | 71.39 | 71.62 | 77.15 | 42.56 | 42.56 | 42.53 | 25.78 | 25.78 | 29.94 | 78.16 | 85.06 | 88.66 |
| ATDGIA | | 87.85 | 87.85 | 91.56 | 66.12 | 66.12 | 71.16 | 50.87 | 50.87 | 53.68 | 91.25 | 91.25 | 93.03 |
| ATDGIA | ✓ | 72.00 | 72.53 | 78.35 | 38.28 | 40.81 | 39.47 | 22.50 | 22.50 | 28.91 | 64.09 | 89.06 | 88.91 |
| MLP | | | 84.11 | | | 52.49 | | | 32.80 | | | 70.69 | |

[↓]The lower number indicates better attack performance. [†]Runs with SeqGIA framework.

**Competing with extreme robust defenses.** To make the evaluation for attacks more reliable, we also adopt two widely used robust tricks Layer Normalization (**LN**) (Ba et al., 2016) and an efficient adversarial training (Goodfellow et al., 2015; Madry et al., 2018) method **FLAG** (Kong et al., 2020). Here, as FLAG can effectively enhance the robustness, we exclude other adversarial training methods for efficiency consideration. More details are given in Appendix H.4.

**Evaluation protocol.** We use a 3-layer GCN as the surrogate model to generate perturbed graphs with various GIA attacks, and report the mean accuracy of defenses from multiple runs. Details are in Appendix H.5. For in-detail analysis of attack performance, we categorize all defenses into three folds by their robustness: Vanilla, **Robust**, and Extreme Robust (**Combo**) (Table 8). To examine how much an attack satisfies the homophily unnoticeability and its upper limits, we report *maximum test accuracy* of both homophily defenders (**Homo**) and defenses from the last two categories.

## 5.2 EMPIRICAL PERFORMANCE

In Table 1 and Table 2, we report the non-targeted and targeted attack performance of various GIA methods, respectively. We bold out the best attack and underline the second-best attack when combating against defenses from each category. Full results are in Appendix J.1 and Appendix J.2.

**Performance of non-targeted attacks.** In Table 1, we can see that HAO significantly improves the performance of *all* attacks on *all* datasets up to 30%, which implies the effectiveness and versatility of HAO. Especially, even coupled with a random injection strategy (PGD), HAO can attack robust models to be comparable with or inferior to simple MLP which does not consider relational information. Meanwhile, adaptive injection strategies outperform previous methods PGD and TDGIA by a non-trivial margin for most cases, which further indicates that they are more suitable for HAO.

**Performance of targeted attack on large-scale graphs.** In Table 2, HAO also improves the targeted attack performance of *all* attack methods on *all* datasets by a significant margin of up to 15%, which implies that the benefits of incorporating HAO are universal. Besides, adaptive injections can further improve the performance of attacks and establish the new state-of-the-art coupled with HAO.

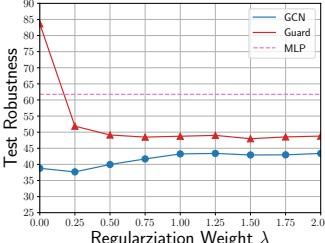

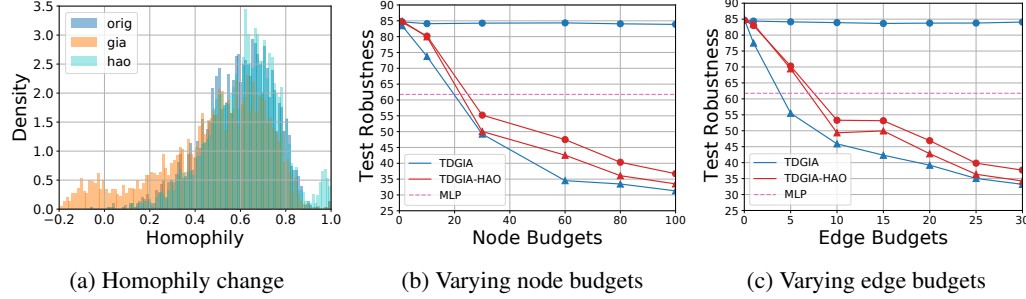

| Model | Cora[†] | Computers[†] | Arxiv[†] | Computers[‡] | Aminer[‡] | Reddit[‡] |
|---|---|---|---|---|---|---|
| Clean | 84.74 | 92.25 | 70.44 | 91.68 | 62.39 | 95.51 |
| PGD | 61.09 | 61.75 | 54.23 | 62.41 | 26.13 | 62.72 |
| +HAO | 56.63 | 59.16 | 45.05 | 59.09 | 22.15 | 56.99 |
| MetaGIA | 60.56 | 61.75 | 53.69 | 62.08 | 32.78 | 60.14 |
| +HAO | 58.51 | 60.29 | 48.48 | 58.63 | 29.91 | 54.14 |
| AGIA | 60.10 | 60.66 | 48.86 | 61.98 | 31.06 | 59.96 |
| +HAO | 53.79 | 58.71 | 48.86 | 58.37 | 26.51 | 56.36 |
| TDGIA | 66.86 | 66.79 | 49.73 | 62.47 | 32.37 | 57.97 |
| +HAO | 65.22 | 65.46 | 49.54 | 59.67 | 22.32 | 54.32 |
| ATDGIA | 61.14 | 65.07 | 46.53 | 64.66 | 24.72 | 61.25 |
| +HAO | 58.13 | 63.31 | 44.40 | 59.27 | 17.62 | 56.90 |

The lower is better. [†]Non-targeted attack. [‡]Targeted attack.

(a) Varying $\lambda$ in HAO        (b) Averaged performance across all defense models

Figure 3: (a): Effects of HAO with different weights; (b) Averaged attack performance of various attacks with or without HAO against both homophily defenders and other defense models.

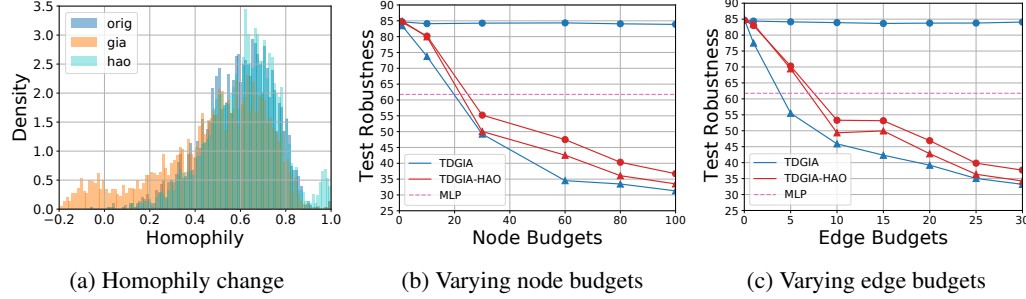

(a) Homophily change      (b) Varying node budgets      (c) Varying edge budgets

Figure 4: (a) Homophily changes after attacked by GIA without HAO (orange) and GIA with HAO (canny); (b), (c) Attack performance against GCN and EGuard with different node and edge budgets. ● indicates attack with defenses and ▲ indicates attack without defenses;

### 5.3 ANALYSIS AND DISCUSSIONS

**Effects of HAO.** Though HAO can substantially improve GIA methods under defenses, we find it essentially trades with the performance under no defenses. In Fig. 3a, as the weight for regularization term $\lambda$ increases, HAO trades slightly more of the performance against GCN for the performance against homophily defenders. Finally, GIA reduces the performance of both GNNs with defenses and without defenses to be inferior to MLP. Additionally, as also shown in Table 3b, the trade-off will not hurt the overall performance while consistently brings benefits up to 5%.

**Analysis of the perturbed graphs.** In Fig. 4a, we also analyze the homophily distribution changes after the attack. It turns out that GIA with HAO can effectively preserve the homophily while still conducting effective attacks. Similar analysis on other datasets can be found in Appendix D.2.

**Attacks with limited budgets.** We also examine the performance of GIA methods with or without HAO varying different node and edge budgets. Fig. 4b and Fig. 4c show that HAO can consistently improve the overall performance by slightly trading with the performance under no defenses.

### 6 CONCLUSIONS

In this paper, we explored the advantages and limitations of GIA by comparing it with GMA. Though we found that GIA can be provably more harmful than GMA, the severe damage to the homophily makes it easily defendable by homophily defenders. Hence we introduced homophily unnoticeability for GIA to constrain the damage and proposed HAO to instantiate it. Extensive experiments show that GIA with HAO can break homophily defenders and substantially outperform all previous works. We provide more discussions and future directions based on HAO in Appendix A.

## ACKNOWLEDGEMENTS

We thank the area chair and reviewers for their valuable comments. This work was partially supported by GRF 14208318 and ECS 22200720 from the RGC of HKSAR, YSF 62006202 from NSFC, Australian Research Council Projects DE-190101473 and DP-220102121.

## ETHICS STATEMENT

Considering the wide applications and high vulnerability to adversarial attacks of GNNs, it is important to develop trustworthy GNNs that are robust to adversarial attacks, especially for safety-critical applications such as financial systems. However, developing trustworthy GNNs heavily rely on the effective evaluation of the robustness of GNNs, i.e., adversarial attacks that are used to measure the robustness of GNNs. Unreliable evaluation will incur severe more issues such as overconfidence in the adopted GNNs and further improve the risks of adopting unreliable GNNs. Our work tackles an important issue in current GIA, hence enhancing the effectiveness of current evaluation on the robustness of GNNs, and further empowering the community to develop more trustworthy GNNs to benefit society. Besides, this paper does not raise any ethical concerns. This study does not involve any human subjects, practices to data set releases, potentially harmful insights, methodologies and applications, potential conflicts of interest and sponsorship, discrimination/bias/fairness concerns, privacy and security issues, legal compliance, and research integrity issues.

## REPRODUCIBILITY STATEMENT

To ensure the reproducibility of our theoretical, we provide detailed proof for our theoretical discovery in Appendix E. Specifically, we provide detailed proof in Appendix E.1 for Theorem 1, proof in Appendix E.3 for Theorem 2, and proof in Appendix E.5 for Theorem 3.

To ensure the reproducibility of our experimental results, we detail our experimental setting in Appendix B.2 and Appendix H, and open source our code.

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

# A  ADDITIONAL DISCUSSIONS AND FUTURE DIRECTIONS

## A.1  DISCUSSIONS ON HAO AND ITS LIMITATIONS

**Discussions on HAO and future implications.** It is widely received that it is difficult to give a proper definition of unnoticeability for graphs (More details are also given in Appendix B.2.2). Based on earliest unnoticeability constraints on degree distribution changes (Zügner et al., 2018; Zügner & Günnemann, 2019), empirical observations that graph adversarial attacks may change some feature statistics and connect dissimilar neighbors are identified, and leveraged as heuristics to develop robust GNNs (Wu et al., 2019; Entezari et al., 2020; Zhang & Zitnik, 2020; Jin et al., 2020). Though empirically effective, however, few of them provide theoretical explanations or relate this phenomenon to unnoticeability. In this work, starting from the comparison of GMA and GIA, we identified GIA would lead to severe damage to the original homophily. Furthermore, the relatively high flexibility of GIA amplifies the destruction and finally results in the break of homophily unnoticeability. The main reason for this phenomenon is mainly because of the poorly defined unnoticeability in graph adversarial attack. Without a proper definition, the developed attacks tend to the shortcut to incur damage instead of capturing the true underlying vulnerability of GNNs. Consequently, using these attacks to evaluate robustness of GNNs will bring unreliable results thus hindering the development of trustworthy GNNs.

To be more specific, due to the poor unnoticeability constraint for graph adversarial learning, the developed attacks tend to leverage the shortcuts to greatly destroy the original homophily, which leads to the break of unnoticeability. Thus, using homophily defenders can easily defend these seemingly powerful attacks, even with a simple design, which however brings us unreliable conclusions about the robustness of homophily defenders. Essentially, HAO penalizes GIA attacks that take the shortcuts, and retain their unnoticeability in terms of homophily. Thus, HAO mitigates the shortcut issue of GIA attacks, urges the attacks to capture the underlying vulnerability of GNNs and brings us a more reliable evaluation result, from which we know simple homophily defenders are essentially not robust GNNs.

In addition, the proposed realization of unnoticeability check for adversarial attacks provides another approach to instantiate the unnoticeability. Especially for the domains that we can hardly leverage inductive bias from human, we can still try to identify their homophily, or the underlying rationales/causality of the data generation process, e.g., grammar correctness, fluency and semantics for natural languages, to instantiate the unnoticeability constraint with the help of external examiners. Since people are likely to be more sensitive to quantitative numbers like accuracy, those external examiners can be conveniently leveraged to the corresponding benchmark or leaderboards to further benefit the community.

**Limitations of HAO and future implications.** Since HAO are mostly developed to preserve the homophily unnoticeability, it counters the greedy strategy of attacks without HAO that destroys the homophily to incur more damage. Therefore, it will inevitably reduce the damage of the attacks without HAO against vanilla GNNs. As observed from the experiments, we find HAO essentially trades the attack performance when against vanilla GNNs for the performance when against homophily defenders. As Fig. 3a shows, the trade-off effects can be further amplified with a large coefficient lambda in HAO. As also shown by Fig. 4b and Fig. 4c, when against vanilla GNNs, compared with GIA without HAO, GIA with HAO show fewer threats. In certain cases, the trade-off might generate the performance of attacks. Thus, it calls for more tailored optimization methods to solve for better injection matrix and node features in the future. Moreover, the trade-off effects

also reflect the importance of homophily to the performance of node classifications and the utility of homophily unnoticeability, where we believe future theoretical works can further study this phenomenon and reveal the underlying causality for node classification or even more other downstream tasks. Thus, we can develop more robust and trustworthy neural graph models that do not depend on spurious correlations to perform the task.

In addition, as homophilous graph is the most common class of graph benchmarks for node classification (Yang et al., 2016; Giles et al., 1998; Hu et al., 2020; Zheng et al., 2021), our discussions are mostly focused on this specific class of graphs. However, when applying HAO to other classes of graphs such as non-attributed graphs, a direct adaption of HAO may not work. Nevertheless, if the underlying information for making correct predictions still resemble the homophily property, for example, in a non-attributed graph, nodes with similar structures tend to have similar labels, it is still promising to introduce the node features with node embeddings, derive a new definition of homophily and apply HAO. Moreover, recently disassortative graphs appear to be interesting to the community (Pei et al., 2020; Zhu et al., 2020), which exhibit heterophily property that neighbors tend to have dissimilar labels, in contrast to homophily. We conduct an investigation on this specific class of graphs and detailed results are given in Table 12, from which we surprisingly find HAO still maintains the advance when incorporating various GIA attacks. The reason might be that GNNs and GIA with HAO can still implicitly learn the homophily such as similarity between class label distributions (Ma et al., 2022), even without explicit definitions. To summarize, we believe future extension of HAO to other classes of graphs is another interesting direction.

Besides, the discussions in this paper are only considered the relationship between adversarial robustness and homophily. However, label noises are another widely existing threats that are deserved to be paid attention to (Liu & Tao, 2016; Han et al., 2018; 2020a;b). Essentially, our discussions in Appendix B.3 are also closely related to the vulnerability of GNNs to label noises, where GNNs can still achieve near perfect fitting to the datasets with full label noises. Thus, it is desirable to broader the attention and discussion to include the label noises when developing trustworthy GNNs.

## A.2 MORE FUTURE DIRECTIONS

Besides the future implications inspired from the limitations of HAO, we believe there are also many promising future works that could be built upon HAO.

**Rethinking the definition of unnoticeability in adversarial robustness.** Though the study of adversarial robustness was initially developed around the deep learning models on image classification (Szegedy et al., 2014; Goodfellow et al., 2015; Madry et al., 2018), images and classification are far from the only data and the task we want to build neural networks for. Deep learning models are widely applied to other types of data, such as natural languages and graphs, where human inductive bias can hardly be leveraged to elaborate a proper definition of unnoticeability. Moreover, for more complicated tasks involving implicit reasoning, even in the domain of images, the original definition of unnoticeability, i.e., L-p norm, may not be sufficient to secure all shortcuts that can be leveraged by adversaries. How to establish and justify a proper definition of unnoticeability in these domains and tasks, is critical for developing trustworthy deep learning models.

**Applications to other downstream tasks.** Given the wide applications of GNNs, we believe the studies on the robustness of GNNs should be extended to other downstream tasks, such as link predictions and graph clustering. Specifically, when with a different task objective, it is interesting to find whether the underlying task still depend on the homophily property and how the different optimization objectives affect the attack optimization trajectory.

**Attack with small budgets.** In real-life scenarios, the budgets of the adversary may be limited to a small number. It is interesting to study how to maximize the damage given limited budgets and its interplay between homophily. For example, how to locate the most vulnerable targets. We show an initial example through ATDGIA.

**Mix-up attack of GMA and GIA.** In real-life scenarios, both GMA and GIA could happen while with different budget limits. It is interesting to see whether and how they could be combined to conduct more powerful attacks.

**Injection for defense.** Actually, not only attackers can inject extra nodes, defenders can also inject some nodes to promote the robustness of the networks. For example, according the Proposition. E.1,

nodes with higher degrees, higher MLP decision margin and higher homophily tend more unlikely to be attacked. Hence, defenders may directly inject some nodes the promote the above properties of vulnerable nodes.

**Attacks on more complicated and deep GNNs.** Most existing graph adversarial works focus on analyzing linearized GNNs and apply the discoveries to more complex cases. However, with the development of deep learning and GNNs, some models with complicated structures fail to fit those theories. For example, methods developed by studying linearized GNNs can hardly adapt to GNNs with normalizations as also revealed from our experiments. Then they can even more hardly be adapted to more complex models such as Transformers. On the other hand, most graph adversarial studies only focus on relatively shallow GNNs. Different from other deep learning models, as GNNs go deep, besides more parameters, they also require an exponentially growing number of neighbors as inputs. How the number of layers would affect their robustness and the threats of attacks remain unexplored. From both theoretical and empirical perspectives, we believe it is very interesting to study the interplay between the number of GNN layers and homophily, in terms of adversarial robustness and threats, and how to leverage the discoveries to probe the weakness of complicated models.

**Reinforcement Learning based GIA.** Reinforcement learning based approaches are shown to exhibit promising performances in previous mixed settings (Dai et al., 2018; Sun et al., 2020). Though we exclude them for the efforts needed to adapt them to our setting, we believe it is promising and interesting to incorporate reinforcement learning to develop more tailored injection strategies and vulnerable nodes selection. Meanwhile, it is also interesting to explore how to leverage the idea of SeqGIA proposed in Sec. 4 to reduce the computation overhead of reinforcement learning approaches and enhance their scalability.

## B    MORE DETAILS AND REASONS ABOUT THE GRAPH ADVERSARIAL ATTACK SETTING

We provide more details about the perturbation constraints and the threat model used in Sec. 2.2.

### B.1    PERTURBATION CONSTRAINTS

Following previous works (Zügner et al., 2018; Zou et al., 2021), Graph adversarial attacks can be characterized into graph modification attacks and graph injection attacks by their perturbation constraints. Moreover, we adopt standardization methods (i.e., arctan transformation) following Graph Robustness Benchmark (Zheng et al., 2021) on input features $X$.

**Graph Modification Attack (GMA).** GMA generates $\mathcal{G}'$ by modifying the graph structure $A$ and the node features $X$ of the original graph $\mathcal{G}$. The most widely adopted constraints in GMA is to limit the number of perturbations on $A$ and $X$, denoted by $\triangle_A$ and $\triangle_X$, respectively, as:

$$\triangle_A + \triangle_X \leq \triangle \in \mathbb{Z}, \|A' - A\|_0 \leq \triangle_A \in \mathbb{Z}, \|X' - X\|_\infty \leq \epsilon \in \mathbb{R}, \tag{12}$$

where the perturbation on $X$ is bounded by $\epsilon$ via L-p norm, since we are using continuous features.

**Graph Injection Attack (GIA).** Differently, GIA generates $\mathcal{G}'$ by injecting a set of malicious nodes $V_{\text{atk}}$ as:

$$X' = \begin{bmatrix} X \\ X_{\text{atk}} \end{bmatrix}, A' = \begin{bmatrix} A & A_{\text{atk}} \\ A_{\text{atk}}^T & O_{\text{atk}} \end{bmatrix}, \tag{13}$$

where $X_{\text{atk}}$ is the features of the injected nodes, $O_{\text{atk}}$ is the adjacency matrix among injected nodes, and $A_{\text{atk}}$ is the adjacency matrix between the injected nodes and the original nodes. Let $d_u$ denote the degree of node $u$, the constraints in GIA are:

$$|V_{\text{atk}}| \leq \triangle \in \mathbb{Z}, \ 1 \leq d_u \leq b \in \mathbb{Z}, X_u \in \mathcal{D}_X \subseteq \mathbb{R}^d, \forall u \in V_{\text{atk}}, \tag{14}$$

where the number and degrees of the injected nodes are limited, $\mathcal{D}_X = \{C \in \mathbb{R}^d, \min(X) \cdot \mathbf{1} \leq C \leq \max(X) \cdot \mathbf{1}\}$ where $\min(X)$ and $\max(X)$ are the minimum and maximum entries in $X$ respectively. In other words, each entry of the injected node features are bounded within the minimal entry and maximal entry of the original node feature matrix, following previous setting (Zou et al., 2021).

## B.2 THREAT MODEL

We adopt a unified setting which is also used by Graph Robustness Benchmark (Zheng et al., 2021), that is evasion, inductive, and black-box. Next we will elaborate more details and reasons for adopting the setting.

### B.2.1 DETAILS OF THE THREAT MODEL

**Evasion.** The attack only happens at test time, which means that defenders are able to obtain the original clean graph $\mathcal{G}_{\text{train}}$ for training, while testing on a perturbed graph $\mathcal{G}'$. The reasons for adopting the evasion setting is as shown in Appendix B.2.2.

**Inductive.** The training and testing of GNNs is performed in an inductive manner. Specifically, $f_\theta$ is trained on the (sub) training graph $\mathcal{G}_{\text{train}}$, which is consist of the training nodes with their labels and the edges among training nodes. While during testing, the model will access the whole graph $\mathcal{G}_{\text{test}} = \mathcal{G}$ for inferring the labels of test nodes. In particular, $\mathcal{G}$ is consist of all of the nodes and the edges, including $\mathcal{G}_{\text{train}}$, the test nodes, the edges among test nodes and the edges between training nodes and the test nodes. In contrast, if the training and testing is performed in a transductive manner, the model can access the whole graph during both training and testing, i.e., $\mathcal{G}_{\text{train}} = \mathcal{G}_{\text{test}} = \mathcal{G}$. Since we adopt the evasion setting where the adversary may modify the $\mathcal{G}_{\text{test}}$ during testing, the GNN has to be learned in an inductive manner. More reasons are as elaborated in Appendix B.2.2.

**Black-box.** The adversary has no information about the target model, but the adversary may obtain the graph and training labels to train a surrogate model for generating perturbed graph $\mathcal{G}'$.

Combining all of the above, conducting effective attacks raises special challenges to adversaries, since defenders can adopt the information extracted from training graph $\mathcal{G}_{\text{train}}$ to learn more robust hidden representations (Zhu et al., 2019), or learn to drop noisy edges (Wu et al., 2019; Zhang & Zitnik, 2020; Jin et al., 2020), or even perform adversarial training (Jin & Zhang, 2021; Feng et al., 2021) which is known as one of the strongest defense mechanisms in the domain of images (Goodfellow et al., 2015; Madry et al., 2018).

### B.2.2 DISCUSSIONS ABOUT THE THREAT MODEL

Different from images where we can adopt the inductive bias from human vision system to use numerical constraints, i.e., L-p norm, to bound the perturbation range (Goodfellow et al., 2015; Madry et al., 2018), we cannot use similar numerical constraints to define the unnoticeability for graphs, as they are *weakly correlated* to the information required for node classification. For example, previous work (Zügner et al., 2018) tries to use node degree distribution changes as the unnoticeability constraints. However, given the same degree distribution, we can shuffle the node features to generate multiple graphs with completely different semantic meanings, which disables the functionality of unnoticeability.

Because of the difficulty to properly define the unnoticeability for graphs, adopting a poisoning setting in graph adversarial attack will enlarge the gap between research and practice. Specifically, poisoning attacks require an appropriate definition of unnoticeability so that the defenders are able to distinguish highly poisoned data from unnoticeable poisoned data and the original data. Otherwise, attackers can always leverage some underlying shortcuts implied by the poorly defined unnoticeability, i.e., homophily in our case, to perform the attacks, since the defenders are blind to these shortcuts. On the other hand, leveraging shortcuts may generate data which is unlikely to appear in real-world applications. For example, in a citation network, medical papers are unlikely to cite or be cited by linguistic papers while the attacks may modify the graphs or inject malicious nodes to make medical papers cite or be cited by lots of linguistic papers, which is apparently impractical. Using these attacks to evaluate the robustness of GNNs may bring unreliable conclusions, i.e., homophily defenders in our case, which will greatly hinders the development of the trustworthy GNNs.

Moreover, under a poor unnoticeability definition, without the presence of the original data, defenders have no idea about to what extent the data is poisoned and whether the original labels remain the correspondence. Furthermore, it is well-known that neural networks have universal approximation power (Hornik et al., 1989), thus can easily overfit the training set (Goodfellow et al., 2016), or even *memorize* the labels appeared during training (Zhang et al., 2017). As a generalization from deep

learning models to graphs, GNNs tend to exhibit similar behaviors, which is shown empirically in our experiments (See Appendix B.3 for details). Thus, even trained on a highly poisoned graph, GNNs may still converge to 100% training accuracy, even though the correspondence between the data and the underlying labels might be totally corrupted. In this case, defenders can hardly distinguish whether the training graph is perturbed hence unlikely to make any effective defenses. Besides, studying the robustness of GNNs trained from such highly poisoned graphs seems to be impractical, since real world trainers are unlikely to use such highly poisoned data to train GNNs.

While in an evasion setting, the defenders are able to use the training graph to tell whether the incoming data is heavily perturbed and make some effective defenses, even simply leveraging some feature statistics (Wu et al., 2019; Jin et al., 2020). Notably, A recent benchmark (Zheng et al., 2021) also has similar positions. Thus, we will focus on the evasion setting in this paper.

Given the evasion setting, GNNs can only perform inductive learning where the test nodes and edges are not visible during training. The reason is that, transductive learning (i.e., the whole graph except test labels is available), requires the training graph and test graph to be the same. However, it can not be satisfied as the adversary will modify the test graph, i.e., changing some nodes or edges during GMA attack, or injecting new malicious nodes during GIA attack. Additionally, inductive learning has many practical scenarios. For example, in an academic network, the graph grows larger and larger day by day as new papers are published and added to the original network. GNN models must be inductive to be applied to such evolving graphs.

### B.3    MEMORIZATION EFFECTS OF GRAPH NEURAL NETWORKS

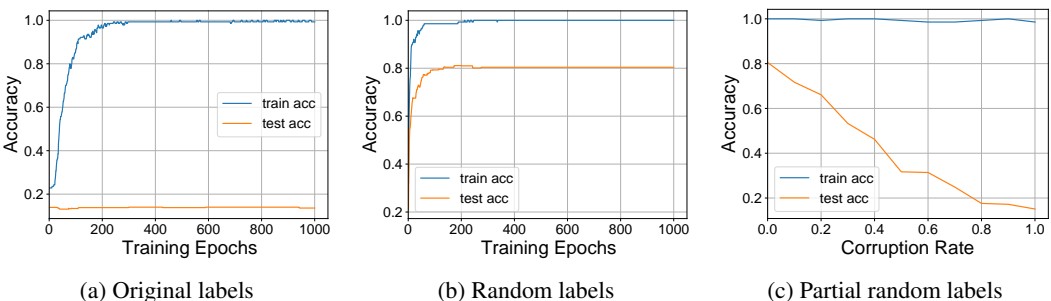

(a) Original labels          (b) Random labels          (c) Partial random labels

Figure 5: Training curve of GCN on Cora with random labels

We conduct experiments with GCN (Kipf & Welling, 2017) on Cora (Yang et al., 2016). The architecture we select is a 2-Layer GCN with 16 hidden units, optimized using Adam (Kingma & Ba, 2015) with a learning rate of 0.01 and a $L_2$ weight decay of $5 \times 10^{-4}$ for the first layer. We train 1000 epochs and report the training accuracy and test accuracy according to the best validation accuracy. We randomly sample certain percent of nodes from the whole graph and reset their labels. It can be seen from Fig. 5 (b) and (c) that even with all random labels, the training accuracy can reach to nearly 100%, which serves as a strong evidence for the existence of memorization effects in GNNs. In other words, even a GNN is trained on a heavily poisoned graph (changes dramatically in the sense of semantic), it can still achieve good training accuracy while the defender has no way to explicitly find it or do anything about it. That is against to the original setting and purpose of adversarial attacks (Szegedy et al., 2014; Goodfellow et al., 2015; Madry et al., 2018). Thus, it urges the community for a proper solution to the ill-defined unnoticeability in current graph adversarial learning. Till the appearance of a silver bullet for unnoticeability on graphs, evasion attack can serve as a better solution than poisoning attack.

## C    MORE DETAILS ABOUT GIA AND GMA COMPARISON

### C.1    IMPLEMENTATION OF GRAPH MODIFICATION ATTACK

Following Metattack (Zügner & Günnemann, 2019), we implement Graph Modification Attack by taking $A$ as a hyper-parameter. Nevertheless, since we are conducting evasion attack, we do not

have meta-gradients but the gradient of $A$ with respect to $\mathcal{L}_{\text{atk}}$, or $\nabla_A \mathcal{L}_{\text{atk}}$. Each step, we take the maximum entry in $\nabla_A \mathcal{L}_{\text{atk}}$, denoted with $\max(\nabla_A \mathcal{L}_{\text{atk}})$, and change the corresponding edge, if it is not contained in the training graph. Then we perform the perturbation as follows:

(a) If $\max(\nabla_A \mathcal{L}_{\text{atk}}) \leq 0$ and the corresponding entry in $A$ is 0, i.e., the edge does not exist before, we will add the edge.

(b) If $\max(\nabla_A \mathcal{L}_{\text{atk}}) \geq 0$ and the corresponding entry in $A$ is 1, i.e., the edge exists before, we will remove the edge.

If the selected entry can not satisfy neither of the above conditions, we will take the next maximum entry to perform the above procedure until we find one that satisfy the conditions. Here we exclude perturbations on node features given limited budgets, since Wu et al. (2019) observed the edge perturbations produce more harm than node perturbations. Besides, as shown in the proof, the damage brought by perturbations on node features is at most the damage brought by a corresponding injection to the targets in GIA, hence when given the same budgets to compare GMA and GIA, we can exclude the perturbations on nodes without loss of generality. Note that given the definitions of direct attack and influencer attack in Nettack (Zügner et al., 2018), our theoretical discussions are applicable to both direct GMA attack and indirect/influencer GMA attack, since the results are derived by establishing mappings between each kind of perturbations in GMA attack that are agnostic to these two types of GMA attacks. Moreover, the GMA attack evaluated in our experiments is exactly the direct attack. As in our case, all of the test nodes become victim nodes and the adversary is allowed to modify the connections and features of these nodes to perform the attack.

### C.2 IMPLEMENTATION OF GRAPH INJECTION ATTACK WITH PLURAL MAPPING

GIA with $\mathcal{M}_2$ is implemented based on the GMA above. For each edge appears in the perturbed graph produced by GMA but does not exist in the original graph, in GIA, we will inject a node to connect with the corresponding nodes of the edge. After injecting all of the nodes, then we use PGD (Madry et al., 2018) to optimize the features of the injected nodes.

## D   MORE HOMOPHILY DISTRIBUTIONS

### D.1   EDGE-CENTRIC HOMOPHILY

In addition to node-centric homophily (Def. 6), we can also define edge-centric homophily as:

**Definition D.1** (Edge-Centric Homophily). *The homophily for an edge $(u, v)$ can be defined as.*

$$h_e = sim(X_u, X_v), \tag{15}$$

*where $sim(\cdot)$ is also a distance metric, e.g., cosine similarity.*

With the definition above, we can probe the natural edge-centric homophily distribution of real-world benchmarks, as shown in Fig. 6. It turns out that the edge-centric homophily distributes follows a Gaussian prior. However, it seems to be improper to utilize edge-centric homophily to instantiate the homophily unnoticeability for several reasons. On the one hand, edge similarity does not consider the degrees of the neighbors which is misaligned with the popular aggregation scheme of GNNs. On the other hand, edge-centric and node-centric homophily basically perform similar functionality to retain the homophily, but if considering the future extension to high-order neighbor relationships, edge similarity might be harder to extend than node-centric homophily. Thus, we utilize the node-centric homophily for most of our discussions.

### D.2   MORE HOMOPHILY DISTRIBUTIONS CHANGES

We provide more homophily distribution results of the benchmarks we used in the experiments for Cora, Computers and Arxiv, shown as in Fig. 7 and Fig. 8, respectively. GIA is implemented with TDGIA (Zou et al., 2021). Note that the budgets for TDGIA here is different from that in the previous sections, which utilized the budgets resulting in the maximum harm when compared with GMA. Similarly, GIA without HAO would severely break the original homophily distribution hence making GIA can be easily defended by homophily defenders. While incorporated with HAO, GIA would retain the original homophily during attack.

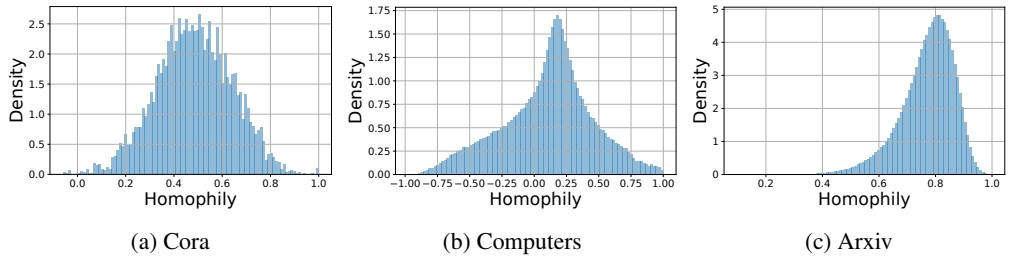

Figure 6: Edge-Centric homophily distributions

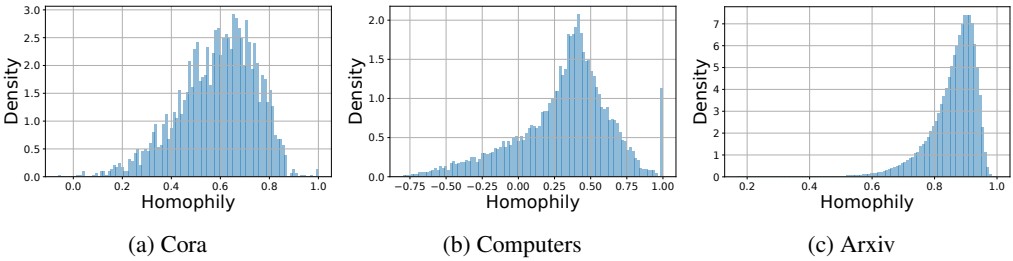

Figure 7: Homophily distributions before attack

## E  PROOFS AND DISCUSSIONS OF THEOREMS

### E.1  PROOF FOR THEOREM 1

**Theorem 1.** *Given moderate perturbation budgets* $\triangle_{GIA}$ *for GIA and* $\triangle_{GMA}$ *for GMA, that is, let* $\triangle_{GIA} \leq \triangle_{GMA} \ll |V| \leq |E|$, *for a fixed linearized GNN* $f_\theta$ *trained on* $\mathcal{G}$, *assume that* $\mathcal{G}$ *has no isolated nodes, and both GIA and GMA adversaries follow the optimal strategy, then,* $\forall \triangle_{GMA} > 0, \exists \triangle_{GIA} \leq \triangle_{GMA}$, *such that:*

$$\mathcal{L}_{atk}(f_\theta(\mathcal{G}'_{GIA})) - \mathcal{L}_{atk}(f_\theta(\mathcal{G}'_{GMA})) \leq 0,$$

*where* $\mathcal{G}'_{GIA}$ *and* $\mathcal{G}'_{GMA}$ *are the perturbed graphs generated by GIA and GMA, respectively.*

*Proof.* The proof sketch is to show that,

(a) Assume the given GNN model has $k$ layers, there exists a mapping, that when given the same budget, i.e., $\triangle_{\text{GIA}} = \triangle_{\text{GMA}} \ll |V| \leq |E|$, for each perturbation generated by GMA intended to attack node $u$ by perturbing edge $(u, v)$, or node attributes of node $u$ or some node $v$ that connects to $u$ within $k$ hops, we can always map it to a corresponding injection attack, that injects node $x_w$ to attack $u$, and lead to the same effects to the prediction.

(b) When the number of perturbation budget increases, the optimal objective values achieved of GIA is monotonically non-increasing with respect to $\triangle_{\text{GIA}}$, that is

$$\mathcal{L}_{\text{atk}}^{k+1}(f_\theta(\mathcal{G}'_{\text{GIA}})) \leq \mathcal{L}_{\text{atk}}^{k}(f_\theta(\mathcal{G}'_{\text{GIA}})),$$

where $\mathcal{L}_{\text{atk}}^{k}(f_\theta(\mathcal{G}'_{\text{GIA}}))$ is the optimal value achieved under the perturbation budget of $k$, which is obvious.

Once we prove both (a) and (b), the $\mathcal{L}_{\text{atk}}(f_\theta(\mathcal{G}'_{\text{GIA}}))$ will approach to $\mathcal{L}_{\text{atk}}^{k}(f_\theta(\mathcal{G}'_{\text{GMA}}))$ from the above as $\triangle_{\text{GIA}}$ approaches to $\triangle_{\text{GMA}}$, hence proving Theorem 1. Furthermore, for the flexibility of the constraints on $X_w$, we may adopt the gradient information of $X_w$ with respect to $\mathcal{L}_{\text{atk}}(f_\theta(\mathcal{G}'_{\text{GIA}}))$ to further optimize $X_w$ and make more damages. Hence, we have $\mathcal{L}_{\text{atk}}(f_\theta(\mathcal{G}'_{\text{GIA}})) \leq \mathcal{L}_{\text{atk}}^{k}(f_\theta(\mathcal{G}'_{\text{GMA}}))$.

To prove (a), the key technique is to show that, under a predefined mapping, there exist a corresponding injection matrix $A_{\text{atk}}$ along with the features of the injected nodes $X_{\text{atk}}$, such that the GIA

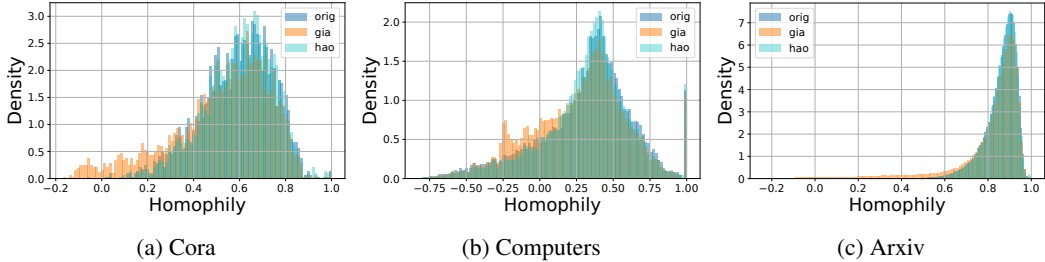

|(a) Cora|(b) Computers|(c) Arxiv|

Figure 8: Homophily distributions after attack

adversary can cause the same damage as GMA. The definition of the mapping mostly derives how the injection matrix is generated. While for the generation of $X_{\text{atk}}$, note that all of the input features $X$ is normalized to a specific range within $[-f_l, f_r]$ where $f_l, f_r \geq 0$, following previous works (Zheng et al., 2021). Thus, for any features $X_v \in \mathcal{D}_X$, $\alpha X_v \in \mathcal{D}_X$ when $0 \leq \alpha \leq 1$. We will use the statement multiple times during the derivation of $X_{\text{atk}}$.

Next, we will start to prove (a). Following Wu et al. (2019), in GMA, adding new connections between nodes from different classes produces the most benefit to the adversarial objective. Hence, given the limited perturbation budget, we give our primary focus to the action of connecting nodes from different classes and will prove (a) also holds for the remaining two actions, i.e., edge deletion and node attribute perturbation.

We prove (a) by induction on the number of linearized layers. First of all, we will show prove (a) holds for 1-layer and 2-layer linearized GNN as a motivating example. The model is as $f_\theta = \hat{A}^2 X \Theta$ with $H = \hat{A} X \Theta$ and $Z = f_\theta$.

**Plural Mapping $\mathcal{M}_2$.** Here we define the mapping $\mathcal{M}_2$ for edge addition. For each edge perturbation pair $(u, v)$ generated by GMA, we can insert a new node $w$ to connect $u$ and $v$. The influence of adversaries can be identified as follows, as $\Theta$ is fixed, we may exclude it for simplicity:
In layer (1):

- Clean graph:

$$H_i = \sum_{t \in \mathcal{N}(i) \cup \{i\}} \frac{1}{\sqrt{d_i d_t}} X_t \tag{16}$$

- GMA:

$$H_i' = \begin{cases} \sum_{t \in \mathcal{N}(i) \cup \{i\}} \frac{1}{\sqrt{d_t(d_i+1)}} X_t + \frac{1}{\sqrt{d_v(d_i+1)}} X_v, & i \in \{u\} \\ \sum_{t \in \mathcal{N}(i) \cup \{i\}} \frac{1}{\sqrt{d_t(d_i+1)}} X_t + \frac{1}{\sqrt{d_u(d_i+1)}} X_u, & i \in \{v\} \\ H_i, & i \notin \{u, v\} \end{cases} \tag{17}$$

- GIA:

$$H_i'' = \begin{cases} \sum_{t \in \mathcal{N}(i) \cup \{i\}} \frac{1}{\sqrt{d_t(d_i+1)}} X_t + \frac{1}{\sqrt{3(d_i+1)}} X_w, & i \in \{u, v\} \\ H_i, & u \notin \{u, v, w\} \\ \frac{1}{\sqrt{3}} \left( \frac{1}{\sqrt{d_u+1}} X_u + \frac{1}{\sqrt{d_v+1}} X_v + \frac{1}{\sqrt{3}} X_w \right), & i \in \{w\} \end{cases} \tag{18}$$

where $d_i$ refers to the degree of node $i$ with self-loops added for simplicity. Thus, in layer (1), to make the influence from GMA and GIA on node $u$ equal, the following constraint has to be satisfied:

$$\frac{1}{\sqrt{3(d_u+1)}} X_w = \frac{1}{\sqrt{(d_v+1)(d_u+1)}} X_v, \tag{19}$$

which is trivially held by setting

$$X_w = \frac{\sqrt{3}}{\sqrt{d_v + 1}} X_v. \tag{20}$$

Normally, GMA does not consider isolated nodes (Zügner et al., 2018; Zügner & Günnemann, 2019) hence we have $d_v \geq 2$ and $X_w \in \mathcal{D}_X$. Note that we can even change $X_w$ to make more affects to node $u$ with gradient information, then we may generate a more powerful perturbation in this way. Then, we go deeper to layer 2. In layer (2):

- Clean graph:

$$Z_i = \sum_{t \in \mathcal{N}(i) \cup \{i\}} \frac{1}{\sqrt{d_i d_t}} H_t \tag{21}$$

- GMA:

$$Z'_i = \begin{cases} \sum_{t \in \mathcal{N}(i)} \frac{H_t}{\sqrt{d_t(d_i + 1)}} + \frac{H'_i}{d_i + 1} + \frac{H'_v}{\sqrt{(d_v + 1)(d_i + 1)}}, & u \in \{u\} \\[2ex] \sum_{t \in \mathcal{N}(i)} \frac{H_t}{\sqrt{d_t(d_i + 1)}} + \frac{H'_i}{d_i + 1} + \frac{H'_u}{\sqrt{(d_u + 1)(d_i + 1)}}, & u \in \{v\} \\[2ex] \sum_{t \in \mathcal{N}(i)} \frac{H'_t}{\sqrt{d_t(d_i + 1)}}, & u \in \mathcal{N}(u) \cup \mathcal{N}(v) \\[2ex] Z_u, & \text{otherwise} \end{cases} \tag{22}$$

- GIA:

$$Z''_i = \begin{cases} \sum_{t \in \mathcal{N}(i)} \frac{1}{\sqrt{d_t(d_i + 1)}} H_t + \frac{1}{d_i + 1} H''_i + \frac{1}{\sqrt{3(d_i + 1)}} H''_w, & i \in \{u, v\} \\[2ex] H_u, & i \notin \{u, v, w\} \\[2ex] \frac{1}{\sqrt{3}} \left( \frac{1}{\sqrt{d_u + 1}} H''_u + \frac{1}{\sqrt{d_v + 1}} H''_v + \frac{1}{\sqrt{3}} H''_w \right), & i \in \{w\} \end{cases} \tag{23}$$

Similarly, to make $Z'_u = Z''_u$, we have to satisfy the following constraint:

$$\frac{1}{d_u + 1} H''_u + \frac{1}{\sqrt{3(d_u + 1)}} H''_w = \frac{1}{d_u + 1} H'_i + \frac{1}{\sqrt{(d_v + 1)(d_u + 1)}} H'_v,$$

$$\frac{\sqrt{3}}{d_u + 1} \sum_{t \in \mathcal{N}(u) \cup \{u\}} \frac{1}{\sqrt{d_t}} X_t + \frac{4}{3} X_w + \frac{1}{\sqrt{3}} \left( \frac{1}{\sqrt{d_u + 1}} X_u + \frac{1}{\sqrt{d_v + 1}} X_v \right)$$

$$=$$

$$\frac{\sqrt{3}}{d_u + 1} \sum_{t \in \mathcal{N}(u) \cup \{u\}} \frac{X_t}{\sqrt{d_t}} + \frac{\sqrt{3} X_v}{\sqrt{d_v + 1}} +$$

$$\frac{\sqrt{3}}{\sqrt{d_v + 1}} \left( \sum_{t \in \mathcal{N}(v) \cup \{v\}} \frac{X_t}{\sqrt{d_t(d_v + 1)}} + \frac{X_u}{\sqrt{(d_u + 1)(d_v + 1)}} \right), \tag{24}$$

$$\frac{4}{3} X_w + \frac{1}{\sqrt{3}} \left( \frac{1}{\sqrt{d_u + 1}} X_u + \frac{1}{\sqrt{d_v + 1}} X_v \right)$$

$$=$$

$$\frac{\sqrt{3} X_v}{\sqrt{d_v + 1}} + \frac{\sqrt{3}}{\sqrt{d_v + 1}} \left( \sum_{t \in \mathcal{N}(v) \cup \{v\}} \frac{X_t}{\sqrt{d_t(d_v + 1)}} + \frac{X_u}{\sqrt{(d_u + 1)(d_v + 1)}} \right),$$

then we let $X_w = \frac{3}{4}(\text{RHS} - \frac{1}{\sqrt{3}}(\frac{1}{\sqrt{d_u+1}} X_u + \frac{1}{\sqrt{d_v+1}} X_v))$ to get the solution of $X_w$ that makes the same perturbation. Similarly, we can infer $X_w \in \mathcal{D}_X$. The following proof also applies to layer 2.

Next, we will prove that, for a linearized GNN with $k$ layers ($k \geq 1$), i.e., $H^{(k)} = \hat{A}^k X \Theta$, once $\exists X_w$, such that the predictions for node $u$ is the same to that perturbed by GMA, i.e., $H_u^{(k-1)} = E_u^{(k-1)}$, then $\exists X_w'$, such that $H_u^{(k)} = E_u^{(k)}$. Here we use $H$ to denote the prediction of GNN attacked by GMA and $E$ for that of GIA. Note that, once the theorem holds, as we have already proven the existence for such $X_w$, it naturally generalizes to an arbitrary number of layers.

To be more specific, when $H_u^{(k-1)} = E_u^{(k-1)}$, we need to show that, $\exists X_w$, s.t.,

$$H_u^{(k)} = \sum_{j \in \mathcal{N}(u)} \frac{1}{\sqrt{d_u+1}\sqrt{d_j}} H_j^{(k-1)} + \frac{1}{d_u+1} H_u^{(k-1)} + \frac{1}{\sqrt{d_u+1}\sqrt{d_v+1}} H_v^{(k-1)},$$

$$E_u^{(k)} = \sum_{j \in \mathcal{N}(u)} \frac{1}{\sqrt{d_u+1}\sqrt{d_j}} E_j^{(k-1)} + \frac{1}{d_u+1} E_u^{(k-1)} + \frac{1}{\sqrt{d_u+1}\sqrt{3}} E_w^{(k-1)}, \quad (25)$$

$$H_u^{(k)} = E_u^{(k)}.$$

Here we make a simplification to re-write Eq. 25 by defining the influence score.

**Definition E.1** (Influence Score). *The influence score from node $v$ to $u$ after $k$ neighbor aggregations with a fixed GNN following Eq. 1, is the weight for $X_v$ contributing to $H_u^{(k)}$:*

$$H_u^{(k)} = \sum_{j \in \mathcal{N}(u) \cup \{u\}} I_{uj}^k \cdot X_j, \quad (26)$$

*which can be calculated recursively through:*

$$I_{uw}^k = \sum_{j \in \mathcal{N}(u) \cup \{u\}} (I_{uj} \cdot I_{jw}^{(k-1)}) + I_{uw}^{(k-1)}. \quad (27)$$

As $\Theta$ is fixed here, we can simply regard $I_{uv}^k = \hat{A}_{uv}^k$. Compared to the predictions after $k$-th propagation onto the clean graph, in GMA, $H_u^{(k)}$ is additionally influenced by node $v$, while in GIA, $H_u^{(k)}$ is additionally influenced by node $v$ and node $w$. Without loss of generality, we may absorb the influence from neighbors of node $v$ into that of node $v$. Hence we can rewrite Eq. 25 as the following:

$$\Delta H_u^{(k)} = I_{\text{GMA}_{uv}}^k X_v,$$
$$\Delta E_u^{(k)} = I_{\text{GIA}_{uv}}^k X_v + I_{\text{GIA}_{uw}}^k X_w, \quad (28)$$
$$\Delta H_u^{(k)} = \Delta E_u^{(k)},$$

where

$$I_{\text{GIA}_{uv}}^k = \sum_{j \in \mathcal{N}(u) \cup \{u\}} I_{\text{GIA}_{uj}} \cdot I_{\text{GIA}_{jv}}^{(k-1)} + I_{\text{GIA}_{uw}} \cdot I_{\text{GIA}_{wv}}^{(k-1)}.$$

Then we can further simplify it as,

$$(I_{\text{GMA}_{uv}}^k - I_{\text{GIA}_{uv}}^k) X_v = I_{\text{GIA}_{uw}}^k X_w. \quad (29)$$

To show the existence of $X_w$ that solves the above equation, it suffices to show $I_{\text{GIA}_{uw}}^k \neq 0$ and $X_w \in \mathcal{D}_X$. Note that $\exists X_w$ s.t.,

$$(I_{\text{GMA}_{uv}}^{(k-1)} - I_{\text{GIA}_{uv}}^{(k-1)}) X_v = I_{\text{GIA}_{uw}}^{(k-1)} X_w. \quad (30)$$

Since $\hat{A}^k \geq \mathbf{0}, \forall k \geq 0$, so we have $I_{\text{GIA}_{uw}}^{(k-1)} > 0$. Moreover,

$$I_{uw}^k = \sum_{j \in \mathcal{N}(u) \cup \{u\}} (\hat{A}_{uj} \cdot \hat{A}_{jw}^{(k-1)}) + I_{uw}^{(k-1)},$$

then it is obvious that the $I_{uw}^k > 0$. Moreover, with the definition of $I_{uv}^k = \hat{A}_{uv}^k$, it is obvious that $I_{\text{GIA}_{uw}}^{(k-1)} \geq I_{\text{GMA}_{uv}}^{(k-1)}$ for $v$ with a degree not less than 1 (i.e., $v$ is not an isolated node). Hence, we have $(I_{\text{GMA}_{uv}}^{(k-1)} - I_{\text{GIA}_{uv}}^{(k-1)})/I_{\text{GIA}_{uw}}^{(k-1)} \leq 1$ and $X_w \in \mathcal{D}_X$.

Now we have proved (a) holds for edge addition. For the remaining actions of GMA, we can use a new mapping $\mathcal{M}_1$ that injects one node $w$ to node $u$ to prove (a).

For an edge deletion of $(u, v)$, given $\mathcal{M}_1$, one may rewrite Eq. 25 for the left nodes other than $v$, as well as the equation involving $I_{uw}^k$, and derive the same conclusions similarly. Intuitively, for edge deletion, considering the classification probability, removing an edge is equivalent to enlarge the predicted classification probability for other classes, hence it fictionalizes likewise the edge addition and we can use a similar proof for this action.

Besides, $\mathcal{M}_1$ can also apply to the perturbation of features to node $u$ or the other neighbor nodes of $u$ within $k$ hops, where we inject one node $w$ to make the same effect. In this case, we can rewrite Eq. 25 and simplify it as following:

$$
\begin{aligned}
\Delta H_u^{(k)} &= I_{\text{GMA}_{uv}}^k \Delta X_v, \\
\Delta E_u^{(k)} &= I_{\text{GIA}_{uw}}^k X_w, \\
\Delta H_u^{(k)} &= \Delta E_u^{(k)},
\end{aligned}
\tag{31}
$$

where $v \in \{\mathcal{N}^k(u) \cup u\}$, i.e., node $u$ or its $k$-hop neighbor, and $\Delta X_v$ is the perturbation to the attributes of node $v$. Similarly, by the definition of $I_{uv}^k$, for node $v$ with a degree not less than $1$ (i.e., $v$ is not an isolated node), we have $I_{\text{GIA}_{uw}}^k \geq I_{\text{GMA}_{uv}}^k$, hence we have $I_{\text{GMA}_{uv}}^k / I_{\text{GIA}_{uw}}^k \leq 1$ and $X_w \in \mathcal{D}_X$.

Thus, we complete the whole proof. $\qquad\qquad\qquad\qquad\qquad\qquad\qquad\qquad\qquad\qquad\qquad$ □

**Theorem 1 for other GNNs**. We can extend Theorem 1 to other GNNs such as GCN, GraphSage, etc. Recall the theorem 1 in Xu et al. (2018):

**Lemma 1.** *Given a $k$-layer GNN following the neighbor aggregation scheme via Eq. 1, assume that all paths in the computation graph of the model are activated with the same probability of success $p$. Then the influence distribution $I_x$ for any node $x \in V$ is equivalent, in expectation, to the $k$-step random walk distribution on $\tilde{\mathcal{G}}$ starting at node $x$.*

To apply Lemma 1, we observe that the definition of $I_{uw}^k$ is analogous to random walk starting from node $u$. Thus, one may replace the definition of $I_{uw}^k$ here to the influence score defined by Xu et al. (2018), conduct a similar proof above with random walk score and obtain the same conclusions, given the mapping $\mathcal{M}_2$, for each edge addition $(u, v)$, $\exists X_w$, such that

$$
\mathbb{E}(\mathcal{L}_{\text{atk}}^k(f_\theta(\mathcal{G}_{\text{GIA}}'))) = \mathbb{E}(\mathcal{L}_{\text{atk}}^k(f_\theta(\mathcal{G}_{\text{GIA}}))).
\tag{32}
$$

Though the original theorem only proves Lemma 1 for GCN and GraphSage, it is obvious one can easily extend the proof in Xu et al. (2018) for aggregation scheme as Eq. 1.

**Cases for Less GIA Budget**. We can reduce GIA budgets in two ways.

(a) For GMA that performs both node feature perturbation and edge addition, considering a edge perturbation $(u, v)$, $\mathcal{M}_2$ essentially also applies for node feature perturbations on $u$ or $v$ without additional budgets.

(b) It is very likely that with the mapping above, GIA will produce many similar nodes. Hence, with one post-processing step to merge similar nodes together and re-optimize them again, GIA tends to require less budgets to make the same or more harm than GMA. That is also reflected in our experiments as shown in Fig. 1b.

## E.2    GIA with Plural Mapping for More GMA Operations

Here we explain how our theoretical results also apply to the remaining actions, i.e., edge deletion and node feature perturbation, of GMA with $\mathcal{M}_2$ (Def. 3.2). In the proof for Theorem 1, we have proved the existence of mappings for edge removal and node feature perturbation. Once the injected node features are set to have the same influence to the predictions on the targets, they can be further optimized for amplifying the damage, thus all of our theoretical results can be derived similarly like that for edge addition operation.

### E.3  PROOF FOR THEOREM 2

**Theorem 2.** *Given conditions in Theorem 1, consider a GIA attack, which* (i) *is mapped by* $\mathcal{M}_2$ *(Def. 3.2) from a GMA attack that only performs edge addition perturbations, and* (ii) *uses a linearized GNN trained with at least one node from each class in* $\mathcal{G}$ *as the surrogate model, and* (iii) *optimizes the malicious node features with PGD. Assume that* $\mathcal{G}$ *has no isolated node, and has node features as* $X_u = \frac{C}{C-1} e_{Y_u} - \frac{1}{C-1} \mathbf{1} \in \mathbb{R}^d$, *where* $Y_u$ *is the label of node* $u$ *and* $e_{Y_u} \in \mathbb{R}^d$ *is a one-hot vector with the* $Y_u$*-th entry being* 1 *and others being* 0. *Let the minimum similarity for any pair of nodes connected in* $\mathcal{G}$ *be* $s_{\mathcal{G}} = \min_{(u,v) \in E} sim(X_u, X_v)$ *with* $sim(X_u, X_v) = \frac{X_u \cdot X_v}{\|X_u\|_2 \|X_v\|_2}$. *For a homophily defender* $g_\theta$ *that prunes edges* $(u, v)$ *if* $sim(X_u, X_v) \leq s_{\mathcal{G}}$, *we have:*

$$\mathcal{L}_{atk}(g_\theta(\mathcal{M}_2(\mathcal{G}'_{GMA}))) - \mathcal{L}_{atk}(g_\theta(\mathcal{G}'_{GMA})) \geq 0.$$

*Proof.* We prove Theorem 2 by firstly show the following lemma.

**Lemma 2.** *Given conditions in Theorem 2, as the optimization on* $X_w$ *with respect to* $\mathcal{L}_{atk}$ *by PGD approaches, we have:*

$$sim(X_u, X_w)^{(t+1)} \leq sim(X_u, X_w)^{(t)},$$

*where* $t$ *is the number of optimization steps.*

We prove Lemma 2 in the follow-up section, i.e., Appendix E.4. With Lemma 2, known that GIA is mapped from GMA with $\mathcal{M}_2$, $X_w$ will be optimized to have the same effects as GMA at first and continue being optimized to a more harmful state, hence for the unit perturbation case as Fig. 2a, we know:

$$sim(X_u, X_w) \leq sim(X_u, X_v), \tag{33}$$

as the optimization on $X_w$ approaches. Furthermore, it follows:

$$h_u^{GIA} \leq h_u^{GMA}, \tag{34}$$

where $h_u^{GIA}$ and $h_u^{GMA}$ denote the homophily of node $u$ after GIA and GMA attack, respectively. Now if we go back to the homophily defender $g_\theta$, for any threshold specified to prune the edge $(u, v)$, as Lemma 2 and Eq. 33 indicates, direct malicious edges in GIA are more likely to be pruned by $g_\theta$. Let $\tau_{GIA}$ and $\tau_{GMA}$ denote the corresponding similarity between $(u, w)$ in GIA and $(u, v)$ in GMA, we have several possibilities compared with $s_{\mathcal{G}} = \min_{(u,v) \in E} sim(X_u, X_v)$:

(a) $\tau_{GIA} \leq \tau_{GMA} \leq s_{\mathcal{G}}$: all the malicious edges will be pruned, Theorem 2 holds;

(b) $\tau_{GIA} \leq s_{\mathcal{G}} \leq \tau_{GMA}$: all the GIA edges will be pruned, Theorem 2 holds;

(c) $s_{\mathcal{G}} \leq \tau_{GIA} \leq \tau_{GMA}$: this is unlikely to happen, otherwise $\tau_{GIA}$ can be optimized to even worse case, Theorem 2 holds;

Thus, we complete our proof. $\square$

Interestingly, we can also set a specific threshold $\tau_h$ for homophily defender s.t., $\tau_h - s_{\mathcal{G}} \leq \epsilon \geq 0$, where some of the original edges will be pruned, too. However, some of previous works indicate promoting the smoothness or slightly dropping some edges will bring better performance (Rong et al., 2020; Yang et al., 2021a; Zhao et al., 2021; Yang et al., 2021b). The similar discussion can also be applied to this case and obtain the same conclusions.

### E.4  PROOF FOR LEMMA 2

*Proof.* To begin with, without loss of generality, we may assume the number of classes is 2 and $Y_u = 0$, which can be similarly extended to the case of multi-class. With the feature assignment in the premise, let the label of node $u$ is $Y_u$, we have:

$$X_u = \begin{cases} [1, -1]^T, & Y_u = 0, \\ [-1, 1]^T, & Y_u = 1. \end{cases} \tag{35}$$

After setting it to having the same influence as that in GMA following Eq. 29, we have:

$$X_w = \frac{(I^k_{\text{GMA}_{uv}} - I^k_{\text{GIA}_{uv}})}{I^k_{\text{GIA}_{uw}}} X_v. \tag{36}$$

Then, let $\mathcal{L}_u$ denote the training loss $\mathcal{L}_{\text{train}}$ on node $u$, we can calculate the gradient of $X_w$:

$$\frac{\partial \mathcal{L}_u}{\partial X_u} = \frac{\partial \mathcal{L}_u}{\partial H_u^{(k)}} \cdot \frac{\partial H_u^{(k)}}{\partial X_w} = \frac{\partial \mathcal{L}_u}{\partial H_u^{(k)}} \cdot I^k_{\text{GIA}_{uw}} \cdot \Theta. \tag{37}$$

With Cross-Entropy loss, we further have:

$$\frac{\partial \mathcal{L}_u}{\partial H_u^{(k)}} = [-1, 1]^T. \tag{38}$$

Then, we can induce the update step of optimizing $X_w$ with respect to $\mathcal{L}_{\text{atk}} = -\mathcal{L}_{\text{train}}$ by PGD:

$$X_w^{(t+1)} = X_w^{(t)} + \epsilon \, \text{sign}(I^k_{\text{GIA}_{uw}} \cdot [-1, 1]^T \cdot \Theta), \tag{39}$$

where $t$ is the number of update steps. As the model is trained on at least nodes with indicator features following Eq. 35 from each class, without loss of generality, here we may assume $\Theta \geq \mathbf{0}$, the optimal $\Theta$ would converge to $\Theta \geq \mathbf{0}$. Thus,

$$\text{sign}(I^k_{\text{GIA}_{uw}} \cdot [-1, 1]^T \cdot \Theta) = \text{sign}(I^k_{\text{GIA}_{uw}} \cdot [-1, 1]^T).$$

Let us look into the change of cosine similarity between node $u$ and node $v$ as:

$$\Delta \text{sim}(X_u, X_w) = \alpha(X_u \cdot X_w^{(t+1)} - X_u \cdot X_w^{(t)}), \tag{40}$$

where $\alpha \geq 0$ is the normalized factor. To determine the sign of $\Delta \text{sim}(X_u, X_w)$, we may compare $X_u \cdot X^{(t+1)}$ with $X_u \cdot X_w^{(t)}$. Here we expand $X_u \cdot X_w^{(t+1)}$. Let $X_{u0}, X_{u1}$ to denote the first and second element in $X_u$ respectively, we have:

$$\begin{aligned} X_u \cdot X_w^{(t+1)} &= \frac{X_u \cdot X_w + \epsilon \, \text{sign}(I^k_{\text{GIA}_{uw}} \cdot [-1, 1]^T) X_u}{\|X_u\|_2 \cdot \left\| X_w^{(t+1)} \right\|_2}, \\ &= \frac{X_u \cdot X_w + \epsilon(X_{u1} - X_{u0})}{\|X_u\|_2 \sqrt{X_{w0}^2 + X_{w1}^2 + \epsilon^2 + 2\epsilon(X_{w1} - X_{w0})}}, \end{aligned} \tag{41}$$

where we omit the sign of $I^k_{\text{GIA}_{uw}}$ for $I^k_{\text{GIA}_{uw}} \geq 0$ according to the definition. Recall that we let $Y_u = 0$, hence we have $(X_{u1} - X_{u0}) \leq 0$. Besides, following Eq. 29, we have $\text{sign}(X_{w1} - X_{w0}) = \text{sign}(X_{v1} - X_{v0})$. As GMA tend to connect nodes from different classes, we further have $\text{sign}(X_{w1} - X_{w0}) \geq 0$. Comparing to $X_u \cdot X_w^{(t)}$, we know in Eq. 41, the numerator decreases and the denominator increases, as $\epsilon \geq 0$, so the overall scale decreases. In other words, we have:

$$\Delta \text{sim}(X_u, X_w) = \alpha(X_u \cdot X_w^{(t+1)} - X_u \cdot X_w^{(t)}) \leq 0, \tag{42}$$

which means that the cosine similarity between node $u$ and node $v$ decreases as the optimization of $X_w$ with respect to $\mathcal{L}_{\text{atk}}$ processes. Thus, we complete our proof for Lemma 2. □

### E.5 Proof for Theorem 3

**Theorem 3.** *Given conditions as Theorem 2, when $\lambda > 0$, we have $m(\mathcal{H}_{\mathcal{G}}, \mathcal{H}_{\mathcal{G}'_{HAO}}) \leq m(\mathcal{H}_{\mathcal{G}}, \mathcal{H}_{\mathcal{G}'_{GIA}})$, hence:*

$$\mathcal{L}_{atk}(g_\theta(\mathcal{G}'_{HAO})) - \mathcal{L}_{atk}(g_\theta(\mathcal{G}'_{GIA})) \leq 0,$$

*where $\mathcal{G}'_{HAO}$ is generated by GIA with HAO, and $\mathcal{G}'_{GIA}$ is generated by GIA without HAO.*

*Proof.* Similar with the proof for Theorem 2, we begin with binary classification, without loss of generality. With the feature assignment in the premise, let the label of node $u$ is $Y_u$, we have:

$$X_u = \begin{cases} [1, -1]^T, & Y_u = 0, \\ [-1, 1]^T, & Y_u = 1. \end{cases} \tag{43}$$

Let $\mathcal{L}_u$ denote the training loss $\mathcal{L}_{\text{train}}$ on node $u$, we look into the gradient of $X_w$ with respect to $\mathcal{L}_u$:

$$\frac{\partial \mathcal{L}_u}{\partial X_u} = \frac{\partial \mathcal{L}_u}{\partial H_u^{(k)}} \cdot \frac{\partial H_u^{(k)}}{\partial X_w} = \frac{\partial \mathcal{L}_u}{\partial H_u^{(k)}} \cdot I_{\text{GIA}_{uw}}^k \cdot \Theta. \tag{44}$$

With Cross-Entropy loss, we further have:

$$\frac{\partial \mathcal{L}_u}{\partial H_u^{(k)}} = [-1, 1]^T. \tag{45}$$

Together with HAO, we can infer the update step of optimizing $X_w$ with respect to $\mathcal{L}_{\text{atk}} = -\mathcal{L}_{\text{train}} + \lambda C(\mathcal{G}, \mathcal{G}')$ by PGD:

$$X_w^{(t+1)} = X_w^{(t)} + \epsilon \operatorname{sign}((I_{\text{GIA}_{uw}}^k \cdot [-1, 1]^T + \lambda [1, -1]^T) \cdot \Theta), \tag{46}$$

where $t$ is the number of update steps. Similarly, without loss of generality, we may assume $\Theta \geq \mathbf{0}$. As the optimization approaches, given $\lambda > 0$, GIA with HAO will early stop to some stage that $(I_{\text{GIA}_{uw}}^k \cdot [-1, 1]^T + \lambda [1, -1]^T) = \mathbf{0}$, hence similar to the proof of Theorem 2, it follows:

$$h_u^{\text{GIA}} \leq h_u^{\text{HAO}}, \tag{47}$$

where $h_u^{\text{GIA}}$ and $h_u^{\text{HAO}}$ denote the homophily of node $u$ after GIA and GIA with HAO attack, respectively. Likewise, we can infer that:

$$\mathcal{L}_{\text{atk}}(g_\theta(\mathcal{G}'_{\text{HAO}})) - \mathcal{L}_{\text{atk}}(g_\theta(\mathcal{G}'_{\text{GIA}})) \leq 0.$$

Thus, we complete our proof. □

### E.6 CERTIFIED ROBUSTNESS OF HOMOPHILY DEFENDER

Here we prove the certified robustness of homophily for a concrete GIA case. We prove via the decision margin as follows:

**Definition E.2** (Decision Margin). *Given a $k$-layer GNN, let $H_{[u,c]}^{(k)}$ denote the corresponding entry in $H_u^{(k)}$ for the class $c$, the decision margin on node $u$ with class label $Y_u$ can be denoted by:*

$$m_u = H_{[u,y_u]}^{(k)} - \max_{c \in \{0,..,C-1\}} H_{[u,c]}^{(k)}.$$

A Multi-Layer Perceptron (MLP) can be taken as a 0-layer GNN which the definition also applies. Then, we specify the certified robustness as follows:

**Proposition E.1** (Certified Robustness of Homophily Defender). *Consider a direct GIA attack uses a linearized GNN trained with at least one node from each class in $\mathcal{G}$, that targets at node $u$ by injecting a node $w$ connecting to $u$, let node features $x_u = \frac{C}{C-1}\text{onehot}(Y_u) - \frac{1}{C-1}\mathbf{1}$, the homophily of $u$ be $\tau$, the decision margin of a MLP on $u$ be $\gamma$, the minimum similarity for any pair of nodes connected in the original graph be $s_\mathcal{G} = \min_{(u,v) \in E} \text{sim}(X_u, X_v)$, homophily defender $g_\theta$ can defend such attacks, if $-\alpha \frac{1}{\sqrt{1+1/d_u}}(\tau + \beta\gamma) \leq s_\mathcal{G}$, and $g_\theta$ prunes edges $(u, v)$ s.t.,*

$$\text{sim}(X_u, X_w) \leq -\alpha \sqrt{\frac{1}{1 + 1/d_u}}(\tau + \beta\gamma),$$

*where $\alpha, \beta \geq 0$ are corresponding normalization factors.*

Intuitively, effective attacks on a node with higher degrees, homophily or decision margin require a lower similarity between node $w$ and $u$ hence more destruction to the homophily of node $u$. GIA without any constraints tends to optimize $\text{sim}(X_u, X_w)$ to a even lower value. Thus, it becomes easier to find a suitable condition for $g_\theta$, with which it can painlessly prune all vicious edges while keeping all original edges.

*Proof.* Analogous to the proof for Lemma 2, without loss of generality, we begin with binary classification, normalized indicator features and $Y_u = 0$ as follows:

$$X_u = \begin{cases} [1, -1]^T, & Y_u = 0, \\ [-1, 1]^T, & Y_u = 1. \end{cases} \tag{48}$$

The decision margin based on $k$-th layer representation can be denoted by

$$m = H_{[u,y_u]}^{(k)} - \max_{c \in \{0,..,C-1\}} H_{[u,c]}^{(k)}, \tag{49}$$

follows the Definition E.2. In our binary classification case, we have

$$\gamma = H_{[u,0]}^{(0)} - H_{[u,1]}^{(0)}, \tag{50}$$

where $H^{(0)}$ is the output of a 0-layer GNN, or MLP (Multi-Layer Perceptron). A $k$-layer GNN can be regarded as generating new hidden representation for node $u$ by aggregating its neighbors, hence, we may induce the decision margin for a $k$-layer GNN at node $u$ as

$$m = H_{[u,0]}^{(k)} - H_{[u,1]}^{(k)} = ([\sum_{j \in \mathcal{N}(u)} I_{uj} X_j]_{[0]} - [\sum_{j \in \mathcal{N}(u)} I_{uj} X_j]_{[1]}) + I_{uu}^{(k)} \gamma, \tag{51}$$

where we can replace the influence from neighbors with homophily of node $u$. Observe that $h_u$ essentially indicates how much neighbors of node $u$ contribute to $H_{[u,0]}^{(k)}$, for example, in binary case, let $\zeta > 0$ be the corresponding normalization factor,

$$h_u = \frac{1}{\zeta}([\sum_{j \in \mathcal{N}(u)} I_{uj} X_j]_{[0]} [X_u]_{[0]} + [\sum_{j \in \mathcal{N}(u)} I_{uj} X_j]_{[1]} [X_u]_{[1]}),$$

which means,

$$[\sum_{j \in \mathcal{N}(u)} I_{uj} X_j]_{[1]} = \frac{1}{[X_u]_{[1]}}(\zeta h_u - [\sum_{j \in \mathcal{N}(u)} I_{uj} X_j]_{[0]} [X_u]_{[0]}),$$

replaced with $X_u = [1, -1]^T$,

$$
\begin{aligned}
m &= H_{[u,0]}^{(k)} - H_{[u,1]}^{(k)} \\
&= ([\sum_{j \in \mathcal{N}(u)} I_{uj} X_j]_{[0]} - [\sum_{j \in \mathcal{N}(u)} I_{uj} X_j]_{[1]}) + \\
&= ([\sum_{j \in \mathcal{N}(u)} I_{uj} X_j]_{[0]} - \frac{1}{[X_u]_{[1]}}(\zeta h_u - [\sum_{j \in \mathcal{N}(u)} I_{uj} X_j]_{[0]} [X_u]_{[0]})) + I_{uu}^{(k)} \gamma \\
&= \zeta h_u + I_{uu}^{(k)} \gamma.
\end{aligned}
\tag{52}
$$

Hence, we have:

$$m = H_{[u,0]}^{(k)} - H_{[u,1]}^{(k)} = \zeta h_u + I_{uu}^{(k)} \gamma,$$

where $\zeta \geq 0$ is the factor of $h_u$. With node $w$ injected, the margin can be rewritten as:

$$m' = \sqrt{\frac{d_u}{d_u + 1}} m + I_{uw}^{(k)}(X_{[w,0]} - X_{[w,1]}). \tag{53}$$

To perturb the prediction of node $u$, we make $m \leq 0$, hence, we have

$$m' = \sqrt{\frac{d_u}{d_u + 1}} m + I_{uw}^{(k)}(X_{[w,0]} - X_{[w,1]}) \leq 0,$$

$$I_{uw}^{(k)}(X_{[w,1]} - X_{[w,0]}) \geq \sqrt{\frac{d_u}{d_u + 1}} m, \tag{54}$$

$$(X_{[w,1]} - X_{[w,0]}) \geq \frac{1}{I_{uw}^{(k)}} \sqrt{\frac{d_u}{d_u + 1}} (\zeta h_u + I_{uu}^{(k)} \gamma).$$

Observe that $\text{sim}(X_u, X_w) = (X_{[w,0]} - X_{[w,1]})$ and $h_u = \tau$, hence, we can write Eq. 54 in a clean form as

$$\text{sim}(X_u, X_w) \leq -\alpha\sqrt{\frac{d_u}{d_u + 1}}(\tau + \beta\gamma), \tag{55}$$

where $\alpha, \beta$ are corresponding normalization factors whose signs are determined by signs of $I_{uw}^k$ and $I_{uu}^k$ respectively. In other words, GIA has to optimize $X_w$ satisfying the above requirement to make the attack *effective*, however, given the premise that all $s_{\mathcal{G}} = \min_{(u,v)\in E}\text{sim}(X_u, X_v) \geq -\alpha\sqrt{\frac{d_u}{d_u+1}}(\tau + \beta\gamma)$, a defense model $g_\theta$ will directly prune all of the vicious edges satisfying the above requirement and make the attack *ineffective*, which is exactly what we want to prove. $\qquad\square$

## F  MORE IMPLEMENTATIONS OF HOMOPHILY DEFENDER

There are many ways to design homophily defenders, inheriting the spirit of recovering the original homophily. In addition to edge pruning, one could leverage variational inference to learn the homophily distribution or the similarity distribution among neighbors. Then we use adversarial training to train the model to denoise. Similarly, learning to promote the smoothness of the graph can also be leveraged to build homophily defenders (Zhao et al., 2021; Yang et al., 2021a;b). Besides, outlier detection can also be adopted to remove or reduce the aggregation weights of malicious edges or nodes. In the following two subsections, we will present two variants that perform better than GNNGuard (Zhang & Zitnik, 2020).

### F.1  DETAILS OF EFFICIENT GNNGUARD

The originally released GNNGuard requires $O(n^2)$ computation for node-node similarity, making it prohibitive to run on large graphs. To this end, we implement an efficient alternative of GNNGuard adopting a similar message passing scheme, let $\tau$ be the threshold to prune an edge:

$$H_u^{(k)} = \sigma(W_k \cdot \sum_{j\in\mathcal{N}(u)\cup\{u\}} \alpha_{uj}H_j^{(k-1)}), \tag{56}$$

where

$$\alpha_{uj} = \text{softmax}\left(\frac{z_{uj}}{\sum_{v\in\mathcal{N}(u)\cup\{u\}} z_{uv}}\right),$$

and

$$z_{uj} = \begin{cases} \dfrac{\mathbf{1}\{\text{sim}(H_j^{(k-1)}, H_u^{(k-1)}) > \tau\} \cdot \text{sim}(H_j^{(k-1)} \cdot H_u^{(k-1)})}{\sum_{v\in\mathcal{N}(u)}\mathbf{1}\{\text{sim}(H_v^{(k-1)}, H_u^{(k-1)}) > \tau\} \cdot \text{sim}(H_v^{(k-1)}, H_u^{(k-1)})}, & u \neq j, \\[3ex] \dfrac{1}{\sum_{v\in\mathcal{N}(u)}\mathbf{1}\{\text{sim}(H_v^{(k-1)} \cdot H_u^{(k-1)}) > \tau\} + 1} & u = j. \end{cases}$$

Essentially, it only requires $O(E)$ complexity. We will present the performance of Efficient GNN-Guard (EGNNGuard) in table 3.

### F.2  DETAILS OF ROBUST GRAPH ATTENTION NETWORK (RGAT)

We introduce another implementation of Robust Graph Attention Network (RGAT). We adopt the same spirit of GCNGuard (Zhang & Zitnik, 2020), that eliminates unlike neighbors during message passing based on neighbor similarity. Specifically, we change the standard GAT (Veličković et al., 2018) attention mechanism as

$$\alpha_{i,j} = \frac{\mathbb{1}\{\text{sim}(x_i, x_j) \geq \tau\}}{\sum_{k\in\mathcal{N}(i)\cup\{i\}}\mathbb{1}\{\text{sim}(x_i, x_k) \geq \tau\}},$$

Additionally, we also adopt the idea of RobustGCN (Zhu et al., 2019) that stabilize the hidden representations between layers, so we add Layer Normalization (Ba et al., 2016) among layers of RGAT. Empirical experiments show that RGAT is a more robust model with or without GIA attacks. For more details, we refer readers to Table 3.

Table 3: Performance of homophily defenders used in experiments

| Model | Natural Accuracy | Test Robustness | Running Time |
|---|---|---|---|
| GNNGuard | 83.58 | 64.96 | $1.76 \times 10^{-3}$ |
| EGNNGuard | 84.45 | 64.27 | $\mathbf{5.39 \times 10^{-5}}$ |
| RGAT | **85.75** | **66.57** | $6.03 \times 10^{-5}$ |
| GCN | 84.99 | 36.62 | $5.87 \times 10^{-5}$ |

### F.3 PERFORMANCE OF HOMOPHILY DEFENDERS

We test the performance of different homophily defenders on Cora. Natural Accuracy refers to the test accuracy on clean graph. Test Robustness refers to their averaged performance against all the attacks. Running time refers to their averaged running time for one training epoch. We repeat the evaluation 10 times to obtain the average accuracy. We can see that EGNNGuard has competitive performance with GNNGuard while $20\times$ faster. RGAT performs slightly better and $10\times$ faster. Hence, for large graphs and adversarial training of GNNGuard, we will use EGNNGuard instead.

## G MORE DETAILS ABOUT ALGORITHMS USED

Here we provide detailed descriptions of algorithms mentioned in Section. 4.2.

### G.1 DETAILS OF METAGIA AND AGIA

#### G.1.1 INDUCTION OF META GRADIENTS FOR METAGIA

With the bi-level optimization formulation of GIA, similar to meta-attack, we can infer the meta-gradients as follows:

$$\nabla_{A_{\text{atk}}}^{meta} = \nabla_{A_{\text{atk}}} \mathcal{L}_{\text{atk}}(f_{\theta^*}(A_{\text{atk}}, X_{\text{atk}}^*)), \quad X_{\text{atk}}^* = \text{opt}_{X_{\text{atk}}} \mathcal{L}_{\text{atk}}(f_{\theta^*}(A_{\text{atk}}, X_{\text{atk}})). \tag{57}$$

Consider the opt process, we have

$$X_{\text{atk}}^{(t+1)} = X_{\text{atk}}^{(t)} - \alpha \nabla_{X_{\text{atk}}^{(t)}} \mathcal{L}_{\text{atk}}(f_{\theta^*}(A_{\text{atk}}, X_{\text{atk}}^{(t)})). \tag{58}$$

With that, we can derive the meta-gradient for $A_{\text{atk}}$:

$$\nabla_{A_{\text{atk}}}^{meta} = \nabla_{A_{\text{atk}}} \mathcal{L}_{\text{atk}}(f_{\theta^*}(A_{\text{atk}}, X_{\text{atk}}^*))$$
$$= \nabla_{X_{\text{atk}}} \mathcal{L}_{\text{atk}}(f_{\theta^*}(A_{\text{atk}}, X_{\text{atk}}^{(t)})) \cdot [\nabla_{A_{\text{atk}}} f_{\theta^*}(A_{\text{atk}}, X_{\text{atk}}^{(t)}) + \nabla_{X_{\text{atk}}^{(t)}} f_{\theta^*}(A_{\text{atk}}, X_{\text{atk}}^{(t)}) \cdot \nabla_{A_{\text{atk}}} X_{\text{atk}}^{(t)}], \tag{59}$$

where

$$\nabla_{A_{\text{atk}}} X_{\text{atk}}^{(t+1)} = \nabla_{A_{\text{atk}}} X_{\text{atk}}^{(t)} - \alpha \nabla_{A_{\text{atk}}} \nabla_{X_{\text{atk}}^{(t)}} \mathcal{L}_{\text{atk}}(f_{\theta^*}(A_{\text{atk}}, X_{\text{atk}}^{(t)})). \tag{60}$$

Note that $X_{\text{atk}}^{(t)}$ depends on $A_{\text{atk}}$ according to Eq. 58, so the derivative w.r.t. $A_{\text{atk}}$ need to be traced back. Finally, the update schema for $A_{\text{atk}}$ is as follows:

$$A_{\text{atk}}^{(t+1)} = A_{\text{atk}}^{(t)} - \beta \nabla_{A_{\text{atk}}^{(t)}}^{meta}. \tag{61}$$

Directly computing the meta gradients is expensive, following Metattack, we adopt approximations like MAML (Finn et al., 2017) for efficiency consideration. We refer readers to the paper of Metattack for the detailed algorithms by replacing the corresponding variables with those above.

### G.2 DETAILS OF AGIA

For optimizing weights of edge entries in $A_{\text{atk}}$, we can use either Adam (Kingma & Ba, 2015), PGD (Madry et al., 2018) or other optimization methods leveraging gradients. For simplicity, we use PGD to illustrate the algorithm description of AGIA as follows:

---

**Algorithm 1:** AGIA: Adaptive Graph Injection Attack with Gradient

---

**Input:** A graph $\mathcal{G} = (A, X)$, a trained GNN model $f_{\theta^*}$, number of injected nodes $c$, degree budget $b$, outer attack epochs $e_{\text{outer}}$, inner attack epochs for node features and adjacency matrix $e_{\text{inner}}^X, e_{\text{inner}}^A$, learning rate $\eta$, weight for sparsity penalty $\beta$, weight for homophily penalty $\lambda$ ;

**Output:** Perturbed graph $\mathcal{G}' = (A', X')$;

1 Random initialize injection parameters $(A_{\text{atk}}, X_{\text{atk}})$;
2 $Y_{\text{orig}} \leftarrow f_{\theta^*}(A, X)$;   /* Obtain original predictions on clean graph */
3 **for** *epoch $\leftarrow 0$ to $e_{\text{outer}}$* **do**
4 $\quad$ Random initialize $X_{\text{atk}}$;
5 $\quad$ **for** *epoch $\leftarrow 0$ to $e_{\text{inner}}^X$* **do**
6 $\quad\quad$ $A' \leftarrow A \parallel A_{\text{atk}}, X' \leftarrow X \parallel X_{\text{atk}}$ ;
7 $\quad\quad$ $X_{\text{atk}} \leftarrow \text{Clip}_{(x_{\min}, x_{\max})}(X_{\text{atk}} - \eta \cdot \nabla_{X_{\text{atk}}}(\mathcal{L}_{\text{atk}}^h))$ ;
8 $\quad$ **for** *epoch $\leftarrow 0$ to $e_{\text{inner}}^A$* **do**
9 $\quad\quad$ $A' \leftarrow A \parallel A_{\text{atk}}, X' \leftarrow X \parallel X_{\text{atk}}$ ;
10 $\quad\quad$ $A_{\text{atk}} \leftarrow \text{Clip}_{(0,1)}(A_{\text{atk}} - \eta \cdot \nabla_{A_{\text{atk}}}(\mathcal{L}_{\text{atk}}^A))$ ;
11 $\quad$ $A_{\text{atk}} \leftarrow \parallel_{i=1}^{k} \arg\max_{\text{top } b}(A_{\text{atk}[i,:]})$ ;

---

Here, $\mathcal{L}_{\text{atk}}^h$ refers to the objective of GIA with HAO for the optimization of $X_{\text{atk}}$. For the optimization of $A_{\text{atk}}$, we empirically find the $\lambda_A$ would degenerate the performance, which we hypothesize that is because of the noises as $A_{\text{atk}}$ is a discrete variable. Hence, we set $\lambda_A = 0$ in our experiments. Additionally, we introduce a sparsity regularization term for the optimization of $A_{\text{atk}}$:

$$\mathcal{L}_{\text{atk}}^A = \mathcal{L}_{\text{atk}} + \beta \frac{1}{|V_{\text{atk}}|} \sum_{u \in V_{\text{atk}}} |b - \|A_{\text{atk}_{u,:}}\|_1|. \tag{62}$$

Besides, we empirically observe that Adam performs better than PGD. Hence, we would use Adam for AGIA in our experiments, and leave other methods for future work. Adopting Adam additionally brings the benefits to utilize momentum and history information to accelerate the optimization escape from the local optimum, which PGD fails to achieve.

### G.3 DETAILS OF SEQGIA

Since gradient methods require huge computation overhead, we propose a novel divide-and-conquer strategy to iteratively select some of the most vulnerable targets with Eq. 11 to attack. Note that it is different from traditional sequential injection methods which still connect the targets in full batch.

For simplicity, we also illustrate the algorithm with PGD, and one may switch to other optimizer such as Adam to optimize $A_{\text{atk}}$. The detailed algorithm is as follows:

---

**Algorithm 2:** SeqGIA: Sequential Adaptive Graph Injection Attack

---

**Input:** A graph $\mathcal{G} = (A, X)$, a trained GNN model $f_{\theta^*}$, number of injected nodes $k$, degree budget $b$, outer attack epochs $e_{\text{outer}}$, inner attack epochs for node features and adjacency matrix $e_{\text{inner}}^X, e_{\text{inner}}^A$, learning rate $\eta$, weight for sparsity penalty $\beta$, weight for homophily penalty $\lambda$, sequential step for vicious nodes $\gamma_{\text{atk}}$, sequential step for target nodes $\gamma_c$ ;
**Output:** Perturbed graph $\mathcal{G}' = (A', X')$;
1 Initialize injection parameters $(A_{\text{atk}}, X_{\text{atk}})$;
2 $\boldsymbol{Y}_{\text{orig}} \leftarrow f_{\theta^*}(A, X)$ ;
                           /* Obtain original predictions on clean graph */
3 **while** *Not Injecting All Nodes* **do**
4      $n_{\text{atk}} \leftarrow \gamma_{\text{atk}} * |V_{\text{atk}}|; n_c \leftarrow \gamma_c * |V_c|$ ;
5      Ranking and selecting $n_c$ targets with Eq. 11;
6      Random initialize $A_{\text{atk}}^{(\text{cur})} \in \mathbb{R}^{n_c \times n_{\text{atk}}}, X_{\text{atk}}^{(\text{cur})} \in \mathbb{R}^{n_{\text{atk}} \times d}$ ;
7      **for** *epoch $\leftarrow$ 0 to $e_{outer}$* **do**
8          **for** *epoch $\leftarrow$ 0 to $e_{inner}^X$* **do**
9              $A' \leftarrow A \parallel A_{\text{atk}} \parallel A_{\text{atk}}^{(\text{cur})}, X' \leftarrow X \parallel X_{\text{atk}} \parallel X_{\text{atk}}^{(\text{cur})}$;
10              $X_{\text{atk}}^{(\text{cur})} \leftarrow \text{Clip}_{(x_{\min}, x_{\max})}(X_{\text{atk}}^{(\text{cur})} - \eta \cdot \nabla_{X_{\text{atk}}^{(\text{cur})}}(\mathcal{L}_{\text{atk}}^h))$ ;
11          **for** *epoch $\leftarrow$ 0 to $e_{inner}^A$* **do**
12              $A' \leftarrow A \parallel A_{\text{atk}} \parallel A_{\text{atk}}^{(\text{cur})}, X' \leftarrow X \parallel X_{\text{atk}} \parallel X_{\text{atk}}^{(\text{cur})}$;
13              $A_{\text{atk}}^{(\text{cur})} \leftarrow \text{Clip}_{(0,1)}(A_{\text{atk}}^{(\text{cur})} - \eta \cdot \nabla_{A_{\text{atk}}^{(\text{cur})}}(\mathcal{L}_{\text{atk}}^A))$ ;
14          $A_{\text{atk}}^{(\text{cur})} \leftarrow \|_{i=1}^{n_{\text{atk}}} \arg\max_{\text{top } b}(A_{\text{atk}[i,:]}^{(\text{cur})})$ ;
15      $A_{\text{atk}} = A_{\text{atk}} \parallel A_{\text{atk}}^{(\text{cur})}; X_{\text{atk}} = X_{\text{atk}} \parallel X_{\text{atk}}^{(\text{cur})}$;

---

Actually, one may also inject few nodes via heuristic based algorithms first, then inject the left nodes with gradients sequentially. Assume that $\alpha$ nodes are injected by heuristic, we may further optimize the complexity from

$$O\left(\frac{1}{\gamma_{\text{atk}}}(|V_c| \log |V_c| + e_{\text{outer}}(e_{\text{inner}}^A |V_c| \gamma_c |V_{\text{atk}}| + e_{\text{inner}}^X |V_{\text{atk}}| d))N_{V_c}\right)$$

to

$$O\left(\alpha \frac{1}{\gamma_{\text{atk}}}(|V_c| \log |V_c| + |V_{\text{atk}}| b + e_{\text{inner}}^X |V_{\text{atk}}| d)N_{V_c} + \right.$$
$$\left.(1-\alpha)\frac{1}{\gamma_{\text{atk}}}(|V_c| \log |V_c| + e_{\text{outer}}(e_{\text{inner}}^A |V_c| \gamma_c |V_{\text{atk}}| + e_{\text{inner}}^X |V_{\text{atk}}| d))N_{V_c}\right)$$

in Table 7.

## H    MORE DETAILS ABOUT THE EXPERIMENTS

### H.1    STATISTICS AND BUDGETS OF DATASETS

Here we provide statistics of datasets used in the experiments as Sec. 5.1. The label homophily utilizes the previous homophily definition (Zhu et al., 2020), while the avg. homophily utilizes the node-centric homophily based on node similarity.

Following previous works (Zou et al., 2021; Zheng et al., 2021), we heuristically specify the budgets for each dataset according to the the number of target nodes and average degrees.

For targeted attack, we follow previous works (Zügner et al., 2018) to select 800 nodes as targets according to the classification margins of the surrogate model. Specifically, we select 200 nodes with the highest classification margin, 200 nodes with lowest classification margin and 400 randomly. For the budgets, we scale down the number of injected nodes and the maximum allowable degrees accordingly.

Table 4: Statistics of datasets

| Datasets | Nodes | Edges | Classes | Avg. Degree | Label Homophily | Avg. Homophily |
|---|---|---|---|---|---|---|
| Cora | 2680 | 5148 | 7 | 3.84 | 0.81 | 0.59 |
| Citeseer | 3191 | 4172 | 6 | 2.61 | 0.74 | 0.90 |
| Computers | 13,752 | 245,861 | 10 | 35.76 | 0.77 | 0.31 |
| Arxiv | 169,343 | 1,166,243 | 40 | 13.77 | 0.65 | 0.86 |
| Aminer | 659,574 | 2,878,577 | 18 | 8.73 | 0.65 | 0.38 |
| Reddit | 232,965 | 11,606,919 | 41 | 99.65 | 0.78 | 0.31 |

Table 5: Budgets for non-targeted attacks on different datasets

| Datasets | Nodes | Degree | Node Per.(%) | Edge Per.(%) |
|---|---|---|---|---|
| Cora | 60 | 20 | 2.24% | 23.31% |
| Citeseer | 90 | 10 | 2.82% | 21.57% |
| Computers | 300 | 150 | 2.18% | 18.30% |
| Arxiv | 1500 | 100 | 0.71% | 10.29% |

Table 6: Budgets of targeted attacks on different datasets

| Datasets | Nodes | Degree | Node Per.(%) | Edge Per.(%) |
|---|---|---|---|---|
| Computers | 100 | 150 | 0.73% | 6.1% |
| Arxiv | 120 | 100 | 0.07% | 1.03% |
| Aminer | 150 | 50 | 0.02% | 0.26% |
| Reddit | 300 | 100 | 0.13% | 0.26% |

## H.2 ADDITIONAL DISCUSSIONS ABOUT ATTACK BASELINES

For the selection of attack baselines, from the literature reviews (Sun et al., 2018; Jin et al., 2021), existing reinforcement learning (RL) based approaches adopt different settings from ours, which either focus on the poisoning attack, transductive learning, edge perturbation or other application tasks. Even for NIPA (Sun et al., 2020) which has the closest setting to ours, since it focuses on poisoning and transductive attack, and the features of the injected nodes are generated heuristically according to the labels assigned by the RL agent, without author released code, the adaption requires lots of efforts including redesigning the markov decision process in NIPA, hence we would like to leave them for future work. More discussions on RL based future works are given in Appendix A.2.

## H.3 COMPLEXITY OF ALGORITHMS

Here we provide complexity analyses of the GIA algorithms used in the experiments as discussed and selected in Sec. 5.1. As also defined in algorithm description section from Appendix G, $e_{\mathrm{inner}}^X$ is the number of epochs optimized for node features, $b$ is the number of maximum degree of vicious nodes, $d$ is the number of feature dimension, $N_{V_c}$ is the number of $k$-hop neighbors of the victim nodes for perform one forwarding of a $k$-layer GNN, $e_{\mathrm{outer}}$ is the number of epochs for optimizing $A_{\mathrm{atk}}$, $\gamma_c$ is the ratio of target nodes to attack in one batch, $\gamma_{\mathrm{atk}}$ is the ratio of vicious nodes to inject in one batch.

## H.4 DETAILS OF DEFENSE BASELINES

Here we provide the categories of defense models used in the experiments as Sec. 5.1. We categorize all models into Vanilla, Robust and Extreme Robust (Combo). Basically, popular GNNs are belong to vanilla category, robust GNNs are belong to robust categorty, and a robust trick will enhance the robust level by one to the next Category. Consistently to the observation in GRB (Zheng et al., 2021), we find adding Layer Normalization (Ba et al., 2016) before or between convulotion layers can enhance the model robustness. We use LN to denote adding layer norm before the first convulotion layer and LNi to denote adding layer norm between convulotion layers.

Table 7: Complexity of various attacks

| Type | Algorithm | Time Complexity | Space Complexity |
|---|---|---|---|
| Gradient | MetaGIA | $O(|V_{\text{atk}}|b(|V_c||V_{\text{atk}}|\log(|V_c||V_{\text{atk}}|) + e^X_{\text{inner}}d(|V_{\text{atk}}| + N_{V_c})))$ | $O(|V_c||V_{\text{atk}}| + e^X_{\text{inner}}d(|V_{\text{atk}}| + N_{V_c}))$ |
| | AGIA | $O(e_{\text{outer}}(e^A_{\text{inner}}|V_c||V_{\text{atk}}| + (e^A_{\text{inner}} + e^X_{\text{inner}})d(N_{V_c} + |V_{\text{atk}}|)))$ | $O(|V_c||V_{\text{atk}}| + e^X_{\text{inner}}d(|V_{\text{atk}}| + N_{V_c}))$ |
| | AGIA-SeqGIA | $O(e_{\text{outer}}(|V_c|\log(|V_c|) + e^A_{\text{inner}}\gamma_c|V_c||V_{\text{atk}}| + (e^A_{\text{inner}} + e^X_{\text{inner}})d(N_{V_c} + |V_{\text{atk}}|)))$ | $O(\gamma_c|V_c|\gamma_{\text{atk}}|V_{\text{atk}}| + e^X_{\text{inner}}d(|V_{\text{atk}}| + N_{V_c}))$ |
| Heuristic | PGD | $O(|V_{\text{atk}}|b + e^X_{\text{inner}}d(|V_{\text{atk}}| + N_{V_c}))$ | $O(|V_{\text{atk}}|b + e^X_{\text{inner}}d(|V_{\text{atk}}| + N_{V_c}))$ |
| | TDGIA | $O((|V_c|\log|V_c| + |V_{\text{atk}}|b + e^X_{\text{inner}}d(|V_{\text{atk}}| + N_{V_c}))$ | $O(|V_{\text{atk}}|b + e^X_{\text{inner}}d(|V_{\text{atk}}| + N_{V_c}))$ |
| | ATDGIA | $O(|V_c|\log|V_c| + |V_{\text{atk}}|b + e^X_{\text{inner}}d(|V_{\text{atk}}| + N_{V_c}))$ | $O(|V_{\text{atk}}|b + e^X_{\text{inner}}d(|V_{\text{atk}}| + N_{V_c}))$ |

Table 8: Defense model categories

| Model | Category | Model | Category | Model | Category | Model | Category |
|---|---|---|---|---|---|---|---|
| GCN | Vanilla | GCN+LN | Robust | GCN+LNi | Robust | GCN+FLAG | Robust |
| GCN+LN+LNi | Combo | GCN+FLAG+LN | Combo | GCN+FLAG+LNi | Combo | GCN+FLAG+LN+LNi | Combo |
| Sage | Vanilla | Sage+LN | Robust | Sage+LNi | Robust | Sage+FLAG | Robust |
| Sage+LN+LNi | Combo | Sage+FLAG+LN | Combo | Sage+FLAG+LNi | Combo | Sage+FLAG+LN+LNi | Combo |
| GAT | Vanilla | GAT+LN | Robust | GAT+LNi | Robust | GAT+FLAG | Robust |
| GAT+LN+LNi | Combo | GAT+FLAG+LN | Combo | GAT+FLAG+LNi | Combo | GAT+FLAG+LN+LNi | Combo |
| Guard | Robust | Guard+LN | Combo | Guard+LNi | Combo | EGuard+FLAG | Combo |
| Guard+LN+LNi | Combo | EGuard+FLAG+LN | Combo | EGuard+FLAG+LNi | Combo | EGuard+FLAG+LN+LNi | Combo |
| RGAT | Robust | RGAT+LN | Combo | RGAT+FLAG | Combo | RGAT+FLAG+LN | Combo |
| RobustGCN | Robust | RobustGCN+FLAG | Combo | | | | |

## H.5 DETAILS OF EVALUATION AND MODEL SETTINGS

### H.5.1 MODEL SETTING

By default, all GNNs used in our experiments have 3 layers, a hidden dimension of 64 for Cora, Citeseer, and Computers, a hidden dimension of 128 for the rest medium to large scale graphs. We also adopt dropout (Srivastava et al., 2014) with dropout rate of 0.5 between each layer. The optimizer we used is Adam (Kingma & Ba, 2015) with a learning rate of 0.01. By default, we set total training epochs as 400 and employ the early stop of 100 epochs according to the validation accuracy. For the set of threshold in homophily defenders, we use PGD (Madry et al., 2018) to find the threshold which performs well on both the clean data and perturbed data. By default, we set the threshold as 0.1, while for Computers and Reddit, we use 0.15 for Guard and EGuard, and for Citeseer and Arxiv we use 0.2 for RGAT.

For adversarial training with FLAG (Kong et al., 2020), we set the step size be $1 \times 10^{-3}$, and train 100 steps for Cora, 50 steps for Citeseer, 10 steps for the rest datasets. We empirically observe that FLAG can enhance both the natural accuracy and robustness of GNNs. We refer readers to the results for more details in Sec. J.1 and Sec. J.2.

### H.5.2 EVALUATION SETTING

For final model selection, we select the final model with best validation accuracy. For data splits, we follow the split methods in GRB (Zheng et al., 2021) which splits the datasets according to the node degrees, except for non-targeted attack on Arxiv where we use the official split to probe the performances of various methods in a natural setting. For non-targeted attack, following previous works (Zou et al., 2021; Zheng et al., 2021), we select all test nodes as targets. While for targeted attacks, we follow previous works (Zügner et al., 2018) to select 200 nodes with highest classification margin and lowest classification margin of the surrogate model. Then we randomly select 400 nodes as targets. In other words, there are 800 target nodes in total for targeted attack. Note for targeted attack, the natural accuracy on the target nodes might be different from normal test accuracy.

We also follow previous works to specify the attack budgets as Table. 5 for non-targeted attack and Table. 6 for targeted attack.

During evaluation, we follow the black-box setting. Specifically, we firstly use the surrogate model to generate the perturbed graph, then we let the target models which has trained on the clean graph to test on the perturbed graph. We repeat the evaluation for 10 times on Cora, Citeseer, Computers, and Arxiv, and 5 times for Aminer and Reddit since model performs more stably on large graphs. Then we report mean test accuracy of the target models on the target nodes and omit the variance due to the space limit.

### H.5.3    ATTACKS SETTING

By default, we use PGD (Madry et al., 2018) to generate malicious node features. The learning step is 0.01 and the default training epoch is 500. We also employ the early stop of 100 epochs according to the accuracy of the surrogate model on the target nodes. While for heuristic approaches such as TDGIA (Zou et al., 2021) and ATDGIA, we follow the setting of TDGIA to update the features. Empirically, we find the original TDGIA feature update suits better for heuristic approaches while they show no advance over PGD for other approaches. Besides, as Table 7 shows, MetaGIA requires huge amount of time originally. Thus, to scale up, we use a batch update which updates the injected edges by a step size of $b$, i.e., the maximum degree of injected nodes, and limit the overall update epochs by $|V_{\text{atk}}|/6$, where we empirically observe this setting performs best in Cora hence we stick it for the other datasets.

For the setting of $\lambda$ for HAO, we search the parameters within $0.5$ to $8$ by a step size of $0.5$ such that the setting of $\lambda$ will not degenerate the performance of the attacks on surrogate model. Besides heuristic approaches, we additionally use a hinge loss to stabilize the gradient information from $\mathcal{L}_{\text{atk}}$ and $C(\mathcal{G}, \mathcal{G}')$, where the former can be too large that blurs the optimization direction of the latter. Take Cross Entropy with $\log\_\text{softmax}$ as an example, we adopt the following to constrict the magnitude of $\mathcal{L}_{\text{atk}}$:

$$
\begin{aligned}
\mathcal{L}_{\text{atk}[u]} = &(-H_{[u,Y_u]}^{(k)}) \cdot \mathbf{1}\{\frac{\exp(H_{[u,Y_u]}^{(k)})}{\sum_i \exp(H_{[u,i]}^{(k)})} \geq \tau\} \\
&+ \log(\sum_i \exp(H_{[u,i]}^{(k)} \cdot \mathbf{1}\{\frac{\exp(H_{[u,i]}^{(k)})}{\sum_j \exp(H_{[u,j]}^{(k)})} \geq \tau\})),
\end{aligned}
\tag{63}
$$

where $\mathbf{1}\{\frac{\exp(H_{[u,Y_u]}^{(k)})}{\sum_i \exp(H_{[u,i]}^{(k)})} \geq \tau\}$ can be taken as the predicted probability for $Y_u = u$ and $\tau$ is the corresponding threshold for hinge loss that we set as $1 \times 10^{-8}$.

For the hyper-parameter setting of our proposed strategies in Sec. 4.2, we find directly adopting $\lambda$ in PGD for $\lambda_X$ and setting $\lambda_A = 0$ performs empirically better. Hence we stick to the setting for $\lambda_A$ and $\lambda_X$. For the weight of sparsity regularization term in AGIA, we directly adopt $1/b$. For the hyper-parameters in heuristic methods, we directly follow TDGIA (Zou et al., 2021). For SeqGIA, we set $\gamma_{\text{atk}}$ be $\min(0.2, \lfloor |V_c|/2b \rfloor)$ and $\gamma_c = \min(|V_c|, \gamma_{\text{atk}}|V_{\text{atk}}|b)$ by default.

### H.6    SOFTWARE AND HARDWARE

We implement our methods with PyTorch (Paszke et al., 2019) and PyTorch Geometric (Fey & Lenssen, 2019). We ran our experiments on Linux Servers with 40 cores Intel(R) Xeon(R) Silver 4114 CPU @ 2.20GHz, 256 GB Memory, and Ubuntu 18.04 LTS installed. One has 4 NVIDIA RTX 2080Ti graphics cards with CUDA 10.2 and the other has 2 NVIDIA RTX 2080Ti and 2 NVIDIA RTX 3090Ti graphics cards with CUDA 11.3.

## I    MORE EXPERIMENTAL RESULTS

In this section, we provide more results from experiments about HAO to further validate its effectiveness. Specifically, we provide full results of averaged attack performance across all defense models, as well as initial experiments of HAO on two disassortative graphs.

Table 9: Full averaged performance across all defense models

| Model | Cora[†] | Citeseer[†] | Computers[†] | Arxiv[†] | Arxiv[‡] | Computers[‡] | Aminer[‡] | Reddit[‡] |
|---|---|---|---|---|---|---|---|---|
| Clean | 84.74 | 74.10 | 92.25 | 70.44 | 70.44 | 91.68 | 62.39 | 95.51 |
| PGD | 61.09 | 54.08 | 61.75 | 54.23 | 36.70 | 62.41 | 26.13 | 62.72 |
| +HAO | 56.63 | 48.12 | 59.16 | 45.05 | 28.48 | 59.09 | 22.15 | 56.99 |
| MetaGIA | 60.56 | 53.72 | 61.75 | 53.69 | 28.78 | 62.08 | 32.78 | 60.14 |
| +HAO | 58.51 | 47.44 | 60.29 | 48.48 | 24.61 | 58.63 | 29.91 | **54.14** |
| AGIA | 60.10 | 54.55 | 60.66 | 48.86 | 32.68 | 61.98 | 31.06 | 59.96 |
| +HAO | **53.79** | 48.30 | **58.71** | 48.86 | 29.52 | **58.37** | 26.51 | 56.36 |
| TDGIA | 66.86 | 52.45 | 66.79 | 49.73 | 31.68 | 62.47 | 32.37 | 57.97 |
| +HAO | 65.22 | 46.61 | 65.46 | 49.54 | **22.04** | 59.67 | 22.32 | 54.32 |
| ATDGIA | 61.14 | 49.46 | 65.07 | 46.53 | 32.08 | 64.66 | 24.72 | 61.25 |
| +HAO | 58.13 | **43.41** | 63.31 | **44.40** | 29.24 | 59.27 | **17.62** | 56.90 |

The lower is better. [†]Non-targeted attack. [‡]Targeted attack.

## I.1 FULL RESULTS OF AVERAGED ATTACK PERFORMANCE

In this section, we provide full results of averaged attack performance across all defense models, as a supplementary for Table 3b.

## I.2 MORE RESULTS ON DISASSORTATIVE GRAPHS

In this section, we provide initial investigation into the non-targeted attack performances of various GIA methods with or without HAO on disassortative graphs. Specifically, we select Chameleon and Squirrel provided by Pei et al. (2020). Statistics and budgets used for attack are given in Table 10 and Table 11.

Table 10: Statistics of the disassortative datasets

| Datasets | Nodes | Edges | Classes | Avg. Degree | Label Homophily | Avg. Homophily |
|---|---|---|---|---|---|---|
| Chameleon | 2277 | 31, 421 | 5 | 27.60 | 0.26 | 0.62 |
| Squirrel | 5201 | 198, 493 | 5 | 76.33 | 0.23 | 0.58 |

We also heuristically specify the budgets for each dataset according the the number of target nodes and average degrees.

Table 11: Budgets for non-targeted attacks on disassortative datasets

| Datasets | Nodes | Degree | Node Per.(%) | Edge Per.(%) |
|---|---|---|---|---|
| Chameleon | 60 | 100 | 2.64% | 19.10% |
| Squirrel | 90 | 50 | 1.73% | 2.27% |

For the settings of hyperparameters in attack methods and evaluation, we basically follow the same setup as given in Appendix H.5. In particular, we find using a threshold of $0.05$ for homophily defenders work best on Chameleon. Besides, we also observe robust tricks can not always improve performances of GNNs on these graphs. For example, we observe that using a large step-size of FLAG may degenerate the performances of GNNs on these datasets, hence we use a smaller step-size of $5 \times 10^{-4}$ as well as a small number of steps of 10. Moreover, using a LN before the first GNN layer may also hurt the performance. For fair comparison, we remove these results from defenses. Finally, in Table 12, we report both categorized defense results as Table 1 as well as the averaged attack performance as Table 3b.

From the results, we observe that, although our methods are not initially designed for disassortative graphs, HAO still brings empirical improvements. Specifically, on Chameleon, HAO improves the attack performance up to $25\%$ against homophily defenders, up to $12\%$ against robust models, up to $10\%$ against extreme robust models, and finally brings up to $3\%$ averaged test robustness of all models. While on Squirrel, the improvements become relatively low while still non-trivial. For

Table 12: Results of non-targeted attacks on disassortative graphs

| | HAO | Chameleon ($\downarrow$) | | | | Squirrel($\downarrow$) | | | |
|---|---|---|---|---|---|---|---|---|---|
| | | Homo | Robust | Combo | AVG. | Homo | Robust | Combo | AVG. |
| Clean | | 61.89 | 65.18 | 64.92 | 62.58 | 37.33 | 43.88 | 45.87 | 40.04 |
| PGD | | 61.89 | 61.89 | 63.61 | 33.24 | 35.66 | 36.28 | 40.54 | 26.03 |
| PGD | ✓ | 52.78 | 57.87 | 59.31 | 38.00 | 33.32 | 39.36 | 35.83 | 26.37 |
| MetaGIA[†] | | 61.89 | 61.89 | 63.61 | 34.38 | 35.66 | **35.66** | 39.40 | 26.09 |
| MetaGIA[†] | ✓ | 49.25 | 55.83 | 55.73 | 33.63 | 34.07 | 38.26 | **35.24** | **25.81** |
| AGIA[†] | | 61.89 | 61.89 | 63.61 | 35.95 | 35.66 | 35.89 | 39.93 | 26.93 |
| AGIA[†] | ✓ | 43.98 | **48.88** | **53.33** | **32.03** | 35.69 | 36.31 | 36.40 | 26.77 |
| TDGIA | | 61.95 | 61.95 | 63.76 | 41.17 | 35.66 | **35.66** | 40.81 | 29.02 |
| TDGIA | ✓ | 46.36 | 51.12 | 55.14 | 38.90 | **31.51** | 38.21 | 35.63 | 28.65 |
| ATDGIA | | 61.95 | 61.95 | 63.76 | 41.11 | 35.66 | **35.66** | 41.62 | 29.62 |
| ATDGIA | ✓ | **36.93** | 57.75 | 59.25 | 38.88 | 32.02 | 40.00 | 40.62 | 30.24 |
| MLP | | | 50.15 | | | | 32.51 | | |

$\downarrow$The lower number indicates better attack performance. [†]Runs with SeqGIA framework on Computers and Arxiv.

Table 13: Detailed results of non-targeted attacks on Cora (1)

| | EGuard+LNi+FLAG+LN | EGuard+FLAG+LN | EGuard+LNi+FLAG | Guard+LNi+LN | RGAT+FLAG+LN | GCN+LNi+FLAG+LN | RobustGCN+FLAG | RGAT+LN | Guard+LN | EGuard+FLAG |
|---|---|---|---|---|---|---|---|---|---|---|
| Clean | 83.48 | 84.17 | 85.9 | 79.56 | 87.29 | 86.37 | 86.21 | 85.29 | 81.72 | 85.56 |
| PGD | 82.53 | 83.94 | 85.74 | 79.74 | 76.78 | 71.10 | 69.57 | 79.56 | 81.44 | 85.35 |
| +HAO | 77.99 | 73.04 | 66.25 | 74.21 | 68.09 | 71.06 | 70.3 | 67.92 | 68.12 | 53.99 |
| MetaGIA | 82.68 | 83.96 | 85.86 | 79.51 | 75.18 | 69.72 | 69.4 | 78.04 | 81.59 | 85.48 |
| +HAO | 69.49 | 65.92 | 66.83 | 63.02 | 66.38 | 71.86 | 76.8 | 57.75 | 55.35 | 56.77 |
| AGIA | 82.75 | 83.69 | 85.78 | 79.56 | 75.77 | 69.25 | 69.10 | 79.10 | 81.43 | 85.34 |
| +HAO | 75.25 | 69.10 | 61.00 | 70.12 | 65.48 | 69.86 | 71.08 | 62.76 | 60.96 | 48.54 |
| TDGIA | 83.13 | 83.65 | 85.72 | 79.13 | 82.37 | 79.31 | 76.11 | 82.2 | 81.37 | 85.39 |
| +HAO | 77.93 | 73.58 | 75.47 | 73.67 | 75.18 | 79.45 | 78.63 | 69.58 | 64.66 | 65.31 |
| ATDIGA | 82.57 | 83.54 | 85.39 | 79.38 | 78.76 | 76.09 | 73.08 | 79.8 | 81.47 | 84.88 |
| +HAO | 74.43 | 71.88 | 71.21 | 66.97 | 72.51 | 76.87 | 76.17 | 60.61 | 62.38 | 63.53 |
| **AVG** | 79.29 | 77.86 | 77.74 | 74.99 | 74.89 | 74.63 | 74.22 | 72.96 | 72.77 | 72.74 |

example, HAO improves the attack performance up to $4\%$ in terms of test robustness against homophily defenders. We hypothesize the reason why HAO also works on disassortative graphs is because GNN can still learn the homophily information implicitly, e.g., similarity between class label distributions (Ma et al., 2022), which we will leave the in-depth analyses to future work.

## J DETAILED RESULTS OF ATTACK PERFORMANCE

### J.1 DETAILED RESULTS OF NON-TARGETED ATTACKS

In this section, we present the detailed non-targeted attack results of the methods and datasets used in our experiments for Table 1. For simplicity, we only give the results of top 20 robust models according to the averaged test accuracy against all attacks.

### J.2 DETAILED RESULTS OF TARGETED ATTACKS

In this section, we present the detailed targeted attack results of the methods and datasets used in our experiments for Table 2. For simiplicity, we only give the results of top 20 robust models according to the averaged test accuracy against all attacks.

Table 14: Detailed results of non-targeted attacks on Cora (2)

|  | RGAT+FLAG | Guard+LNi | RobustGCN | GCN+FLAG+LN | GCN+LNi+FLAG | RGAT | GAT+LNi+FLAG+LN | Sage+LNi+FLAG+LN | Guard | GCN+LNi+LN |
|---|---|---|---|---|---|---|---|---|---|---|
| Clean | 87.21 | 83.18 | 84.63 | 85.86 | 86.36 | 85.74 | 86.55 | 84.95 | 83.61 | 84.47 |
| PGD | 76.93 | 83.11 | 63.20 | 62.55 | 60.68 | 79.28 | 61.29 | 61.84 | 83.08 | 58.46 |
| +HAO | 62.35 | 53.68 | 62.60 | 63.60 | 61.69 | 52.60 | 62.81 | 62.34 | 44.02 | 58.78 |
| MetaGIA | 75.14 | 83.08 | 63.53 | 59.18 | 60.36 | 77.97 | 57.88 | 61.01 | 83.61 | 58.10 |
| +HAO | 61.53 | 57.31 | 69.83 | 67.00 | 66.64 | 49.25 | 65.82 | 65.69 | 45.41 | 61.94 |
| AGIA | 76.04 | 83.08 | 62.67 | 61.26 | 59.09 | 78.95 | 57.84 | 58.61 | 83.44 | 57.05 |
| +HAO | 57.17 | 49.12 | 61.59 | 62.65 | 59.25 | 47.24 | 59.80 | 59.56 | 39.87 | 55.62 |
| TDGIA | 82.02 | 83.04 | 71.34 | 71.35 | 73.47 | 81.79 | 71.52 | 70.30 | 83.44 | 70.69 |
| +HAO | 70.52 | 67.04 | 73.38 | 73.52 | 75.00 | 56.95 | 71.96 | 71.56 | 50.79 | 72.90 |
| ATDIGA | 79.06 | 82.85 | 66.96 | 69.61 | 65.89 | 79.91 | 65.57 | 63.81 | 83.07 | 62.95 |
| +HAO | 64.50 | 55.13 | 70.30 | 72.46 | 70.94 | 42.18 | 69.26 | 67.59 | 40.46 | 65.53 |
| **AVG** | 72.04 | 70.97 | 68.18 | 68.09 | 67.22 | 66.53 | 66.39 | 66.11 | 65.53 | 64.23 |

Table 15: Detailed results of non-targeted attacks on Citeseer (1)

|  | RGAT+LN | RGAT+FLAG+LN | EGuard+LNi+FLAG+LN | Guard+LNi+LN | RGAT | EGuard+FLAG+LN | RGAT+FLAG | EGuard+LNi+FLAG | EGuard+FLAG | GCN+LNi+FLAG+LN |
|---|---|---|---|---|---|---|---|---|---|---|
| Clean | 74.82 | 75.72 | 75.44 | 74.25 | 74.85 | 73.64 | 75.56 | 74.75 | 73.57 | 75.67 |
| PGD | 71.00 | 71.32 | 75.19 | 74.21 | 69.33 | 73.55 | 69.84 | 74.83 | 73.57 | 57.97 |
| +HAO | 71.00 | 70.82 | 66.07 | 73.04 | 69.05 | 61.55 | 65.78 | 50.01 | 47.54 | 58.77 |
| MetaGIA | 70.32 | 70.21 | 75.15 | 74.21 | 68.42 | 73.55 | 68.90 | 74.83 | 73.57 | 56.36 |
| +HAO | 70.37 | 69.77 | 64.00 | 71.25 | 68.04 | 59.94 | 63.10 | 49.70 | 46.95 | 57.17 |
| AGIA | 71.45 | 70.51 | 75.29 | 74.21 | 70.31 | 73.60 | 69.40 | 74.83 | 73.61 | 56.50 |
| +HAO | 71.80 | 70.70 | 64.54 | 70.58 | 70.24 | 59.32 | 62.31 | 50.33 | 46.77 | 58.02 |
| TDGIA | 72.29 | 73.81 | 75.26 | 74.21 | 70.99 | 73.55 | 73.34 | 74.85 | 73.57 | 63.01 |
| +HAO | 72.51 | 70.18 | 68.04 | 56.69 | 60.91 | 65.70 | 53.99 | 56.73 | 52.86 | 66.52 |
| ATDIGA | 72.23 | 72.82 | 75.12 | 74.21 | 70.61 | 73.55 | 72.37 | 74.82 | 73.54 | 61.55 |
| +HAO | 71.22 | 69.63 | 65.82 | 52.97 | 61.08 | 64.51 | 53.76 | 52.94 | 51.20 | 64.04 |
| **AVG** | 71.73 | 71.41 | 70.90 | 69.98 | 68.53 | 68.41 | 66.21 | 64.42 | 62.43 | 61.42 |

Table 16: Detailed results of non-targeted attacks on Citeseer (2)

|  | Guard+LN | Guard+LNi | RobustGCN+FLAG | Guard | GCN+LNi+FLAG | Sage+LNi+FLAG+LN | GAT+LNi+FLAG+LN | RobustGCN | GCN+LNi+LN | Sage+LNi+FLAG |
|---|---|---|---|---|---|---|---|---|---|---|
| Clean | 73.97 | 74.41 | 75.87 | 74.78 | 75.45 | 73.89 | 75.60 | 75.46 | 74.65 | 73.70 |
| PGD | 74.07 | 74.28 | 53.81 | 74.70 | 47.56 | 46.82 | 45.00 | 39.77 | 40.69 | 40.11 |
| +HAO | 48.48 | 38.91 | 51.10 | 33.83 | 49.19 | 46.93 | 44.06 | 39.72 | 40.79 | 40.88 |
| MetaGIA | 74.07 | 74.28 | 53.11 | 74.70 | 47.14 | 46.13 | 44.76 | 39.84 | 40.87 | 40.13 |
| +HAO | 45.32 | 38.98 | 50.85 | 33.95 | 49.03 | 46.42 | 44.08 | 39.79 | 41.02 | 40.90 |
| AGIA | 74.07 | 74.29 | 53.12 | 74.72 | 47.30 | 46.29 | 44.07 | 40.16 | 41.76 | 40.73 |
| +HAO | 43.47 | 41.04 | 50.88 | 36.51 | 49.61 | 47.28 | 45.66 | 41.53 | 42.32 | 42.82 |
| TDGIA | 74.07 | 74.28 | 55.01 | 74.76 | 49.47 | 47.06 | 41.08 | 37.94 | 40.68 | 36.21 |
| +HAO | 36.83 | 36.50 | 60.37 | 26.45 | 57.45 | 49.82 | 49.74 | 47.44 | 43.85 | 40.83 |
| ATDIGA | 74.07 | 74.21 | 54.95 | 74.72 | 45.09 | 41.89 | 36.24 | 34.65 | 32.10 | 31.17 |
| +HAO | 30.21 | 28.74 | 55.40 | 21.70 | 52.22 | 45.66 | 45.19 | 40.35 | 35.05 | 38.81 |
| **AVG** | 58.97 | 57.27 | 55.86 | 54.62 | 51.77 | 48.93 | 46.86 | 43.33 | 43.07 | 42.39 |

Table 17: Detailed results of non-targeted attacks on Computers (1)

| | EGuard+LNi+FLAG+LN | Guard+LNi+LN | EGuard+FLAG+LN | Guard+LN | RGAT+FLAG+LN | RGAT+FLAG | EGuard+LNi+FLAG | Guard+LNi | RGAT+LN | RGAT |
|---|---|---|---|---|---|---|---|---|---|---|
| Clean | 91.04 | 90.88 | 91.40 | 91.23 | 93.21 | 93.32 | 92.16 | 91.95 | 93.20 | 93.17 |
| PGD | 90.94 | 90.87 | 91.41 | 91.24 | 81.59 | 80.19 | 88.24 | 87.93 | 79.68 | 79.05 |
| +HAO | 87.83 | 87.59 | 80.41 | 75.94 | 81.80 | 82.26 | 64.18 | 62.69 | 79.29 | 79.33 |
| MetaGIA | 90.94 | 90.87 | 91.41 | 91.24 | 81.58 | 80.18 | 88.23 | 87.91 | 79.68 | 79.06 |
| +HAO | 90.25 | 90.21 | 90.11 | 88.32 | 81.64 | 81.72 | 78.11 | 76.58 | 79.29 | 78.96 |
| AGIA | 90.98 | 90.90 | 91.40 | 91.22 | 78.09 | 76.59 | 88.25 | 87.86 | 76.62 | 75.56 |
| +HAO | 86.02 | 85.77 | 75.97 | 71.49 | 77.55 | 78.17 | 63.96 | 62.74 | 75.23 | 75.14 |
| TDGIA | 90.97 | 90.91 | 91.40 | 91.24 | 77.07 | 75.40 | 90.26 | 89.94 | 75.94 | 74.66 |
| +HAO | 90.42 | 90.34 | 90.35 | 89.00 | 77.12 | 76.61 | 74.58 | 74.22 | 75.71 | 74.77 |
| ATDIGA | 90.97 | 90.90 | 91.41 | 91.24 | 82.42 | 81.77 | 89.24 | 88.84 | 81.29 | 80.76 |
| +HAO | 84.60 | 83.93 | 74.38 | 69.33 | 82.97 | 83.50 | 69.92 | 68.50 | 80.92 | 80.86 |
| **AVG** | 89.54 | 89.38 | 87.24 | 85.59 | 81.37 | 80.88 | 80.65 | 79.92 | 79.71 | 79.21 |

Table 18: Detailed results of non-targeted attacks on Computers (2)

| | GAT+FLAG+LN | EGuard+FLAG | Guard | RobustGCN+FLAG | GAT+LNi+FLAG+LN | RobustGCN | Sage+LNi+FLAG+LN | GAT+LNi+FLAG | GCN+LNi+FLAG+LN | GAT+LNi+LN |
|---|---|---|---|---|---|---|---|---|---|---|
| Clean | 92.17 | 91.68 | 91.55 | 92.46 | 92.40 | 92.24 | 91.71 | 92.45 | 93.22 | 92.05 |
| PGD | 82.31 | 85.82 | 84.91 | 73.27 | 77.91 | 67.14 | 63.83 | 67.61 | 54.96 | 52.20 |
| +HAO | 69.83 | 55.62 | 54.31 | 72.73 | 65.08 | 68.80 | 62.55 | 54.93 | 63.28 | 69.19 |
| MetaGIA | 82.31 | 85.81 | 84.91 | 73.28 | 77.91 | 67.14 | 63.83 | 67.62 | 54.96 | 52.21 |
| +HAO | 77.39 | 69.73 | 67.90 | 70.42 | 69.52 | 64.76 | 62.45 | 58.24 | 59.31 | 63.69 |
| AGIA | 79.60 | 86.08 | 85.21 | 71.95 | 75.01 | 66.01 | 60.72 | 64.25 | 52.34 | 50.69 |
| +HAO | 63.02 | 56.48 | 55.35 | 72.18 | 61.22 | 68.84 | 60.68 | 53.95 | 62.78 | 67.54 |
| TDGIA | 80.42 | 88.64 | 88.32 | 72.23 | 75.27 | 69.45 | 63.87 | 68.58 | 64.96 | 58.98 |
| +HAO | 79.19 | 69.75 | 68.76 | 71.39 | 70.84 | 69.11 | 63.72 | 63.45 | 66.56 | 65.81 |
| ATDIGA | 82.42 | 87.11 | 86.03 | 76.96 | 79.13 | 71.92 | 68.42 | 71.15 | 66.01 | 53.34 |
| +HAO | 60.74 | 61.46 | 58.81 | 76.79 | 64.38 | 74.26 | 68.33 | 57.90 | 72.34 | 73.82 |
| **AVG** | 77.22 | 76.20 | 75.10 | 74.88 | 73.52 | 70.88 | 66.37 | 65.47 | 64.61 | 63.59 |

Table 19: Detailed results of non-targeted attacks on Arxiv (1)

| | Guard+LNi+LN | RGAT+LN | RGAT+FLAG+LN | EGuard+LNi+FLAG+LN | EGuard+FLAG+LN | Guard+LN | RobustGCN+FLAG | RobustGCN | GCN+LNi+FLAG+LN | Guard+LNi |
|---|---|---|---|---|---|---|---|---|---|---|
| | 71.15 | 70.95 | 70.84 | 69.50 | 69.46 | 69.76 | 67.85 | 67.50 | 71.40 | 70.99 |
| PGD | 71.11 | 66.57 | 66.61 | 69.28 | 69.24 | 69.62 | 60.60 | 60.81 | 55.99 | 70.26 |
| | 68.68 | 66.68 | 66.60 | 61.05 | 61.02 | 58.92 | 62.99 | 62.89 | 60.02 | 47.84 |
| MetaGIA | 71.09 | 67.87 | 67.67 | 69.23 | 69.22 | 69.59 | 64.10 | 64.10 | 63.58 | 70.40 |
| | 69.97 | 66.81 | 66.52 | 66.14 | 66.13 | 65.70 | 63.20 | 63.30 | 64.13 | 58.58 |
| AGIA | 70.97 | 65.22 | 64.46 | 68.23 | 68.17 | 68.57 | 59.26 | 59.23 | 57.26 | 64.60 |
| | 63.57 | 57.02 | 56.60 | 58.27 | 58.20 | 57.73 | 60.77 | 60.72 | 61.50 | 58.08 |
| TDIGA | 71.02 | 67.54 | 67.28 | 68.37 | 68.33 | 68.72 | 63.70 | 63.56 | 61.01 | 65.63 |
| | 64.31 | 61.61 | 60.99 | 59.73 | 59.74 | 58.33 | 63.08 | 63.30 | 62.81 | 53.04 |
| ATDGIA | 71.01 | 68.49 | 68.45 | 68.18 | 68.14 | 68.49 | 64.95 | 64.88 | 63.95 | 66.39 |
| | 69.92 | 68.67 | 68.58 | 66.34 | 66.35 | 65.47 | 65.56 | 65.62 | 65.83 | 55.42 |
| **AVG** | 69.34 | 66.13 | 65.87 | 65.85 | 65.82 | 65.54 | 63.28 | 63.26 | 62.50 | 61.93 |

Table 20: Detailed results of non-targeted attacks on Arxiv (2)

| | RGAT+FLAG | GCN+LNi+LN | RGAT | GCN+FLAG+LN | GAT+FLAG+LN | EGuard+LNi+FLAG | GCN+LN | EGuard+FLAG | GCN+LNi+FLAG | GAT+LNi+FLAG+LN |
|---|---|---|---|---|---|---|---|---|---|---|
| | 70.63 | 71.38 | 70.77 | 70.00 | 70.28 | 69.37 | 70.42 | 69.34 | 71.31 | 71.00 |
| PGD | 66.49 | 54.46 | 66.26 | 54.21 | 57.44 | 68.04 | 51.97 | 68.03 | 48.00 | 57.65 |
| | 57.18 | 58.40 | 55.38 | 55.51 | 59.16 | 37.02 | 52.45 | 36.80 | 52.75 | 53.97 |
| MetaGIA | 67.42 | 62.88 | 67.68 | 58.54 | 61.92 | 68.48 | 57.04 | 68.40 | 55.73 | 61.56 |
| | 58.21 | 63.35 | 57.05 | 59.65 | 51.65 | 50.32 | 57.39 | 50.23 | 57.72 | 54.63 |
| AGIA | 63.75 | 57.12 | 64.49 | 49.55 | 45.96 | 59.35 | 48.54 | 59.25 | 54.55 | 49.14 |
| | 50.31 | 61.29 | 49.36 | 58.25 | 49.71 | 49.24 | 57.24 | 49.20 | 58.10 | 48.78 |
| TDGIA | 66.74 | 58.91 | 66.95 | 55.47 | 56.30 | 62.18 | 52.39 | 62.10 | 48.86 | 52.58 |
| | 47.88 | 61.90 | 45.59 | 59.20 | 49.44 | 45.08 | 56.42 | 44.91 | 54.68 | 47.80 |
| ATDGIA | 67.97 | 62.21 | 68.07 | 58.61 | 63.36 | 62.73 | 55.26 | 62.67 | 54.19 | 58.50 |
| | 60.82 | 64.82 | 59.32 | 62.69 | 57.51 | 46.94 | 59.50 | 46.83 | 57.90 | 56.58 |
| **AVG** | 61.58 | 61.52 | 60.99 | 58.33 | 56.61 | 56.25 | 56.24 | 56.16 | 55.80 | 55.65 |

Table 21: Detailed results of targeted attacks on Computers (1)

| | EGuard+LNi+FLAG+LN | Guard+LNi+LN | EGuard+FLAG+LN | Guard+LN | Guard+LNi | EGuard+LNi+FLAG | RobustGCN+FLAG | RGAT+FLAG | RGAT+FLAG+LN | EGuard+FLAG |
|---|---|---|---|---|---|---|---|---|---|---|
| Clean | 90.96 | 90.76 | 91.56 | 91.11 | 91.12 | 91.29 | 91.85 | 92.83 | 92.78 | 90.75 |
| PGD | 90.96 | 90.76 | 91.56 | 91.11 | 89.38 | 89.54 | 72.36 | 72.17 | 74.28 | 88.36 |
| +HAO | 85.81 | 85.75 | 79.51 | 73.71 | 65.01 | 64.15 | 72.58 | 74.40 | 74.08 | 56.50 |
| MetaGIA | 90.96 | 90.76 | 91.56 | 91.11 | 88.93 | 89.10 | 73.81 | 70.58 | 72.24 | 88.10 |
| +HAO | 85.83 | 85.69 | 78.46 | 72.61 | 65.62 | 65.53 | 73.50 | 72.10 | 72.00 | 56.12 |
| AGIA | 91.00 | 90.82 | 91.58 | 91.06 | 89.11 | 89.33 | 72.96 | 68.85 | 69.64 | 88.00 |
| +HAO | 85.72 | 85.71 | 79.50 | 74.28 | 64.71 | 63.90 | 73.12 | 72.61 | 72.22 | 56.18 |
| TDGIA | 90.96 | 90.76 | 91.56 | 91.11 | 89.15 | 89.36 | 72.06 | 72.42 | 72.58 | 87.75 |
| +HAO | 77.15 | 75.64 | 65.21 | 62.97 | 69.78 | 70.43 | 73.08 | 74.33 | 74.00 | 64.31 |
| ATDGIA | 90.96 | 90.76 | 91.56 | 91.11 | 88.99 | 89.22 | 75.15 | 75.68 | 73.32 | 88.43 |
| +HAO | 78.35 | 77.67 | 62.87 | 59.65 | 63.75 | 63.15 | 74.06 | 75.78 | 74.14 | 56.51 |
| **AVG** | 87.15 | 86.83 | 83.18 | 80.89 | 78.69 | 78.64 | 74.96 | 74.70 | 74.66 | 74.64 |

Table 22: Detailed results of targeted attacks on Computers (2)

| | Guard | RobustGCN | RGAT | RGAT+LN | GAT+FLAG+LN | GAT+LNi+FLAG+LN | GAT+LNi+FLAG | GCN+LNi+FLAG+LN | GAT+LNi+LN | GCN+LNi+FLAG |
|---|---|---|---|---|---|---|---|---|---|---|
| Clean | 90.50 | 92.07 | 92.68 | 92.76 | 91.07 | 91.90 | 91.92 | 92.25 | 91.56 | 92.35 |
| PGD | 88.13 | 70.40 | 71.85 | 72.65 | 77.69 | 75.25 | 72.57 | 63.08 | 58.46 | 60.79 |
| +HAO | 54.96 | 70.76 | 71.78 | 71.40 | 71.03 | 66.01 | 62.46 | 66.01 | 70.49 | 64.17 |
| MetaGIA | 87.67 | 71.78 | 70.44 | 71.33 | 74.93 | 73.12 | 70.89 | 62.54 | 57.40 | 60.71 |
| +HAO | 55.00 | 71.61 | 70.21 | 70.35 | 69.56 | 64.82 | 62.58 | 64.81 | 67.57 | 63.04 |
| AGIA | 87.57 | 70.92 | 68.36 | 68.58 | 73.00 | 71.03 | 68.50 | 61.08 | 56.62 | 59.26 |
| +HAO | 54.89 | 71.58 | 69.96 | 69.99 | 68.44 | 64.81 | 61.00 | 64.68 | 69.39 | 62.28 |
| TDGIA | 87.21 | 69.86 | 71.54 | 71.28 | 74.24 | 72.86 | 70.60 | 62.74 | 57.54 | 60.35 |
| +HAO | 61.62 | 71.62 | 71.39 | 71.92 | 54.19 | 60.51 | 66.69 | 66.79 | 66.74 | 63.97 |
| ATDGIA | 87.85 | 73.33 | 74.39 | 72.19 | 73.36 | 75.24 | 74.06 | 65.14 | 56.22 | 62.67 |
| +HAO | 54.93 | 72.53 | 72.00 | 71.49 | 62.03 | 63.19 | 62.14 | 68.50 | 73.15 | 66.06 |
| **AVG** | 73.67 | 73.31 | 73.15 | 73.09 | 71.78 | 70.79 | 69.40 | 67.06 | 65.92 | 65.06 |

Table 23: Detailed results of targeted attacks on Arxiv (1)

|  | Guard+LNi+LN | EGuard+LNi+FLAG+LN | Guard+LNi | EGuard+FLAG+LN | EGuard+LNi+FLAG | Guard+LN | EGuard+FLAG | Guard | RobustGCN+FLAG | RGAT |
|---|---|---|---|---|---|---|---|---|---|---|
|  | 71.34 | 71.22 | 71.22 | 69.59 | 70.59 | 69.78 | 68.88 | 69.41 | 67.28 | 67.03 |
| PGD | 71.31 | 71.16 | 71.16 | 69.47 | 70.47 | 69.69 | 68.69 | 69.19 | 39.91 | 39.13 |
|  | 69.38 | 65.69 | 33.78 | 47.41 | 29.12 | 38.00 | 14.31 | 13.94 | 36.12 | 36.06 |
| MetaGIA | 71.03 | 71.22 | 70.53 | 69.59 | 70.59 | 69.78 | 68.84 | 69.28 | 42.56 | 41.81 |
|  | 42.56 | 48.06 | 33.94 | 31.84 | 34.94 | 26.75 | 20.34 | 18.28 | 38.66 | 38.44 |
| AGIA | 71.06 | 70.94 | 70.19 | 69.25 | 67.72 | 69.38 | 64.38 | 63.66 | 39.94 | 39.47 |
|  | 38.56 | 37.22 | 35.06 | 24.63 | 35.31 | 22.09 | 16.19 | 14.09 | 42.53 | 42.56 |
| TDIGA | 71.00 | 71.16 | 69.78 | 68.97 | 68.22 | 69.41 | 66.09 | 66.12 | 41.25 | 41.31 |
|  | 38.72 | 34.19 | 38.78 | 23.41 | 33.94 | 20.78 | 17.66 | 16.06 | 38.38 | 38.28 |
| ATDGIA | 71.06 | 70.88 | 70.56 | 69.19 | 69.03 | 69.56 | 66.09 | 66.19 | 44.06 | 43.75 |
|  | 68.97 | 61.03 | 37.88 | 41.69 | 33.69 | 34.25 | 19.16 | 17.28 | 39.03 | 38.84 |
| **AVG** | 62.27 | 61.16 | 54.81 | 53.19 | 53.06 | 50.86 | 44.60 | 43.95 | 42.70 | 42.43 |

Table 24: Detailed results of targeted attacks on Arxiv (2)

|  | RobustGCN | RGAT+LN | RGAT+FLAG+LN | GCN+LNi+FLAG | RGAT+FLAG | GCN+LNi+LN | GCN+LNi | GCN+LNi+FLAG+LN | GAT+LN | GAT+FLAG+LN |
|---|---|---|---|---|---|---|---|---|---|---|
|  | 67.69 | 72.06 | 71.41 | 71.34 | 71.16 | 71.97 | 71.59 | 71.75 | 69.94 | 69.94 |
| PGD | 38.66 | 40.31 | 38.06 | 32.19 | 37.78 | 29.09 | 29.97 | 29.72 | 36.34 | 38.84 |
|  | 37.22 | 37.06 | 34.28 | 32.75 | 23.69 | 28.91 | 29.56 | 29.28 | 28.88 | 30.47 |
| MetaGIA | 35.00 | 42.56 | 41.28 | 30.28 | 41.03 | 28.91 | 28.59 | 28.50 | 16.00 | 14.84 |
|  | 33.22 | 34.09 | 32.53 | 30.03 | 27.81 | 27.50 | 27.97 | 27.47 | 19.44 | 21.50 |
| AGIA | 41.06 | 42.12 | 42.06 | 32.53 | 39.75 | 33.09 | 32.56 | 31.84 | 23.84 | 21.12 |
|  | 41.97 | 23.84 | 23.66 | 35.19 | 23.03 | 34.03 | 34.47 | 34.25 | 16.97 | 14.94 |
| TDIGA | 44.28 | 43.84 | 43.91 | 36.31 | 42.12 | 36.34 | 35.12 | 36.16 | 27.38 | 24.50 |
|  | 40.81 | 32.38 | 31.50 | 39.47 | 28.31 | 38.50 | 38.62 | 37.91 | 27.56 | 29.28 |
| ATDGIA | 43.12 | 44.34 | 44.22 | 34.47 | 41.91 | 33.53 | 33.44 | 33.28 | 31.06 | 24.19 |
|  | 37.97 | 39.00 | 37.84 | 33.84 | 30.19 | 30.53 | 30.47 | 30.59 | 30.28 | 33.69 |
| **AVG** | 41.91 | 41.05 | 40.07 | 37.13 | 36.98 | 35.67 | 35.67 | 35.52 | 29.79 | 29.39 |

Table 25: Detailed results of targeted attacks on Aminer (1)

|  | EGuard+LNi+FLAG | EGuard+LNi+FLAG+LN | Guard+LNi | Guard+LNi+LN | EGuard+FLAG | Guard | RGAT+FLAG | Guard+LN | EGuard+FLAG+LN | RGAT |
|---|---|---|---|---|---|---|---|---|---|---|
|  | 59.03 | 58.06 | 60.72 | 60.85 | 57.06 | 57.25 | 61.75 | 58.50 | 58.81 | 62.78 |
| PGD | 55.25 | 48.47 | 56.31 | 49.40 | 53.03 | 53.16 | 41.84 | 49.72 | 48.31 | 40.72 |
|  | 39.06 | 39.47 | 37.03 | 39.40 | 35.16 | 34.62 | 33.53 | 29.69 | 29.97 | 31.75 |
| MetaGIA | 52.09 | 50.66 | 52.35 | 49.81 | 49.03 | 48.97 | 46.19 | 48.34 | 47.59 | 45.81 |
|  | 42.09 | 45.16 | 40.26 | 43.42 | 37.00 | 37.09 | 41.47 | 36.88 | 36.62 | 41.12 |
| AGIA | 54.06 | 48.00 | 54.82 | 48.17 | 51.28 | 51.34 | 48.72 | 48.78 | 47.59 | 48.25 |
|  | 26.44 | 29.94 | 23.25 | 28.08 | 19.84 | 18.97 | 26.50 | 23.19 | 24.06 | 25.78 |
| TDIGA | 52.75 | 46.72 | 53.68 | 46.92 | 50.75 | 50.87 | 42.50 | 47.66 | 46.28 | 40.81 |
|  | 24.31 | 28.91 | 18.54 | 26.07 | 16.12 | 15.06 | 24.00 | 19.69 | 20.66 | 22.5 |
| ATDGIA | 53.44 | 51.00 | 53.69 | 49.32 | 50.34 | 50.50 | 45.44 | 49.97 | 49.59 | 45.25 |
|  | 38.19 | 42.66 | 35.93 | 41.07 | 33.72 | 33.72 | 36.72 | 31.91 | 31.69 | 35.94 |
| **AVG** | 45.16 | 44.46 | 44.23 | 43.86 | 41.21 | 41.05 | 40.79 | 40.39 | 40.11 | 40.06 |

Table 26: Detailed results of targeted attacks on Aminer (2)

|  | RGAT+FLAG+LN | GCN+LNi+FLAG+LN | RGAT+LN | Sage+LNi+FLAG+LN | GCN+LNi+FLAG | GCN+LNi+LN | Sage+LNi+LN | GAT+LNi+LN | GAT+LNi+FLAG+LN | Sage+LNi+FLAG |
|---|---|---|---|---|---|---|---|---|---|---|
|  | 62.66 | 64.41 | 63.78 | 65.56 | 63.91 | 66.88 | 65.44 | 66.97 | 65.78 | 64.34 |
| PGD | 31.97 | 28.03 | 29.75 | 26.22 | 26.81 | 22.65 | 23.78 | 17.00 | 16.66 | 22.03 |
|  | 29.06 | 28.16 | 27.06 | 26.44 | 26.81 | 23.17 | 23.88 | 17.58 | 16.53 | 22.06 |
| MetaGIA | 41.38 | 41.12 | 40.78 | 37.56 | 36.72 | 38.17 | 36.56 | 38.40 | 37.31 | 31.25 |
|  | 39.62 | 42.16 | 38.03 | 37.38 | 36.03 | 37.89 | 36.03 | 37.60 | 37.31 | 31.12 |
| AGIA | 38.34 | 34.62 | 37.47 | 31.94 | 33.97 | 31.21 | 31.31 | 29.96 | 29.62 | 29.50 |
|  | 28.19 | 29.03 | 27.06 | 27.19 | 28.00 | 27.14 | 26.31 | 22.00 | 21.09 | 25.25 |
| TDIGA | 30.47 | 28.44 | 28.25 | 24.41 | 24.97 | 20.85 | 22.19 | 15.39 | 15.16 | 20.56 |
|  | 27.12 | 27.53 | 24.97 | 24.56 | 24.84 | 22.19 | 22.22 | 15.75 | 14.03 | 20.16 |
| ATDGIA | 39.28 | 36.62 | 38.03 | 33.38 | 32.09 | 32.44 | 32.47 | 34.83 | 35.12 | 27.62 |
|  | 32.66 | 37.72 | 31.50 | 31.87 | 31.78 | 32.00 | 30.72 | 33.97 | 33.06 | 26.12 |
| **AVG** | 36.43 | 36.17 | 35.15 | 33.32 | 33.27 | 32.24 | 31.90 | 29.95 | 29.24 | 29.09 |

Table 27: Detailed results of targeted attacks on Reddit (1)

|         | Guard+LNi+LN | RobustGCN | RobustGCN+FLAG | Guard+LNi | Guard+LN | EGuard+LNi+FLAG+LN | EGuard+FLAG+LN | Sage+LNi+FLAG+LN | Guard | EGuard+FLAG |
|---------|------|------|------|------|------|------|------|------|------|------|
|         | 94.47 | 95.08 | 95.30 | 94.42 | 94.61 | 94.61 | 94.60 | 97.10 | 94.05 | 94.08 |
| PGD     | 92.91 | 84.81 | 83.84 | 93.03 | 92.69 | 92.69 | 92.53 | 76.25 | 92.44 | 92.72 |
|         | 80.03 | 86.12 | 84.94 | 75.53 | 68.53 | 69.31 | 69.34 | 75.25 | 56.44 | 58.03 |
| MetaGIA | 93.53 | 88.25 | 87.22 | 93.28 | 93.38 | 93.66 | 93.59 | 80.72 | 92.40 | 92.88 |
|         | 77.47 | 90.06 | 90.44 | 69.91 | 65.28 | 68.00 | 68.34 | 83.62 | 46.75 | 48.59 |
| AGIA    | 93.62 | 86.09 | 87.84 | 92.84 | 93.16 | 92.78 | 92.69 | 81.59 | 92.19 | 91.31 |
|         | 88.66 | 85.06 | 87.84 | 85.34 | 83.09 | 77.06 | 77.31 | 72.19 | 78.16 | 67.06 |
| TDIGA   | 93.03 | 90.19 | 89.91 | 92.25 | 92.59 | 92.91 | 92.53 | 80.94 | 91.25 | 91.59 |
|         | 86.03 | 89.06 | 88.91 | 80.38 | 78.69 | 81.56 | 81.25 | 79.78 | 64.09 | 66.62 |
| ATDGIA  | 93.34 | 87.34 | 84.91 | 92.38 | 92.69 | 93.97 | 93.81 | 76.53 | 91.62 | 93.00 |
|         | 90.78 | 88.84 | 88.38 | 87.94 | 88.06 | 79.44 | 79.25 | 78.66 | 80.69 | 63.22 |
| **AVG** | 89.44 | 88.26 | 88.14 | 87.03 | 85.71 | 85.09 | 85.02 | 80.24 | 80.01 | 78.10 |

Table 28: Detailed results of targeted attacks on Reddit (2)

|         | EGuard+LNi+FLAG | Sage+LNi+LN | GAT+LNi+FLAG+LN | Sage+FLAG+LN | Sage+LN | GAT+LNi+LN | GAT+FLAG+LN | GAT+LN | Sage+LNi+FLAG | GCN+LNi+FLAG+LN |
|---------|------|------|------|------|------|------|------|------|------|------|
|         | 94.07 | 97.10 | 95.19 | 97.13 | 97.11 | 95.37 | 94.49 | 94.77 | 97.09 | 95.84 |
| PGD     | 92.69 | 74.94 | 75.91 | 67.47 | 63.75 | 70.38 | 73.53 | 78.12 | 57.16 | 71.28 |
|         | 58.12 | 73.91 | 79.59 | 67.72 | 64.72 | 72.91 | 78.16 | 76.47 | 56.62 | 70.91 |
| MetaGIA | 92.84 | 78.63 | 68.16 | 84.03 | 82.53 | 67.28 | 59.91 | 62.94 | 65.69 | 62.13 |
|         | 48.69 | 80.56 | 78.84 | 80.06 | 76.59 | 75.22 | 74.34 | 66.12 | 69.59 | 59.66 |
| AGIA    | 91.31 | 69.50 | 59.53 | 62.66 | 57.19 | 51.75 | 43.22 | 50.28 | 67.88 | 59.28 |
|         | 66.97 | 58.47 | 74.19 | 51.16 | 49.19 | 51.12 | 72.91 | 46.19 | 55.75 | 52.53 |
| TDIGA   | 91.59 | 78.09 | 73.12 | 74.00 | 68.62 | 70.34 | 64.81 | 73.00 | 65.28 | 58.84 |
|         | 66.41 | 77.22 | 73.91 | 75.72 | 71.75 | 69.72 | 64.75 | 67.97 | 65.44 | 58.09 |
| ATDGIA  | 93.03 | 73.31 | 64.25 | 68.44 | 63.59 | 64.53 | 53.66 | 57.91 | 65.12 | 62.34 |
|         | 63.34 | 73.78 | 72.53 | 69.62 | 65.00 | 65.22 | 62.97 | 65.50 | 63.66 | 57.97 |
| **AVG** | 78.10 | 75.96 | 74.11 | 72.55 | 69.09 | 68.53 | 67.52 | 67.21 | 66.30 | 64.44 |

