# OpenReview forum: "Understanding and Improving Graph Injection Attack by Promoting Unnoticeability"
_ICLR.cc/2022/Conference — ICLR 2022 Poster_

### Official Review · Reviewer_iqJd · 2021-11-02

**Correctness:** 4
**Technical Novelty And Significance:** 2
**Empirical Novelty And Significance:** 2
**Recommendation:** 6
**Confidence:** 5

**Main Review:**

Strength:
1. The paper is well written and easy to follow
2. The paper provides theoretical analysis to understand GIA and improve GIA, which is very solid
3. The proposed method makes sense and experimental results show the effectiveness of the proposed method

Weakness:
1. The authors didn’t well motivate why evasion and inductive setting is a practical setting in real-world. It is unclear what is the real-world application scenario for such setting. The authors need to provide some motivation examples.

2. The authors claim that ``nodes and edges involving test nodes are invisible to model f_\theta during training.’’ It is unclear how the authors make the test nodes not involved in the training. During training, the authors adopt 3-layer GNN, which means that the 3-hop subgraphs centered at each training nodes involved in the training of the model. The 3-hop subgraphs will contain many unlabeled nodes, which might also contain test nodes. From the description of the experiments, I didn’t find how the authors make sure that these test nodes are not involved in the subgraphs.

3. The paper assumes that the graph follows homophily assumption. What if the graph is a disassortative graph, does the proposed method still work? It would be interesting if the authors can discuss this.


**Summary Of The Paper:**

This paper first theoretically shows that graph injection attack (GIA) is more powerful than graph modification (GMA), and GIA will lead to great damage to homophily distribution, which makes GIA easily defended by homophily-based defense. To mitigate the issue, the authors further propose to add the homophily unnoticeability constraint to preserve homophily. The authors also theoretically show that with the GIA with homophily unnoticeability constraint is more powerful than GIA without the constraint.

**Summary Of The Review:**

In summary, this paper studies an important problem, provides theoretically analysis to understand GIA and further propose a new regualrizer to improve GIA. Extensive experimental results also show the effectiveness of the proposed method.

---

> ### Author Response · Authors · 2021-11-16
> **Response to Reviewer iqJd [3/3]**
>
>
> Thank you again for your constructive comments. We hope our answers would clarify your concerns and please let us know if you have any further questions.
>
> ---
> [1] Daniel Zügner, Amir Akbarnejad, and Stephan Günnemann. Adversarial Attacks on Neural Networks for Graph Data. KDD 2018.
> [2] Kurt Hornik, Maxwell B. Stinchcombe, and Halbert White. Multilayer feedforward networks are universal approximators. Neural Networks, 1989.
> [3] Ian Goodfellow, Yoshua Bengio, and Aaron Courville. Deep Learning. 2016. http://www.deeplearningbook.org.
> [4] Chiyuan Zhang, Samy Bengio, Moritz Hardt, Benjamin Recht, and Oriol Vinyals. Understanding deep learning requires rethinking generalization. ICLR 2017.
> [5] Huijun Wu, Chen Wang, Yuriy Tyshetskiy, Andrew Docherty, Kai Lu, and Liming Zhu. Adversarial Examples for Graph Data: Deep Insights into Attack and Defense. IJCAI 2019.
> [6] Wei Jin, Yao Ma, Xiaorui Liu, Xianfeng Tang, Suhang Wang, and Jiliang Tang. Graph Structure Learning for Robust Graph Neural Networks. KDD 2020.
> [7] Qinkai Zheng, Xu Zou, Yuxiao Dong, Yukuo Cen, Da Yin, Jiarong Xu, Yang Yang, and Jie Tang. Graph Robustness Benchmark: Benchmarking the Adversarial Robustness of Graph Machine Learning. NeurIPS 2021 Datasets and Benchmarks Track.
> [8] Weihua Hu, Matthias Fey, Marinka Zitnik, Yuxiao Dong, Hongyu Ren, Bowen Liu, Michele Catasta, and Jure Leskovec. Open graph benchmark: Datasets for machine learning on graphs. NeurIPS 2020.
> [9] Hongbin Pei, Bingzhe Wei, Kevin Chen-Chuan Chang, Yu Lei, and Bo Yang. Geom-GCN: Geometric Graph Convolutional Networks. ICLR 2020.
> [10] Yao Ma, Xiaorui Liu, Neil Shah, and Jiliang Tang. Is Homophily a Necessity for Graph Neural Networks? ArXiv 2021.
> [11] Jiong Zhu and Danai Koutra.Revisiting the problem of heterophily for GNNs. https://www.jiongzhu.net/revisiting-heterophily-GNNs/.

---

> ### Author Response · Authors · 2021-11-16
> **Response to Reviewer iqJd [2/3]**
>
> **Q2: About how we deal with the test nodes and edges during training.**
>
> **A2**: Basically, as we are using an inductive learning setting, we will train the GNNs only with the training graph, which is a subgraph of the original graph that only contains the nodes in the training set and edges among these nodes. While during testing, the trained GNN model will infer the labels for the test nodes given the whole graph that includes the training graph, test nodes, edges among the test nodes, and edges between test nodes and training nodes (if any). We have provided a detailed description of the setting in Appendix B in the revision.
>
> **Q3: About the disassortative graph.**
>
> **A3**: Yes, our focus in this paper is homophilous graphs, which we have highlighted in the revision. Although it is a specific class of graphs, it is also the most common class for the task of node classification, such as Cora, Citeseer, OGB [8], and GRB [7]. We also find that it’s very interesting to assess the performance of GIA attacks with or without HAO on disassortative graphs. Thus, we conduct some initial experiments on the two relatively large disassortative benchmarks, i.e., Chameleon and Squirrel provided in [9].
>
> |         | | Chameleon       |       |       | | Squirrel       |       |       |
> |:-------:|:---------:|:------:|:-----:|:-----:|:--------:|:------:|:-----:|:-----:|
> |         |    Homo   | Robust | Combo |  Avg.  |   Homo   | Robust | Combo |  Avg.  |
> |  Clean  |   61.89   |  65.18 | 64.92 | 62.58 |   37.33  |  43.88 | 45.87 | 40.04 |
> |   PGD   |   61.89   |  61.89 | 63.61 | 33.24 |   35.66  |  36.28 | 40.54 | 26.03 |
> |    +HAO |   52.78   |  57.87 | 59.31 | 38.00 |   33.32  |  39.36 | 35.83 | 26.37 |
> | MetaGIA |   61.89   |  61.89 | 63.61 | 34.38 |   35.66  |  **35.66** |  39.4 | 26.09 |
> |    +HAO |   49.25   |  55.83 | 55.73 | 33.63 |   34.07  |  38.26 | **35.24** | **25.81** |
> |   AGIA  |   61.89   |  61.89 | 63.61 | 35.95 |   35.66  |  35.89 | 39.93 | 26.93 |
> |    +HAO |   43.98   |  **48.88** | **53.33** | **32.03** |   35.69  |  36.31 |  36.4 | 26.77 |
> |  TDGIA  |   61.95   |  61.95 | 63.76 | 41.17 |   35.66  |  **35.66** | 40.81 | 29.02 |
> |    +HAO |   46.36   |  51.12 | 55.14 | 38.90 |   **31.51**  |  38.21 | 35.63 | 28.65 |
> |  ATDGIA |   61.95   |  61.95 | 63.76 | 41.11 |   35.66  |  **35.66** | 41.62 | 29.62 |
> |    +HAO |   **36.93**   |  57.75 | 59.25 | 38.88 |   32.02  |  40.00 | 40.62 | 30.24 |
> |   MLP   |  50.15   |        |       |       |  32.51    |       |       |       |
>
>
> Surprisingly, though our methods are not initially designed for disassortative graphs, we can still observe certain benefits by incorporating HAO into GIA. Then the three proposed adaptive injection methods can even improve the performance further. These results can serve as strong evidence for the generality of HAO. We refer the reasons to some recent discussions in this field [10, 11], where GNNs and GIA with HAO can still implicitly leverage the ‘homophily’ even without explicit definition, e.g., similarity among inter-class label distributions of different nodes. In the future, we believe it would be interesting and promising to further conduct an in-depth analysis of the discoveries to fully realize the potential of HAO on disassortative graphs.

---

> ### Author Response · Authors · 2021-11-16
> **Response to Reviewer iqJd [1/3]**
>
> Thank you for taking time to review our paper and for your positive comments. Please see our detailed responses to your questions and suggestions below.
>
> **Q1: About the motivation for adopting evasion and an inductive setting.**
>
> **A1**: Our motivations mainly come from practical considerations:
>
> First of all, we want to highlight an important point: **it is difficult to define the unnoticeability for graphs**. Different from images where we can adopt the inductive bias from human vision system to use numerical constraints, i.e., L-p norm, to bound the perturbation range, we cannot use similar numerical constraints to define the unnoticeability for graphs, as they are weakly correlated to the information required for node classification. For example, [1] attempts to use degree distribution changes as the unnoticeability constraint.  However, given the same degree distribution, we can shuffle the node features to generate multiple graphs with completely different semantic meanings, which would disable the functionality of unnoticeability.
>
> Because of the difficulty to properly define the unnoticeability for graphs, **adopting a poisoning setting in graph adversarial attack will enlarge the gap between research and practice**.
>
> Specifically, poisoning attacks require an appropriate definition of unnoticeability so that the defenders are able to distinguish highly poisoned data from unnoticeable poisoned data and the original data. Otherwise, attackers can always leverage some underlying shortcuts implied by the poorly defined unnoticeability, i.e., homophily in our case, to perform the attacks, since the defenders are blind to these shortcuts. On the other hand, leveraging shortcuts may generate data that is unlikely to appear in real-world applications. For example, in a citation network, medical papers are unlikely to cite or be cited by linguistic papers while the attacks may modify the graphs or inject malicious nodes to make medical papers cite or be cited by lots of linguistic papers, which is apparently impractical. Using these attacks to evaluate the robustness of GNNs may bring unreliable conclusions, i.e., homophily defenders in our case, which will greatly hinder the development of trustworthy GNNs.
>
> Moreover, under a poor unnoticeability definition, without the presence of the original data, defenders have no idea about to what extent the data is poisoned and whether the original labels remain the correspondence. And worse, because of the universal approximation power of neural networks [2], GNNs tend to overfit the training set [3] or even memorize the labels that appear during training [4], which we also validate in Fig. 5 from Appendix B.2. Thus, even trained on a highly poisoned graph, GNNs may still converge to 100% training accuracy, even though the correspondence between the data and the underlying labels might be totally corrupted. In this case, defenders can hardly distinguish whether the training graph is perturbed and are hence unlikely to make any effective defenses. Besides, studying the robustness of GNNs trained from such highly poisoned graphs seems to be impractical, since real-world trainers are unlikely to use such highly poisoned data to train GNNs.
>
> While in an evasion setting, the defenders are able to use the training graph to tell whether the incoming data is heavily perturbed and make some effective defenses, even simply leveraging some feature statistics [5, 6]. Notably, A recent benchmark [7] also has similar positions.
>
> Meanwhile, **given the evasion setting, GNNs can only perform inductive learning where the test nodes and edges are not visible during training**. The reason is that transductive learning (i.e., the whole graph except test labels is available) requires the training graph and test graph to be the same. It can not be satisfied as we will modify the test graph, i.e., changing some nodes or edges during the GMA attack, or injecting new malicious nodes during the GIA attack. Additionally, inductive learning has many practical scenarios. For example, in an academic network, the graph grows larger and larger day by day as new papers are published and added to the original network. GNN models must be inductive to be applied to such evolving graphs.
>
> Thank you again for your constructive suggestion. We have highlighted the above discussions in Appendix B in our revised version.

---

### Official Review · Reviewer_1USU · 2021-11-03

**Correctness:** 3
**Technical Novelty And Significance:** 3
**Empirical Novelty And Significance:** 3
**Recommendation:** 6
**Confidence:** 3

**Main Review:**

Overall, this paper is very well written and the experiments are also comprehensive. However, I have the following major concerns:
1. The proof of the equivalence between node modification attack and node injection attacks assumes the search space of the features of the injected node is sufficiently large such that the equality in Eq. (17) and Eq. (21) hold. However, in practice, this may not always be true as it depends on the size of the search space of injected nodes and the degree of node $v$. Even with large valid search space, the equality may still not hold because of some strange degree values of node $v$. In addition, in the paper, the valid search space (of injected nodes) is set as the minimum and maximum of all the node features in the graph. I am not sure if such an assumption is realistic in practice, as maybe some features might be harder for the attacker to manipulate. The authors should provide more evidence on why the current search space for injected nodes is realistic.
2. When showing that node injection strategies are stronger compared to node modification attacks under no defense, the paper assumes the node modification attack only performs indirect attack by changing some adjacent nodes to the test victim, but not the victim itself. However, previous work (Zugner et al., 2018) showed that direct attack (directly changing the test victim features and edges) is more powerful than the indirect attack. Therefore, the results in the paper become less interesting as there is still no clear relationship between direct node modification attacks and node injection attacks. When comparing the relationship between indirect node modification and node injection attacks, the problem also becomes somewhat trivial: when the search space of the injected node is sufficiently large, node injection will be better as the node modification attack are constrained to be within the (small) bounded $L_p$-norm ball around the original (semantically meaningful) node features.
3. I also did not follow the proof of Theorem 1. In the proof sketch, the authors say that once (a) and (b) are proved, $\mathcal{L}_{atk}(f_{\theta}(\mathcal{G}'_{GIA}))$ approaches $\mathcal{L}_{atk}^{k}(f_{\theta}(\mathcal{G}'_{GMA}))$ from below when $\Delta_{GIA}$ approaches $\Delta_{GMA}$. How come the optimal loss function value of GIA is lower than that of GMA when $\Delta_{GIA}$ approaches $\Delta_{GMA}$ from below? Having smaller perturbation budget can only increase the value of the optimal loss function value. This part needs clear explanation and I am not sure if this will lead to any changes in the statement of $\Delta_{GIA}\leq \Delta_{GMA}$ in Theorem 1. (For some reason, the math is not rendered correctly here, but the authors can past the original text into other typesetting tools to see the formula).
4. I am not sure how significant the proposed harmonious adversarial objective is. The whole optimization problem encourages each node to be as similar to the neighbors as possible while performing the attack. So, the contribution of the paper somewhat becomes spotting common weakness of node injection (and some node modification) attacks: injecting significantly different nodes to the graph and such a "unconstrained" behavior makes them vulnerable to simple homophily based detector. And then the paper enhances the original node injection attacks by adding homophily preservation as a regularizer. So the paper is more about designing adaptive attacks to a simple homophily defense and I am concerned that the contribution may not be significant.

**Summary Of The Paper:**

This paper studies the advantages and drawbacks of node injection attacks to graph neural networks. The authors demonstrate that, in general, node injection attacks are more powerful than the node modification attacks when there is no defense. But when the model trainer adopts some homophily based defense, the node injection attacks suddenly become ineffective and even underperforms the node modification attacks. Based on the observation, the authors propose to add add homophily preservation as an additional regularizer for the attack to remain stealthy against defenses. Extensive experiments demonstrate that the proposed homophily indeed improve attack effectiveness against defenses.

**Summary Of The Review:**

I am leaning towards weakly rejecting the paper because 1) node injection attacks are only shown to be stronger than indirect node modification attacks, which becomes less interesting. In addition, the equivalence proof between the node injection and modification attacks cannot always hold in practice, 2) the proposed harmonious adversarial objective is of some limited novelty.

---

> ### Author Response · Authors · 2021-11-16
> **Response to Reviewer 1USU [4/4]**
>
> Thank you again for your efforts in reviewing our paper and for your valuable comments and suggestions. Please let us know if you have any further questions.
>
> ---
> [1] KDDCUP 2020: https://www.biendata.xyz/competition/kddcup_2020_formal/.
> [2] Xu Zou, Qinkai Zheng, Yuxiao Dong, Xinyu Guan, Evgeny Kharlamov, Jialiang Lu, and Jie Tang. TDGIA: Effective injection attacks on graph neural networks. KDD 2021.
> [3] Qinkai Zheng, Xu Zou, Yuxiao Dong, Yukuo Cen, Da Yin, Jiarong Xu, Yang Yang, Jie Tang. Graph Robustness Benchmark: Benchmarking the Adversarial Robustness of Graph Machine Learning. NeurIPS 2021 Datasets and Benchmarks Track.
> [4] Daniel Zügner, Amir Akbarnejad, Stephan Günnemann. Adversarial Attacks on Neural Networks for Graph Data. KDD 2018.
> [5] Huijun Wu, Chen Wang, Yuriy Tyshetskiy, Andrew Docherty, Kai Lu, and Liming Zhu. Adversarial Examples for Graph Data: Deep Insights into Attack and Defense. IJCAI 2019.
> [6] Xiang Zhang, and Marinka Zitnik. GNNGuard: Defending Graph Neural Networks against Adversarial Attacks. NeurIPS 2020.
> [7] Wei Jin, Yao Ma, Xiaorui Liu, Xianfeng Tang, Suhang Wang, and Jiliang Tang. Graph Structure Learning for Robust Graph Neural Networks. KDD 2020.
> [8] Anish Athalye, Nicholas Carlini, and David Wagner. Obfuscated gradients give a false sense of security: Circumventing defenses to adversarial examples. ICML 2018.

---

> ### Author Response · Authors · 2021-11-16
> **Response to Reviewer 1USU [3/4]**
>
> Besides, we would also like to highlight some other contributions of our paper:
>
> 1. **The established comparison reveals a general drawback in GIA, i.e., the destruction of the homophily**. Through the established comparison (which is the first one that bridges GMA and GIA to the best of our knowledge), we know the flexibility in GIA is one of the main sources for its threats. More importantly, it also relates the optimization trajectories in graph adversarial attacks to the homophily (which is also the first one that does the job), thus revealing a general pitfall existing in current graph adversarial attacks, i.e., the destruction to the homophily. Additionally, the previously identified flexibility makes the situation worse for GIA attacks. These insights can hardly be obtained without the comparison.
>
> 2. **By relating this phenomenon to unnoticeability, we reveal the fatality of the drawback, i.e., the break of the homophily unnoticeability**. Specifically, we theoretically derive the form of homophily defenders along with a simple edge pruning implementation and verify its robustness against previous GIA methods from both theoretical and empirical perspectives. In Particular, it can defend GIA to little-to-no threats and yet it turns out that the homophily defenders actually are not robust. In other words, the break of homophily unnoticeability disables GIA’s ability to evaluate GNNs’ robustness. In contrast, it would bring unreliable conclusions that would greatly hinder the development of trustworthy GNNs. Given only previous empirical observations and intuitions [5-7], the seriousness of this pitfall could hardly be recognized without our theoretical framework and analysis.
>
> 3. **Towards mitigating the issue, we propose a novel realization of unnoticeabiilty in terms of homophily**. The existence of the fatal pitfall in GIA is mainly because they take shortcuts to maximize the damage by breaking the original homophily, since the defenders are blind to this behavior with a poorly defined homophily. However, different from images, it is difficult to leverage some inductive bias to give the definition of unnoticeability for graphs. Nevertheless, our theory points out another promising approach. Through the lens of homophily defenders derived from our theory, we can use them as external examiners to check whether the attacks satisfy the unnoticeability constraint. An additional advantage of our approach is that external examiners can also be easily incorporated into the evaluation pipeline of attackers, serving for upper bounding the performance of attacks. Besides, **the proposed realization of unnoticeability is also general**. Using the same framework, we can also instantiate the unnoticeability constraint for other domains like natural languages, where the external examiners can be implemented by checking the grammar, fluency or semantics of the underlying adversarial example.
>
> 4. **To satisfy the homophily unnoticeability constraint, we propose HAO coupled with both theoretical guarantees as well as extensive empirical validations**. The development of previous methods can hardly offer theoretical explanations for their success [5-7]. However, by bridging adversarial threats with homophily, our theory offers explicit explanations for HAO. Specifically, HAO obeys the homophily unnoticeability constraint by penalizing the shortcut behavior, hence passing the homophily examiners and capturing the underlying drawbacks of GNN models. We conduct extensive experiments with comprehensive defense models to show HAO’s generality and effectiveness. It turns out **HAO can significantly improve *all* of the GIA attacks across *all* datasets and *all* attack scenarios**. The significant improvements of HAO are largely attributed to its theoretical foundations. Additionally, due to HAO’s lightweight nature, we can easily incorporate it into existing attack methods within three steps minimally as shown in the codebase. We hope HAO for graph adversarial learning would serve a role similar to  “the criterion of  gradient obfuscation for adversarial defense [8]”.
>
> 5. Furthermore, we also propose three adaptive injection strategies to better suit HAO. **Extensive empirical analyses show that these adaptive strategies can further advance the state-of-the-art performances of GIA with HAO**, which could also provide guidelines for developing future works based on HAO.

---

> ### Author Response · Authors · 2021-11-16
> **Response to Reviewer 1USU [2/4]**
>
>
> **Q4: About the significance of the proposed harmonious adversarial objective (HAO).**
>
> **A4**: We agree that the proposed HAO is simple, but the problem it aims to tackle is important and the yielded improvements are also significant as validated by our extensive experiments. **The identified problem, i.e., the destruction of homophily unnoticeability during GIA attacks, is common while fatal to existing GIA attacks**. In particular, without considering homophily unnoticeability, GIA attacks tend to take shortcuts by destroying the homophily as much as possible to maximize its damage. If we use such attacks to evaluate the robustness of GNN models, simple designs such as the edge pruning version of homophily defenders could obtain promising results (even lower the GIA damage to little-to-no), which apparently is not a reliable conclusion and greatly hinders the process of developing trustworthy GNNs. In contrast, HAO can regularize GIA not taking the shortcut while capturing the true underlying vulnerabilities of GNNs, thus deriving more reliable conclusions about the GNNs’ robustness.
>
> **Q5: “So the paper is more about designing adaptive attacks to a simple homophily defense.”**
>
> **A5**: We respectfully disagree with this point, given the unignorable theoretical findings and significant empirical improvements. Proposing adaptive attacks is not the focus of our paper. Rather, **we want to develop GIA methods for the better and more reliable evaluation of GNNs’ robustness**. Through the lens of the comparison between GMA and GIA, we identify a fatal drawback under the empirical success of GIA, that is, the overlooking of the unnoticeability constraint in terms of homophily, which will bring unreliable conclusions about the robustness of GNNs. Thus, we shall carefully consider the homophily unnoticeability constraint when designing new attacks in the future. In this sense, **HAO is not designed only for attacking a simple homophily defense**. The main motivation for HAO is to perform attacks while not breaking the homophily unnoticeability. In this way, HAO tends to capture the underlying drawbacks of GNNs and shows significant improvements in extensive experiments.

---

> ### Author Response · Authors · 2021-11-16
> **Response to Reviewer 1USU [1/4]**
>
> Thank you for taking the time to review our paper. Please see our detailed responses to your questions and suggestions below.
>
> **Q1: About the search space for injected nodes and more evidence in practice.**
>
> **A1**: We agree that we should consider realistic applications when defining the search space. Basically, we **follow the previous GIA works [1-3] to define the constraints on the search space for the features of the injected nodes**, since proposing a new GIA setup is not the focus of this paper.
>
> For this setup, **an intuition from practical scenarios is that GIA tends to have more flexibility on the search space of the injected nodes than GMA**, since it tends to be more expensive to modify existing nodes’ features than crafting new nodes. For example, in a social network, modifying nodes’ features need to hack into the user accounts or the databases to change the profiles or relationships of the users. However, it’s cheaper and more flexible to create fake accounts with desired profiles and inject them into the existing network to perturb the predictions of GNNs.
>
> **Q2: About what strategies of the modification attack are discussed in the comparison.**
>
> **A2**: We would like to clarify that **our theoretical findings are applicable to both direct and indirect GMA attacks**. In the proof, we establish mappings that can map each kind of GMA perturbation, i.e., edge addition, edge deletion, and node modification, to a corresponding GIA perturbation with equal or fewer budgets, as the example shown in Fig. 2(a). As these perturbation actions are agnostic to direct or indirect GMA attacks, hence our theoretical results are applicable to both of them.
>
> For the empirical part, **the GMA adversary used in our empirical verification takes direct attacks**. Specifically, we evaluate GMA non-targeted attack performance in the evasion setting (reasons are as given in A1 to Reviewer ​​iqJd). Thus, under the definition of direct attack and influencer attack/indirect attack in [4], the GMA adversary can only modify edges or nodes from the test set and hence the victim nodes are also attacked nodes.  Therefore, both GMA and GIA adversaries follow the premise in Theorem 1 that they try to maximize the damage.
>
> Besides, we would like to highlight that our comparison framework is not intended to distinguish which attack is better, while we try to bridge the relatively independent attack methods in the graph adversarial literature and **enrich our understanding of their advantages and limitations**. It turns out that this comparison reveals a fatal drawback in GIA, drives us to develop a more proper definition of unnoticeability and its realizations through homophily defenders, and further motivates us to propose the harmonious adversarial objective (HAO) by taking care of the homophily unnoticeability. The importance of the unnoticeability problem and the significance of improvements brought by HAO are also recognized by Reviewer nMcc. As a starting point, the comparison motivates us to build a more reliable evaluation pipeline about the robustness of GNNs.
>
>
> **Q3: About the proof of Theorem 1.**
>
> **A3**: Thank you for pointing out this typo in our proof. It should be “approaches from the above”. We have fixed it in the revised version.
> This typo **won’t invalidate the proof**. As you also highlight that when the perturbation budget of GIA approaches to that of GMA, the optimal value of Eq. (2) achieved by GIA can not increase (b). Thus, once we find that there exists a valid budget for GIA that is lower than that of GMA, which will bring the same effects to the predictions of the GNN model (a), we know that GIA can result in at least the same harm as GMA. Since **the remaining proof of Theorem 1 focuses on proving the statement (a)**, the typo will not affect the remaining part of the proof.

---

### Official Review · Reviewer_tbGc · 2021-11-03

**Correctness:** 4
**Technical Novelty And Significance:** 4
**Empirical Novelty And Significance:** 3
**Recommendation:** 8
**Confidence:** 4

**Main Review:**

I liked the approach and thoroughness of this paper. The claims and proofs are solid. The problem addressed is very interesting in terms of pointing out a major deficiency in robustness of neural graph embedding models against GIA attacks. I believe the contribution of this work is significant in that further work can be built on their results to address the discovered vulnerability in deep graph models.

Some questions/suggestions for the authors:
-	If the classifier does not rely on the node attributes and solely uses the graph structure (e.g., k-hop message passing), it seems that GIA and HAO both lose their advantage because homophily, as defined in eq. (6), will be meaningless. In this case, GIA will be at best as powerful as GMA. How can HAO be modified to still offer a better performance than GIA/GMA in this scenario?
-	Apart from the point mentioned above, what are other limitations of your model? I feel the paper lacks this discussion.
-	Theoretically, how small can the attack budget of GIA/HAO get while still maintaining their utility?
-	Intuitively, why does HAO work? (or why do we expect it to work?) The answer to this question is important for guiding the future work in building more robust deep graph models and I believe it will be valuable to have a section discussing such intuition and possible future directions.
-	Do GIAs maintain the same advantage over GMA for different downstream tasks other than node classification (e.g., link prediction, graph clustering, etc.)?


Minor modifications:
-	First line of section 2.2 should be changed to: “… fool a GNN model, f, trained on a ...”
-	The definition of D_x in Section 2.2 is not clear. How are the min() and max() operators on the matrix X defined? What is the intuition behind D_x?
-	Last line of the paragraph after Definition 4.1 should be changed to: “…will trivially clip…”



**Summary Of The Paper:**

The paper focuses on graph injection attacks (GIA) in which an adversary introduces new nodes and links such that the predicted label by a trained classifier for a victim node is changed. They compare GIA versus graph modification attacks (GMA), in which the existing links are manipulated, and theoretically prove the superiority of GIA in causing damage. They further address the vulnerability of GIA against homophily-based defense and propose a GIA attack that preserves the homophily and is robust against such defense mechanism.

**Summary Of The Review:**

The claims and proofs are solid. The problem addressed is very interesting in terms of pointing out a major deficiency in robustness of neural graph embedding models against GIA attacks. I believe the contribution of this work is significant in that further work can be built on their results to address the discovered vulnerability in deep graph models. There is possibility of improvement (such as adding discussion on intuition behind the method, limitations of it, and in-depth discussion of implications and future directions). But, overall,  I evaluate this paper as “accept”.

---

> ### Author Response · Authors · 2021-11-16
> **Response to Reviewer tbGc [3/3]**
>
> **6. About the minor modifications.**
>
> Thank you for your detailed comments. We have revised our draft accordingly. Besides, for the definition of $D_x$ in Sec. 2.2, thank you for pointing out a confusing point in the setting description. Basically, we follow the settings of prior GIA works [7-9] and restrict the features of injected nodes to be within the minimum and maximum entries of the original feature $X$, where we use $min(X)$ and $max(X)$ to represent the minimum and maximum entries, and $D_x$ to represent the feature space, respectively.
>
>
> ---
> [1] Bryan Perozzi, Rami Al-Rfou, and Steven Skiena. DeepWalk: Online Learning of Social Representations. KDD 2014.
> [2] Yao Ma, Xiaorui Liu, Neil Shah, and Jiliang Tang. Is Homophily a Necessity for Graph Neural Networks? arXiv 2021.
> [3] Jiong Zhu and Danai Koutra.Revisiting the problem of heterophily for GNNs. https://www.jiongzhu.net/revisiting-heterophily-GNNs/ .
> [4] Shuchang Tao, Qi Cao, Huawei Shen, Junjie Huang, Yunfan Wu, and Xueqi Cheng. Single Node Injection Attack against Graph Neural Networks. CIKM 2021.
> [5] Wei Jin, Yaxing Li, Han Xu, Yiqi Wang, Shuiwang Ji, Charu Aggarwal, and Jiliang Tang. Adversarial attacks and defenses on graphs. SIGKDD Explorations Newsletter 2021.
> [6] Lichao Sun, Yingtong Dou, Carl Yang, Ji Wang, Philip S. Yu, and Bo Li. Adversarial attack and defense on graph data: A survey. arXiv 2018.
> [7] KDDCUP 2020: https://www.biendata.xyz/competition/kddcup_2020_formal/.
> [8] Xu Zou, Qinkai Zheng, Yuxiao Dong, Xinyu Guan, Evgeny Kharlamov, Jialiang Lu, and Jie Tang. TDGIA: Effective injection attacks on graph neural networks. KDD 2021.
> [9] Qinkai Zheng, Xu Zou, Yuxiao Dong, Yukuo Cen, Da Yin, Jiarong Xu, Yang Yang, and Jie Tang. Graph Robustness Benchmark: Benchmarking the Adversarial Robustness of Graph Machine Learning. NeurIPS 2021 Datasets and Benchmarks Track.

---

> > ### Comment · Reviewer_tbGc · 2021-11-22
> > **Rebuttal Response**
> >
> > I thank authors for the great effort they have put during the rebuttal phase to answer all reviewers' concerns, including mine. They helped me understand their motivation and approach better and their revisions have addressed the lack of discussion on the limitations of the model.

---

> > > ### Author Response · Authors · 2021-11-23
> > > **Thank You**
> > >
> > > We gratefully appreciate your efforts in reviewing our paper, your insightful comments, and your support! We will continue to refine our work and develop more solid extensions based on these valuable comments.

---

> ### Author Response · Authors · 2021-11-16
> **Response to Reviewer tbGc [2/3]**
>
> **3. Why do we expect HAO to work? Its limitations, the trade-off phenomenon, and further implications.**
>
> The main reason why HAO would work is that it constrains the attack to obey a more proper definition of unnoticeability, instead of taking the shortcut to incur damage by destroying the original homophily. To be more specific, due to the poor unnoticeability constraint for graph adversarial learning, the developed attacks are likely to leverage the shortcuts that maximize the damage by greatly destroying the original homophily. In GIA, this effect is further amplified due to its flexibility. Thus, using homophily defenders can easily defend these seemingly powerful attacks, even with a simple design, which however brings us unreliable conclusions about the robustness of homophily defenders. Essentially, HAO penalizes GIA attacks that take shortcuts and promotes their unnoticeability in terms of homophily. Thus, HAO mitigates the shortcut issue of GIA attacks, urges the attacks to capture the underlying vulnerability of GNNs, and brings us a more reliable evaluation of GNNs’ robustness, where we know that simple homophily defenders are essentially not robust GNNs. Additionally, we believe the proposed realization of unnoticeability also serves as a promising paradigm to instantiate the unnoticeability for other domains such as natural languages.
>
> In terms of the limitations of HAO, since HAO is mostly developed to preserve the homophily unnoticeability, it counters the greedy nature of attacks without HAO that destroys the homophily to incur more damage. As also observed from the experiments, we find HAO essentially trades the attack performance when against vanilla GNNs for the performance when against homophily defenders. As shown in Fig. 3(a), the trade-off effects can be further amplified with a large coefficient lambda in HAO. As shown in Fig 4(b) and Fig. 4(c), when against vanilla GNNs, compared with GIA without HAO, GIA with HAO show fewer threats. In certain cases, the trade-off might generate the performance of attacks. Thus, it calls for more tailored optimization methods to solve for better injection matrix and node features in the future.
>
> We also provide a detailed discussion in Appendix A. Please find more details in our revised version.
>
> **4. About applications to more downstream tasks.**
>
> The reason for only considering the semi-supervised node classification task is mainly to follow the previous works [5, 6]. We may consider more applications of graph adversarial attacks on other downstream tasks from both theoretical and empirical perspectives.
> Specifically, from the theoretical side, we believe most of the results derived from the comparison can still be applied to other tasks, since many existing approaches still rely on the learned neural graph embeddings and the underlying homophily presented in data to perform the task. For the other analyses that are based on the optimization trajectories, since different tasks rely on different objectives, in-depth analyses of the interplay between homophily and the adversarial threats present more challenges, which we believe would require an extensive amount of effort and is out of the scope of this work, but we will certainly consider it for future work.
>
>
> **5. Further discussions about existing limitations and future works.**
>
> As far as we know, most graph adversarial studies [5, 6] only focus on relatively shallow GNNs. Different from other deep learning models, as GNNs go deep, besides more parameters, they also require an exponentially growing number of neighbors as inputs. How the number of layers would affect their robustness and the threats of attacks remains unexplored. From both theoretical and empirical perspectives, we believe it’s very interesting to study the interplay between the number of GNN layers and homophily, in terms of adversarial robustness and threats, and how to leverage the discoveries to explore the weakness of more complicated GNN models.
>
> We also offer more discussions on the future directions implied by HAO. Please find more details in Appendix A of our revised version.

---

> ### Author Response · Authors · 2021-11-16
> **Response to Reviewer tbGc [1/3]**
>
> Thank you for your detailed comments and constructive suggestions. Please see our detailed responses to your questions and suggestions below (we reorganize the suggestions/questions a bit for clarity).
>
> **1. Applying GIA/HAO to non-attributed graphs, and other classes of graphs.**
>
> Thank you for pointing out this promising direction. Although the current version of our paper does not focus on this specific class of graphs, we believe studying non-attributed graphs or other classes of graphs is a promising future direction, and we hope our discussion below could address your question.
>
> As non-attributed graphs do not have features, neural graph models have to make predictions according to the structural information and the given label information. If the success of neural graph models on this specific class of graphs still adheres to the spirit of homophily, i.e., “nodes from the same class have certain structural patterns”, we can also define its “homophily” by leveraging this intuition. For example, we can use deepwalk embeddings [1] as node features to resemble the intuition that “nodes from the same class are more likely to appear in the same run of random walk”. Thus, we can adopt the same framework proposed in our paper to constrain the attacks to remain homophily unnoticeable.
>
> For other classes of graphs, such as disassortative graphs as mentioned by Reviewer i1Jd, though our framework is not originally designed for these graphs, we can still observe certain empirical improvements brought by HAO. The reason might be that GNNs and GIA with HAO can still implicitly leverage the “homophily” even without explicit definition (e.g., the similarity of inter-class label distribution as discussed in [2, 3]).
>
> In terms of the comparison between GMA and GIA’s threats, we believe it’s a very important question. Since there are few previous works in this direction, we might need new setups for the comparison to ensure its non-triviality and we would like to leave it to future work.
>
>
> **2. Attacking with small budgets.**
>
> We appreciate this interesting point. Theoretically, given a small budget for GMA and GIA, e.g. allowing GMA to modify one node or edge, and GIA to inject one node, respectively, when without HAO and without any defenses, GIA can still maintain the superiority as our theories do not depend on the budget size. With HAO, GIA might be less powerful when against vanilla GNN models,  while GIA can still be more powerful than GMA when against homophily defenses. The upper limit of GIA and HAO, in this case, could depend on the homophily distribution and degree distribution of the original graph, which is also reflected during the proof of the certified robustness of homophily defenders (Proposition E.1) in Appendix E.6.
>
> From the empirical side, given a certain small number of budgets, as shown in Fig. 4(b) and Fig. 4(c), GIA with HAO can still maintain excellent utility. Interestingly, we notice there is a recent work [4] providing strong empirical evidence about the threats of GIA given only one node budget. We are interested to get more results and deriving more theoretical explanations in our future work.

---

### Official Review · Reviewer_nMcc · 2021-11-03

**Correctness:** 3
**Technical Novelty And Significance:** 3
**Empirical Novelty And Significance:** 4
**Recommendation:** 6
**Confidence:** 3

**Main Review:**

It is an interesting research topic to enhance the unnoticeability of graph adversarial attack and bypass various defenders. The proposed attack is straightforward and empirically effective against different defense strategies. The performance proposed method is also impressive under the combination of different defense methods.  However, there exist several concerns:

1. For the analysis on  the comparison between GMA and GIA (Theorem 1), the budgets of these two attacks are totally different and not directly comparable. The budget of GMA is measured by the changes in entries of adjacency matrix and feature matrix while that of GIA is measured by the number of nodes.  If we only consider the number of perturbed nodes, the budget of GMA would be much smaller and Theorem 1 may not hold.  Since GIA can inject nodes with arbitrary features, it is intuitively more flexible and harmful than GMA.
2. The novelty of this paper is somewhat weak. It is widely received that attackers tends to connect dissimilar features and many defenders are designed to mitigate such influence. Thus, it is quite natural to circumvent those defenders by penalizing the homophily change brought by the perturbations. I think the main contribution of this paper is from the empirical side.


**Summary Of The Paper:**

This paper studies the problem of adversarial attack in graph neural networks. It aims to improve the unnoticeability of graph injection attack which injects carefully-crafted nodes into the graph data. In detail, it proposes the concept of homophily unnoticeability and studies the power of graph injection attack (GIA). Further, it observes that GIA can be easily defended by homophily defenders and proposes a harmonious adversarial objective to preserve homophily. Extensive experiments have demonstrated the effectiveness of the proposed method under various defenders.

**Summary Of The Review:**

This paper advances the field of graph adversarial attack by promoting homophily unnoticeability. The experimental results are impressive while there are still some concerns regarding some analysis.

---

> ### Author Response · Authors · 2021-11-16
> **Response to Reviewer nMcc**
>
> We appreciate your efforts in reviewing our paper and your positive feedback about our work. Please see our detailed responses to your comments and suggestions below.
>
>
>
> **Q1: The budgets of them are totally different and not directly comparable.**
>
> **A1**: We agree that the different settings of GMA and GIA in the literature make a direct comparison hard. However, as these two attacks have their reasonable constraints of perturbations established in previous works [1-3] which depict their unique behaviors, studying their maximum possible harm when taking optimal strategies can reveal more about their intersections and distinctions, such as how their unique actions will contribute to their threats. Thus, the comparison could further enrich our understanding about their advantages and limitations. Starting from the comparison, we identify a fatal drawback in GIA and propose solutions that could substantially improve the utility of GIA for the evaluation of GNNs’ robustness, as validated both theoretically and empirically.
>
>
> **Q2: The novelty of this paper is somewhat weak. It is widely received… I think the main contribution of this paper is from the empirical side.**
>
> **A2**: We believe without the theoretical analyses, it would be difficult to understand the significance of the problem beyond empirical observations and to **identify the connection between this phenomenon and the homophily unnoticeability** in GIA. In fact, many previous works have focused on empirical analyses and developed robust methods based on the empirical observations (see [4-6] below), but they offer limited theoretical explanations. In contrast, starting from the comparison between GMA and GIA, we a) theoretically identify that the flexibility of GIA is the source of its threats at the sacrifice of homophily unnoticeability, b) reveal the necessity of taking care of homophily unnoticeability during GIA attack through the lens of homophily defenders derived theoretically, and c) propose a simple and yet effective objective for GIA to instantiate the homophily unnoticeability from a theoretical perspective. We would like to highlight that all of the above three points are not covered in the literature.
>
> Thank you again for taking the time to review our paper. We hope our responses could clarify your concerns and please let us know if you have any further questions.
>
>
> ---
>
> [1] Daniel Zügner, Amir Akbarnejad, and Stephan Günnemann. Adversarial Attacks on Neural Networks for Graph Data. KDD 2018.
> [2] Xu Zou, Qinkai Zheng, Yuxiao Dong, Xinyu Guan, Evgeny Kharlamov, Jialiang Lu, and Jie Tang. TDGIA: Effective injection attacks on graph neural networks. KDD 2021.
> [3] Qinkai Zheng, Xu Zou, Yuxiao Dong, Yukuo Cen, Da Yin, Jiarong Xu, Yang Yang, and Jie Tang. Graph Robustness Benchmark: Benchmarking the Adversarial Robustness of Graph Machine Learning. NeurIPS 2021 Datasets and Benchmarks Track.
> [4] Huijun Wu, Chen Wang, Yuriy Tyshetskiy, Andrew Docherty, Kai Lu, and Liming Zhu. Adversarial Examples for Graph Data: Deep Insights into Attack and Defense. IJCAI 2019.
> [5] Xiang Zhang, and Marinka Zitnik. GNNGuard: Defending Graph Neural Networks against Adversarial Attacks. NeurIPS 2020.
> [6] Wei Jin, Yao Ma, Xiaorui Liu, Xianfeng Tang, Suhang Wang, and Jiliang Tang. Graph Structure Learning for Robust Graph Neural Networks. KDD 2020.

---

### Author Response · Authors · 2021-11-16
**We have uploaded a revised version [Updated on 22 Nov.]**

Dear reviewers,

We have revised our paper following the suggestions/comments from all the reviewers. Some changes are:

- Section 2, Page 3: we refine the sentences to elaborate the constraints on feature space, following previous works (Zheng et al., 2021);
- Appendix A, Page 14-15: we add additional discussions on our methods, its limitations and future implications, following Reviewer tbGc’s suggestions;
- Appendix B, Page 16-17: we elaborate the motivations for our inductive and evasion setting, following Reviewer iqJd’s comments;
- Appendix C, Page 18: we highlight that our theoretical discussions are applicable to both direct and indirect attacks, while we evaluate direct GMA attacks during our experiments, following Reviewer 1USU’s comments;
- Appendix I, Page 34-35: we add the initial experimental results of HAO on two disassortative graphs, following Reviewer iqJd’s comments;

We also fixed some typos that will not affect the other findings and results, such as fixing “approaches from below” to “approaches from above” in the proof for Theorem 1, following Reviewer 1USU’s comments.

Besides, we also provided a link of our codes for reproducing the results in our paper: https://anonymous.4open.science/r/GIA-HAO-05A6/ .

== Additional Updates on 22 Nov. ==
- Section 2, Page 3: we make further refinements about the constraints on feature space, following previous works (Zheng et al., 2021), according to Reviewer 1USU’s comments;
- Appendix A, Page 15: we provide more discussions on future works about reinforcement learning based approaches, following Reviewer 1USU’s comments;
- Appendix C, Page 18: we explicitly highlight (a) holds for direct perturbations on target node features, following Reviewer 1USU’s comments;
- Appendix C, Page 18 and Page 23: we explicitly highlight (a) holds for direct perturbations on target node features, following Reviewer 1USU’s comments;
- Appendix H, Page 32: we give more discussions on the selection of attack baselines, following Reviewer 1USU’s comments;

We again thank all reviewers for their efforts and many helpful comments/suggestions.

---

### Decision · Program_Chairs · 2022-01-20

**Decision:**

Accept (Poster)

**Comment:**

The reviewers agree that this paper studies an important problem, provides theoretically analysis to understand graph injection attack.
The authors propose a new regularizer to improve the attack success. Extensive experimental results also show the effectiveness of the proposed method.